**A grid dataset of leaf age-dependent LAI seasonality product (Lad-**
**LAI) over tropical and subtropical evergreen broadleaved forests**
Xueqin Yang[1,2,3], Xiuzhi Chen[1,*], Jiashun Ren[1,4], Wenping Yuan[1], Liyang Liu[5], Juxiu
Liu[6], Dexiang Chen[7], Yihua Xiao[7], Shengbiao Wu[8], Lei Fan[9], Xiaoai Dai[4], Yunpeng
Wang[3], and Yongxian Su[2]
[1]Guangdong Province Data Center of Terrestrial and Marine Ecosystems Carbon Cycle,
Guangdong Province Key Laboratory for Climate Change and Natural Disaster Studies,
School of Atmospheric Sciences, Sun Yat-sen University & Southern Marine Science
and Engineering Guangdong Laboratory (Zhuhai), Zhuhai 519082, China
[2]Key Lab of Guangdong for Utilization of Remote Sensing and Geographical
Information System, Guangdong Open Laboratory of Geospatial Information
Technology and Application, Guangzhou Institute of Geography, Guangdong Academy
of Sciences, Guangzhou 510070, China
[3]Guangzhou Institute of Geochemistry, Chinese Academy of Sciences, Guangzhou,
510640, China
[4]College of Earth Sciences, Chengdu University of Technology, Chengdu 610000,
China;
[5]Laboratoire des Sciences du Climat et de l'Environnement, IPSL, CEA-CNRS-UVSQ,
Université Paris-Saclay, 91191 Gif sur Yvette, France
[6]Dinghushan Forest Ecosystem Research Station, South China Botanical Garden,
Chinese Academy of Sciences, Guangzhou 510650, China;
[7]Pearl River Delta Forest Ecosystem Research Station, Research Institute of Tropical
Forestry, Chinese Academy of Forestry, Guangzhou 510650, China;
[8]School of Biological Sciences, The University of Hong Kong, Pokfulam, Hong Kong
[9]Chongqing Jinfo Mountain Karst Ecosystem National Observation and Research
Station, School of Geographical Sciences, Southwest University, Chongqing 400715,
China
*Correspondence: Xiuzhi Chen (chenxzh73@mail.sysu.edu.cn)*

## Abstract

Quantification of large-scale leaf age-dependent leaf area index has been lacking in tropical and subtropical evergreen broadleaved forests (TEFs) despite the recognized importance of leaf age in influencing leaf photosynthetic capacity in this biome. Here, we simplified the canopy leaves of TEFs into three age cohorts (i.e., young, mature and old one with different photosynthesis capacity ($V_{c,max}$)) and proposed a novel neighbor-based approach to develop a first grid dataset of monthly leaf age-dependent LAI product (**referred to as Lad-LAI**) at 0.25-degree spatial resolution over the continental scale during 2001-2018 from satellite observations of sun-induced chlorophyll fluorescence (SIF) that was reconstructed from MODIS and TROPOMI (the TROPOspheric Monitoring Instrument). The new Lad-LAI products show good performance in capturing the seasonality of three LAI cohorts, i.e., young ($LAI_{young}$) (R=0.36), mature ($LAI_{mature}$) (R=0.77) and old ($LAI_{old}$) (R=0.59) leaves, at the eight sites (four in south America, three in subtropical Asia and one in Congo) and can also represent their interannual dynamics at the Barrocolorado site, with R being equal to 0.54, 0.64 and 0.49 for $LAI_{young}$, $LAI_{mature}$ and $LAI_{old}$, respectively. Additionally, the abrupt drops in $LAI_{old}$ are mostly consistent with the seasonal litterfall peaks at 53 *in situ* measurements across the whole tropical region (R=0.82). The LAI seasonality of young and mature leaves also agrees well with the seasonal dynamics of Enhanced Vegetation Index (EVI) (R=0.61), which is a good proxy of effective leaves. Spatially, the grid Lad-LAI captures a dry-season green-up of canopy leaves across the wet Amazonia areas where mean annual precipitation exceeds 2,000 mm $yr^{-1}$, consistent with previous satellite-based analyses. The spatial patterns clustered from the three LAI cohorts also coincide with those clustered from climatic variables over the whole TEF region. The seasonality of $LAI_{young}$, $LAI_{mature}$ and $LAI_{old}$ derived from the estimated GPP based on a simple linear SIF-GPP relationship show the highest correlation with the *in situ* measurements at 8 observed sites compared with those derived from Orbiting Carbon Observatory-2-based SIF (GOSIF) GPP and eddy covariance flux tower

measurements (FLUXCOM) GPP. Additionally, the Lad-LAI products developed by the neighbor-based approach using 2*2 and 4*4 neighboring pixels show stable seasonality in $LAI_{young}$, $LAI_{mature}$ and $LAI_{old}$ across the whole tropical region, respectively. We provide the average seasonality of three LAI cohorts as the main dataset, and their time-series as a supplementary dataset. These two products are available at https://doi.org/10.6084/m9.figshare.21700955.v3 (Yang et al., 2022).

## 1. Introduction

Tropical and subtropical evergreen broadleaved forests (TEFs) account for approximately 34% of global terrestrial primary productivity (GPP) (Beer et al., 2010) and 40-50% of the world's gross forest carbon sink (Pan et al., 2011; Saatchi et al., 2011). Despite a perennial canopy, TEFs shed and rejuvenate their leaves continuously throughout the year, leading to significant seasonality in canopy leaf demography (Wu et al., 2016; Chen et al., 2021). This phenological changes in leaf demography is the primary cause of GPP seasonality in TEFs (Saleska et al., 2003; Sayer et al., 2011; Leff et al., 2012) and thus largely regulates their seasonal carbon sinks (Beer et al., 2010; Aragao et al., 2014; Saatchi et al., 2011).

A key plant trait linking canopy phenology with GPP seasonality was shown to be leaf age (Wu et al., 2017; Xu et al., 2017). At leaf scale, the newly-flushed young leaves and maturing leaves show higher maximum carboxylation rates ($V_{c,max}$) than the old leaves being replaced (De Weirdt et al., 2012; Chen et al., 2020). Such age-dependent variations in $V_{c,max}$ is associated with changes in leaf nutritional contents (nitrogen, phosphorus and potassium etc.) and stomatal conductance over time (Menezes et al., 2021). Xu et al. (2017) and Menezes et al. (2021) monitored *in situ* leaf age and leaf demography combined with leaf-level $V_{c,max}$ in Amazonian TEFs and found that $V_{c,max}$ of newly-flushed leaves increases rapidly with leaf longevity, peaks at approximately 2-month old and then declines gradually as leaf grows older (leaf age > 2 months). At canopy scale, it was hypothesized that leaf demography and seasonal differences in leaf

age compositions of tree canopies control the GPP seasonality in TEFs (Wu et al., 2016;
Albert et al., 2018). Similar mechanism was also observed by the ground-based LiDAR
which showed an increasing trend in upper canopy leaf area index (LAI) during the dry
season, whereas a decrease in lower canopy LAI (more old leaves) (Smith et al., 2019).
Wu et al. (2016) classified canopy leaves of Amazonian TEFs into three leaf age cohorts
(young: 1-2 months, mature: 3-5 months and old: $\geq 6$ months). LAI of young and mature
leaves increases during the dry seasons and consequently promotes dry-season canopy
photosynthesis. Based on above age-dependent $V_{c,max}$ at leaf scale (Xu et al., 2017) and
LAI seasonality of different leaf age cohorts at canopy scale (Wu et al., 2016), Chen et
al. (2020; 2021) developed a climate-triggered leaf litterfall and flushing model and
successfully represented the seasonality of canopy leaf demography and GPP at four
Amazonian TEF sites. Overall, leaf age-dependent LAI seasonality is one of the vital
biotic factors in influencing the GPP seasonality in TEFs (Wu et al., 2016; Chen et al.,

98     2020).

99        Although the leaf age-dependent LAI seasonality can be well documented at site

level using phenology cameras (Wu et al., 2016), it is still rarely studied and remains
unclear at the continental scale. The key causation is that leaf flushing and litterfall of
TEFs in different climatic regions experience different seasonal constraints of water
and light availability during recurrent dry and wet seasons (Brando et al., 2010; Chen
et al., 2020; Davidson et al., 2012; Xiao et al., 2005). Thus, the seasonal patterns of LAI
in different leaf age cohorts become very complex at the continental scale (Chen et al.,
2020; Xu et al., 2015). Satellite-based remote sensing (Saatchi et al., 2011, Guan et al.,
2015) and land surface model (LSM) technologies (De Weirdt et al., 2012; Chen et al.,
2020; 2021) are two commonly used approaches for detecting the spatial heterogeneity
of plant phenology at a large scale. However, for satellite-based studies, most optical
signals are saturated in TEFs due to the dense covered canopies and thus fail to capture
the seasonality of total LAI in TEFs, much less decompose the LAI into different leaf
age cohorts. These limitations prevent satellite-based studies from accurately

representing the age-dependent LAI seasonality. Moreover, most ESM models also show poor performances in simulating the LAI seasonality in different leaf age cohorts (De Weirdt et al., 2012; Chen et al., 2020). This is because that the underling mechanisms linking seasonal water and light availability with leaf flushing and litterfall seasonality are currently highly debated and remain elusive at regional scale (Leff et al., 2012; Saleska et al., 2003; Sayer et al., 2011). This vague notion imposes a challenge for accurately modeling continental-scale GPP seasonality in most LSMs (Restrepo-Coupe et al., 2017; Chen et al., 2021).

To fill the research gap, this study aims to produce a global grid dataset of leaf age-dependent LAI seasonality product (Lad-LAI) over the whole TEF biomes from 2001 to 2018. For this purpose, we first simplified that canopy GPP was composed of three parts that were produced from young, mature and old leaves, respectively. GPP was then expressed as a function of the sum of the product of each LAI cohort (i.e., young, mature and old leaves, denoted as $LAI_{young}$, $LAI_{mature}$, and $LAI_{old}$, respectively) and corresponding net $CO_2$ assimilation rate (An, denoted as $An_{young}$, $An_{mature}$, and $An_{old}$ for young, mature and old leaves, respectively) (**Equation 1**). Then, we proposed a novel neighbor-based approach to derive the values of three LAI cohorts. It was hypothesized that forests in adjacent four cells in the grid map exhibited consistent seasonality in both GPP, and LAI cohorts ($LAI_{young}$, $LAI_{mature}$, and $LAI_{old}$). Based on this assumption, we applied **Equation 1** to each pixel and combined the four equations of 2*2 neighboring pixels to derive the three LAI cohorts using a linear least-squares with constrained method. An was calculated using the Farquhar-von Caemmerer-Berry (FvCB) leaf photochemistry model (Farquhar et al., 1980); and GPP was linearly derived from an arguably better proxy—TROPOMI (the TROPOspheric Monitoring Instrument) Solar-Induced Fluorescence (SIF) based on a simple SIF-GPP relationship established by Chen et al. (2022) (see **Methods** for details). This grid dataset of three LAI cohorts provides new insights into tropical and subtropical phenology with more details of sub-canopy level of leaf seasonality in different leaf age cohorts and will be

helpful for developing accurate tropical phenology model in ESMs.

## 2. Study area and material

### 2.1 Tropical and subtropical evergreen broadleaved forest biomes

In this study, we focused on the whole tropical and subtropical evergreen broadleaf
forests (TEFs). The pixels labeled TEFs according to the International Geosphere-
Biosphere Program (IGBP) classification were extracted as the study area based on the
0.05° spatial resolution MODIS land cover map (**Fig. 1**) (MCD12C1, Sulla-Menashe et
al., 2018). The study area contains three regions: South America (30°S–18°N; 40°W–
90°W), the world's largest and most biodiverse tropical rain forest, Congo (10°S–10°N;
10°W−30°E), the western part of the Africa TEF region, and Tropical Asia (20°S–30°N;
70°E−150°E), covering the Indo-China Peninsula, the majority of the Malay
Archipelago and the northern Australia.

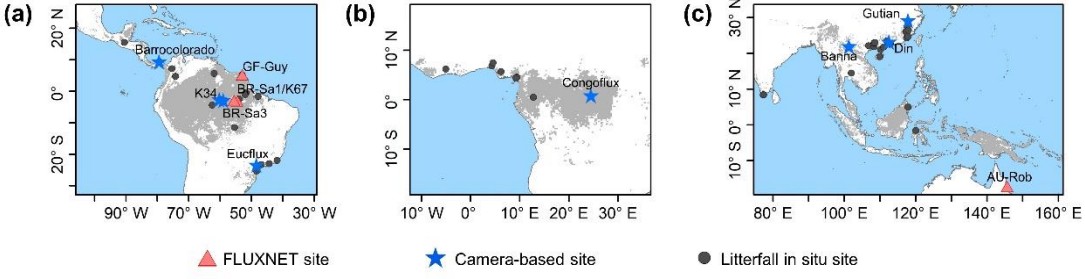


**Figure 1.** Study areas over tropical and subtropical evergreen broadleaves forests (TEF).
Red triangles: observed GPP seasonality at four eddy covariance (EC) tower sites. Blue
pentangles: observed LAI cohorts at eight camera-based observation sites. Black circles:
observed litterfall seasonality at 53 observation sites.

### 2.2 Input datasets for calculating GPP and An parameters

The TROPOspheric Monitoring Instrument (TROPOMI) Solar-Induced
Fluorescence (SIF) data were used to derive the continent-scale GPP (denoted as
RTSIF-derived GPP) according to the SIF-GPP relationship established by Chen et al.
(2022) which used 15.343 as a transformation coefficient to covert SIF to GPP. The air

temperature data from ERA5-Land (Zhao, Gao et al., 2020), vapor pressure deficits (VPD) data from ERA-Interim (Yuan et al., 2019) and downward shortwave solar radiation (SW) from Breathing Earth System Simulator (BESS) (Ryu et al., 2018) were used to calculate $K_C$, $K_O$, $\Gamma^*$, $R_{dark}$ and $V_{c,max}$ and thus to calculate An according to equations in **Table S4** . The calculation processes were illustrated in **Fig. 2**. All datasets were aggregated at the same spatial (0.125°) and temporal resolutions (month) (**Table S3**).

**2.3 Datasets for validating leaf age-dependent LAI seasonality**

**Ground-based seasonal LAI cohorts and litterfall data**. Top-of-canopy imageries observed by ground-based phenology cameras were used to decompose canopy LAI into $LAI_{young}$, $LAI_{mature}$ and $LAI_{old}$. In total, imageries from eight observation sites across the whole TEF region were used to validate the simulating results (blue pentangles in **Fig. 1**, **Table S1**). Additionally, the seasonal litterfall data from 53 *in situ* sites (black circles in **Fig. 1**, **Table S6**) spanning the TEFs were collected from globally published articles to compare with the phase of simulated $LAI_{old}$ seasonality (see **Methods** for details). The multiyear monthly litterfall data were averaged to the monthly mean to compare with the seasonality of simulated $LAI_{old}$. Four eddy covariance flux tower sites (red triangles in **Fig. 1**, **Table S2**) provided *in situ* seasonal GPP data to evaluate the seasonality of RTSIF-derived GPP.

**Satellite-based seasonal EVI data**. To evaluate the LAI seasonality of photosynthesis-effective leaves, i.e., young and mature leaves, this study used satellite-based MODIS Enhanced Vegetation Index (EVI) (Huete et al., 2002; Lopes et al., 2016; Wu et al., 2018) as a remotely sensed proxies alternatives of effective leaf area changes and new leaf flush, i.e., $LAI_{young+mature}$ (Wu et al., 2016; Xu et al., 2015). To prove the robustness of the products over a large spatial coverage, the seasonal LAI cohorts of young and mature leaves were evaluated against the enhanced vegetation index (EVI) product, which was considered as a proxy for leaf area changes of photosynthetic

effective leaves (Xu et al., 2015; Wu et al., 2016; de Moura et al., 2017).

## 3. Methods
### 3.1 Decomposing LAI cohorts (young, mature and old) from SIF-derived GPP
**Figure 2** illustrates the overall framework used to generate leaf age-dependent LAI
seasonality product (Lad-LAI). The majority of the tropical and subtropical EBFs retain
leaves year-round and their total LAI shows marginally small spatial and seasonal
changes (Wu et al., 2016) (**Figs. S3, S4**). Therefore, previous modelling studies have
assumed a constant value for the total LAI in tropical and subtropical EBFs (Cramer et
al., 2001; Arora and Boer, 2005; De Weirdt et al., 2012). Based on this, we collected
observed seasonal LAI dynamics in tropical and subtropical EBFs from previously
published literatures which showed a constant value of LAI around 6.0 (**Figs. S3**, **S4**,
**Table S5**). Thus, in this study, we simplified to assume that the seasonal LAI was
approximately equaling to 6.0 in tropical and subtropical EBFs. We grouped the canopy
leaves of tropical and subtropical EBFs into three leaf age cohorts, i.e., young, mature
and old leaves, respectively. Then, the total GPP was defined as the sum of those
produced by the young, mature and old leaves, respectively. According to the Farquhar-
von Caemmerer-Berry (FvCB) leaf photochemistry model (Farquhar et al., 1980), GPP
can be expressed as function of the sum of the products of each LAI cohort ($LAI_{young}$,
$LAI_{mature}$, and $LAI_{old}$) and corresponding net $CO_2$ assimilation rate ($An_{young}$, $An_{mature}$,
and $An_{old}$) (**Equation 1**).
$$GPP = LAI_{young} \times An_{young} + LAI_{mature} \times An_{mature} + LAI_{old} \times An_{old} \quad (1)$$
where $LAI_{young}$, $LAI_{mature}$ and $LAI_{old}$ are the leaf area index of young, mature and old
leaves, respectively; $An_{young}$, $An_{mature}$ and $An_{old}$ are the net rate of $CO_2$ assimilation
dependent on three leaf age classes; GPP is canopy total gross primary production. The
sum of $LAI_{young}$, $LAI_{mature}$ and $LAI_{old}$ was set as a constant in this study, equaling to 6.0.
The grid GPP data over the whole EBFs were derived from SIF (denoted as RTSIF-
derived GPP) using a linear SIF-GPP regression model (see sect. 3.2) which were

established based on *in situ* GPP from 76 eddy covariance (EC) sites (Chen et al., 2022). The $An_{young}$, $An_{mature}$ and $An_{old}$ were calculated according to the FvCB biochemical model (Farquhar et al., 1980; Bernacchi et al., 2003) (see section 3.3). As there were three unknow variables (i.e., $LAI_{young}$, $LAI_{mature}$ and $LAI_{old}$) to be solved in **Equation 1**, we hypothesized that the adjacent four pixels exhibited homogenous EBFs and consistent leaf demography and canopy photosynthesis. Then, we used the GPP and An data from adjacent four pixels to estimate their $LAI_{young}$, $LAI_{mature}$ and $LAI_{old}$ based on **Equation 1** using a linear least-squares with constrained method. The inputs grid datasets (i.e., RTSIF-derived GPP and An derived from $T_{air}$, VPD and SW) (**Table S3, Fig. 2**) were sampled at 0.125-degree spatial resolution; while the output maps of $LAI_{young}$, $LAI_{mature}$, and $LAI_{old}$ were at 0.25-degree spatial resolution. Therefore, the output maps of $LAI_{young}$, $LAI_{mature}$, and $LAI_{old}$ were at a 0.25-degree spatial resolution. Additionally, to test the robustness of the neighbor-based decomposition approach, we increased the number of adjacent pixels from 4 (2*2) to 16 (4*4) to produce another version of Lad-LAI products with spatial resolution of 0.5-degree. All our analyses were conducted using the Python (version 3.7, http://www.python.org) and Matlab (version R2019b) software.

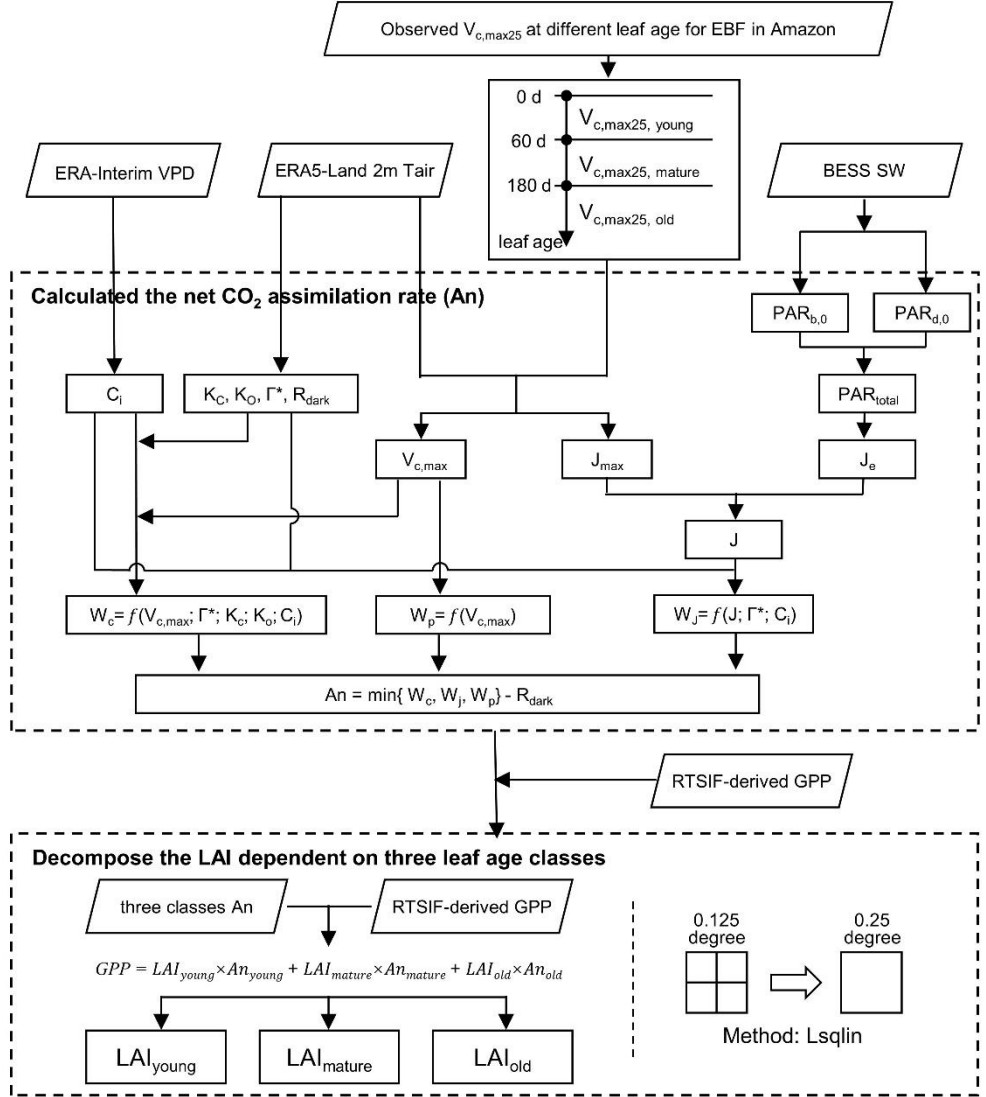

238

**Figure 2.** The workflow for mapping Lad-LAI using the Lsqlin method. Lsqlin is the

abbreviation of Linear least-squares solver with bounds or linear constraints. All the

abbreviations were described in supplementary **Tables S4**.

242

**3.2 Calculating the GPP (RTSIF-derived GPP) from TROPOMI SIF**

Satellite-retrieved solar-induced chlorophyll fluorescence (SIF) is a widely used

proxy for canopy photosynthesis (Yang et al., 2015; Dechant et al., 2020). Here, we

used a long-term reconstructed TROPOMI SIF dataset (RTSIF) (Chen et al., 2022) to

estimate GPP seasonality. Previous analyses showed that RTSIF was strongly linearly

correlated to eddy covariance (EC) GPP and used 15.343 as a transformation coefficient

to covert RTSIF to GPP (Fig. 8a in Chen et al., 2022). In this study, we followed
previously published literatures to set a constant value of LAI around 6.0 for the whole
tropical and subtropical EBFs (**Figs. S3**, **S4**, **Table S5**). We collected seasonal GPP data
observed at four EC sites from the FLUXNET 2015 Tier 1 dataset (**Table S2**; Pastorello
et al., 2020) and validated the Chen's simple SIF-GPP relationship (**Fig. S1**). Results
confirmed the robustness of Chen's simple SIF-GPP relationship in estimating the GPP
seasonality in tropical and subtropical EBFs (R>0.49). Despite potential overestimation
(**Fig. S1 b**) or underestimation (**Fig. S1 h**) of the magnitudes, RTSIF-derived GPP
mostly captured the seasonality of the EC GPP at all the four sites ($d_{phase}$ < 0.26).

### 259 3.3 Calculating the net rate of $CO_2$ assimilation (An)

We calculated the net $CO_2$ assimilation (An) using the FvCB biochemical model
(Farquhar et al., 1980). In this model, the parameter An was calculated as the minimum
of Rubisco ($W_c$), RuBP regeneration ($W_j$) and TPU ($W_p$) to minus dark respiration ($R_{dark}$)
(Bernacchi et al., 2013). The formulas for calculating An, $W_c$, $W_j$, $W_p$, $R_{dark}$ and
corresponding intermediate variables were listed in **Tables S4**.
*Calculation of $W_c$.* $W_c$ is expressed as a function of internal $CO_2$ concentration ($c_i$),
Michaelis-Menton constant for carboxylase ($K_c$), Michaelis-Menton constant for
oxygenase ($K_o$), $CO_2$ compensation point ($\Gamma^*$) and maximum carboxylation rate ($V_{c,max}$)
(**Table S4-part1**) (Lin et al., 2015; Bernacchi et al., 2013; Ryu et al., 2011; Medlyn et
al., 2011; June et al., 2004; Farquhar et al., 1980). The $K_c$, $K_o$, $\Gamma^*$ and $V_{c,max}$ are
temperature-dependent variables. Thus, we used **Equation 2** to calculate their values at
$T_k$ by converting from those at 25°C. Then, we used the Medlyn's stomatal conductance
model (Medlyn et al., 2011) to estimate internal $CO_2$ concentration ($c_i$) (**Equation 3**),
which is expressed as a function of vapor pressure deficit (VPD) rather than relative
humidity (Lin et al., 2015). The method for calculating the $V_{c,max}$ of each LAI cohort
was introduced in section 3.4. The formulas for calculating corresponding intermediate
parameters were presented in **Table S4**.
$$Para = Para_{25} \times exp\left(\frac{(T_k - 298.15) \times \Delta H_{para}}{R \times T_k \times 298.15}\right) \qquad (2)$$
where *Para* denotes a correction factor arising from the temperature dependence of
$V_{c,max}$; $Para_{25}$ are values of the temperature-dependent parameters ($K_c$, $K_o$, $\Gamma^*$ and $V_{c,max}$)
at the temperature 25°C; $T_k$ denotes temperature in Kelvin; $\Delta H_{para}$ is the activation
energy for temperature dependence; $R$ is the universal gas constant.
$$c_i = c_a \times \left(1 - \frac{1}{1.6 \times \left(1 + \frac{g_1}{\sqrt{VPD}}\right)}\right) \qquad (3)$$
where $c_a$ is atmospheric $CO_2$ concentration, 380 ppm; VPD is calculated from air
temperature and dew point temperature of the global ERA-Interim reanalysis dataset
(Dee et al., 2011) using the method of Yuan et al. (2019). The calculation formula of
VPD was described in supplementary files. In this study, we used the value of 3.77 for
the stomatal slope ($g_1$) in the stomatal conductance model according to Lin et al. (2015).
**Calculation of $W_p$.** $W_p$ was calculated as the function of $V_{c,max}$, which were given
different values for different LAI cohorts based on multiple *in situ* observations (section

290 3.4).

**Calculation of $W_j$.** $W_j$ was calculated from $V_{c,max}$, $c_i$ and the rate of electrons
through the thylakoid membrane ($J$) (Bernacchi et al., 2013). The parameter $J$ was
calculated from maximum electron transport rate ($J_{max}$) and the rate of whole electron
transport provided by light ($J_e$) (Bernacchi et al., 2013). $J_{max}$ was expressed as a
temperature dependence function of maximum electron transport rate ($J_{max,25}$) at 25°C
and temperature ($T_{air}$) and $J_e$ was expressed as a function of total PAR absorbed by
canopy ($PAR_{total}$) that was the sum of active radiation in beam ($PAR_{b,0}$) and diffuse
($PAR_{d,0}$) light firstly (Weiss et al., 1985), which were calculated from downward short-
wave radiation (SW) (Ryu et al., 2018). The formula for $PAR_{total}$ was given in **Equation**
**4** and formulas for other intermediate parameters (i.e., $PAR_{b,0}$, $PAR_{d,0}$, $\rho_{cb}$, $\rho_{cd}$, $k'_b$, $k'_d$,
and *CI*) were listed in **Table S4**.
$$PAR_{total} = (1 - \rho_{cb}) \times PAR_{b,0} \times \left(1 - exp(-k'_b \times CI \times LAI_{total})\right) + (1 - \rho_{cd}) \times$$
$$PAR_{d,0} \times \left(1 - exp(-k'_d \times CI \times LAI_{total})\right) \qquad (4)$$
where $PAR_{total}$ is total PAR absorbed by canopy; $PAR_{b,0}$ is the active radiation; $PAR_{d,0}$
is diffuse radiation; $LAI_{total}$ is a total LAI. Here, we used a constant value of 6.0
according to De Weirdt et al. (2012).

**3.4 Classifying three LAI cohorts with different $V_{c,max}$**

In this study, we collected *in situ* samples of $V_{c,max25}$ data against different leaf age
across tropical and subtropical EBFs from previous publications. Mature leaves (leaf
age: 70-160 days) show highest $V_{c,max25}$ than those of new flushed leaves (leaf age: <60
days) and old leaves(leaf age: >200 days) as Menezes et al. (2021). Therefore, in this
study, we classified the canopy leaves into three cohorts: young (leaf age: <2 months),
mature (leaf age: 3-5 months) and old cohorts (leaf age: >6 months) as Wu et al. (2016).
The $V_{c,max25}$ for young, mature and old cohorts were set as 60, 40 and 20 $\mu mol\ m^{-2}\ s^{-1}$,
respectively, according to previous ground-based observations by Chen et al. (2020).

**3.5 Decomposing camera-based LAI into three leaf age cohorts**

We classified the canopy leaves into young, mature and old age cohorts based on
the green-color band from the top-of-canopy imageries observed by RGB camera. It is
because that the brightness of different leaf age leaves differs greatly in the values of
green-color band. Raster density slicing is a useful classification method for detecting
the attributes of various ground objects (Kartikeyan et al., 1998). Therefore, we set three
brightness thresholds to divide young (blue), mature (green), old (yellow) leaves and
background (gray) for the same canopy extent in each month (**Fig. S2**). This analysis
was conducted in ENVI5.3 software.

**3.6 Evaluating the $LAI_{young+mature}$ seasonality and its spatial patterns using satellite-based EVI products**

To compare the seasonality of $LAI_{young+mature}$ with those of EVI, we calculate mean
squared deviation (MSD) and their three components—$d_{bias}$, which denotes the
differences about absolute value, $d_{var}$, which denotes the differences of seasonal
fluctuations, and $d_{phase}$, which denotes the differences of peak phase to evaluate this
consistence, comprehensively (see section 3.8). Additionally, we compared the spatial
patterns of the wet- minus dry-season differences ($\Delta$) between observed and simulated
variables, following the work of Guan et al. (2015). To determine the wet and dry
seasons in each grid cell, we defined a month as dry one when its monthly average
precipitation was smaller than the potential evapotranspiration (PET) computed using
the method of Maes et al. (2019); other months were classified as wet ones. The wet-
minus dry-season $LAI_{young+mature}$ (denoted as $\Delta LAI_{young+mature}$) was calculated for each
grid cell as the wet-season average $LAI_{young+mature}$ value minus the dry-season average
value of $LAI_{young+mature}$.

**3.7 Evaluating the $LAI_{old}$ seasonality using ground-based litterfall data**

Litterfall is closely related to the seasonal dynamics of old leaves, i.e., $LAI_{old}$ (Chen
et al., 2020; Yang et al., 2021). Previous analyses indicated that, in general, a sharping
decrease in $LAI_{old}$ corresponded to a peak in litterfall (Pastorello et al., 2020; Midoko
Iponga et al., 2019; Ndakara, 2011; Barlow et al., 2007; Dantas and Phillipson, 1989).
Based on this causal relationship between litterfall and $LAI_{old}$, we compared the time
of seasonal litterfall peak with the time of abrupt drops in $LAI_{old}$, to indirectly evaluate
the simulated $LAI_{old}$ seasonality. To accurately detect the onset date of old leaves
shedding and the day of litterfall peak, we used a least-square regression analysis
method developed by Piao et al. (2006) to smoothen $LAI_{old}$ and litterfall seasonal curves.
The sixth-degree polynomial function (n=6) was applicable to the regression (**Equation**
**5**).
$LAI_{old} = a_0 + a_1x + a_2x^2 + a_3x^3 + a_4x^4 + a_5x^5 + a_6x^6$ \qquad (5)
where x is the day of a year.
The slope of seasonal LAI ($LAI_{old, ratio}$) was calculated in **Equation 6**. The date of
abrupt drops in $LAI_{old}$ was defined as the time with most negative values of $LAI_{old, ratio}$.
$$LAI_{old,ratio(t)} = \left(LAI_{old(t+1)} - LAI_{old(t)}\right)/\left(LAI_{old(t)}\right) \qquad (6)$$
where $LAI_{old, ratio}$ is the slope of seasonal $LAI_{old}$ curve. $LAI_{old(t+1)}$ and $LAI_{old(t)}$ are the
corresponding monthly LAI at time t+1 and t, respectively.

**3.8 Evaluation Metrics**
Two metrics were chosen to evaluate the seasonality of Lad-LAI against the that of
other proxies: the Kobayashi decomposition of the Mean Square Difference between
model and observation (Kobayashi and Salam, 2000) and the Pearson correlation
coefficient (Pearson, 1896) for gridded fields.
***Mean squared deviation (MSD).*** The mean squared deviation (MSD) was given
by Kobayashi and Salam (2000):
$$MSD = \frac{1}{n}\sum_{i=1}^{n}(x_i - y_i)^2 \qquad (7)$$
$$SB = (\bar{x} - \bar{y})^2 \qquad (8)$$
$$SD_s = \sqrt{\frac{1}{n}\sum_{i=1}^{n}(x_i - \bar{x})^2} \qquad (9)$$
$$SD_m = \sqrt{\frac{1}{n}\sum_{i=1}^{n}(y_i - \bar{y})^2} \qquad (10)$$
$$SDSD = (SD_s - SD_m)^2 \qquad (11)$$
$$LCS = 2SD_sSD_m(1 - r) \qquad (12)$$
where mean squared deviation is the square of RMSD; i.e., MSD = $RMSD^2$; and $x_i$ is
the simulated data at time t, and $y_i$ is the observed one at time t (month). The lower the
value of MSD, the closer the simulation is to the measurement. MSD can be
decomposed into the sum of three components: the squared bias ($d_{bias}$), $d_{bias}=SB$; the
squared difference between standard deviations (variance-related difference, $d_{var}$),
$d_{var}=SDSD$; and the lack of correlation weighted by the standard deviations (phase-
related difference, $d_{phase}$), $d_{phase}=LCS$; $r$ indicates the correlation coefficient between $x$
and $y$.
***Pearson correlation coefficient (R)***. The Pearson correlation coefficient is a
measure of linear correlation between two variables (Merkl and Waack, 2009). The
correlation coefficient between X and Y was as:
$\rho_{X,Y} = \dfrac{cov(X,Y)}{\sigma_X \sigma_Y} = \dfrac{E\big((X-\mu_X)(Y-\mu_Y)\big)}{\sigma_X \sigma_Y}$     (13)

**3.9 The quality control (QC) for the Lad-LAI product**

To warn potential uncertainties, we provided information of data quality control
(QC) along with the Lad-LAI product (**Fig. S5**). In the QC system (**Table S7**), data
quality was divided into four levels: level 1 represents the highest quality; level 2 and
level 3 represent good and acceptable quality, respectively; and level 4 warns to be used
cautiously. This QC product was generated according to residual sum of squares (RSS)
(Melgosa et al., 2008) obtained from the constrained least-squares method that was used
to estimate derive monthly Lad-LAI data.

## 4. Results

**4.1 Comparison of LAI cohort seasonality with sparse site observations**

The simulated leaf age-dependent LAI seasonality product was validated against
the camera-based measurements of $LAI_{young}$, $LAI_{mature}$, and $LAI_{old}$ at four sites in south
America, one site in Congo and three sites in China. Overall, the LAI seasonality of
mature and old classes from the new Lad-LAI products agrees well at these sites with
very fine-scale collections of monthly LAI of mature (R=0.77, MSD=0.69) and old
leaves (R=0.59, MSD=0.62). However, the seasonality of simulated LAI from young
leaves performs a little poor (R=0.36, MSD=0.45). It is also interesting to note that the
canopy leaf phenology of TEFs at these sites differ greatly. In south America, at K67,
K34 and Eucflux sites, both *in situ* and simulated $LAI_{young}$ and $LAI_{mature}$ decrease at
early dry season around February and convert to increase at early wet season around
June (**Fig. 3 a, b, d, e, j, k**). At the Barrocolorado site, $LAI_{young}$ increases from the late
dry to early wet season around Mar in response to the increasing incoming shortwave
radiation and in contrast, $LAI_{mature}$ starts to increase at wet season around June (**Fig. 3**
**g, h**). However, in subtropical Asia, $LAI_{young}$ and $LAI_{mature}$ increase during the wet

season and peak with largest rainfall at June or July at Din, Gutian and Banna sites (**Fig. 5 a, b, d, e, g, h**). In Congo, we only found one site (Congoflux) with six months observation period (from May to October). The seasonality of $LAI_{young}$ and $LAI_{mature}$ are similar as those in tropical Asia while having smaller variations in magnitude due to the moderate seasonality of sunlight the Equator region (**Fig. 4 a, b**). Overall, there is a reverse pattern for $LAI_{old}$ seasonality compared to $LAI_{mature}$ for all the eight sites.

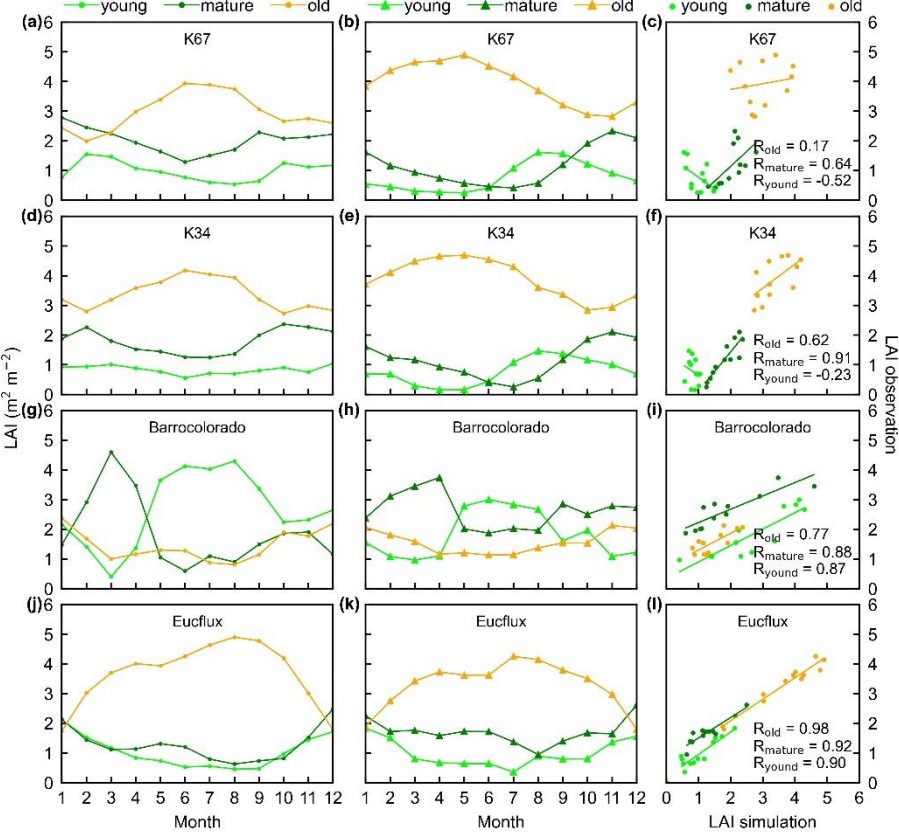

**Figure 3**. Seasonality of simulated $LAI_{young}$, $LAI_{mature}$, and $LAI_{old}$ in comparison with observed data at 4 sites in south America. (Panels a, d, g and j) simulated LAIs; (panels b, e, h and k) observed LAIs; (panels c, f, i and l) scatterplots between simulated and observed LAIs. Limegreen dots are $LAI_{young}$; green dots are $LAI_{mature}$; orange dots are $LAI_{old}$.

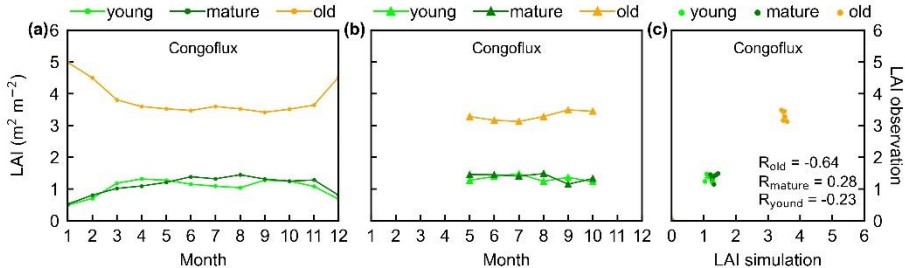

428

**Figure 4**. Seasonality of simulated LAI$_{young}$, LAI$_{mature}$, and LAI$_{old}$ in comparison with observed data at one site in Congo. (a) Simulated LAIs; (b) observed LAIs; and (c) scatterplots between simulated and observed LAIs. Limegreen dots are LAI$_{young}$; green dots are LAI$_{mature}$; orange dots are LAI$_{old}$.

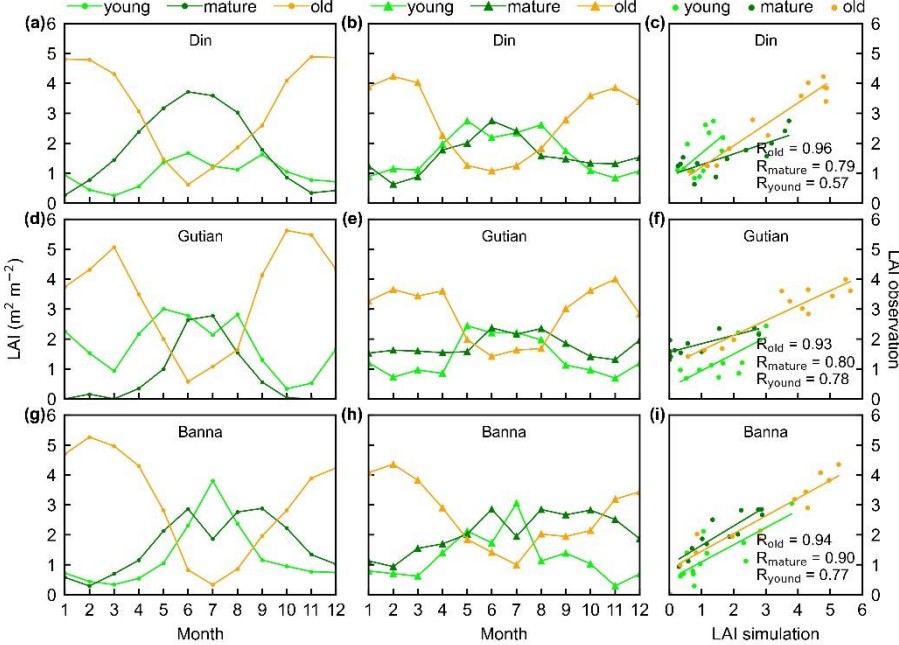

433

**Figure 5.** Seasonality of simulated LAI$_{young}$, LAI$_{mature}$, and LAI$_{old}$ in comparison with observed data at 3 sites in tropical Asia. (Panels a, d and g) simulated LAIs; (panels b, e and h) observed LAIs; (panels c, f and i) scatterplots between simulated and observed LAIs. Limegreen dots are LAI$_{young}$; green dots are LAI$_{mature}$; orange dots are LAI$_{old}$.

438

Additionally, only one ground site (Barrocolorado site in Panama) had time-series camera-based phenological imageries, which was then used to evaluate capacity of Lad-LAI in representing the interannual dynamics of three LAI cohorts, with R values being equal to 0.54, 0.64, 0.49 for LAI$_{young}$, LAI$_{mature}$, LAI$_{old}$, respectively (**Fig. 6**). However,

more *in situ* long-term observations are in need to test the robustness of the time-series
variations. The temporal variations of LAI$_{young}$, LAI$_{mature}$, LAI$_{old}$ across 8 sub-regions
classified by the *K*-means clustering analysis were shown in **Fig. S6**. Results showed
that, for example, the LAI$_{mature}$ increased significantly due to 2015 drought in Amazon
basin (e.g., sub-region S2, **Fig. S6**) and southeast Asia (e.g., sub-region S7, **Fig. S6**),
indicating good capability of detecting the dynamics of LAI$_{young}$, LAI$_{mature}$, LAI$_{old}$ in
response to climate disturbances.

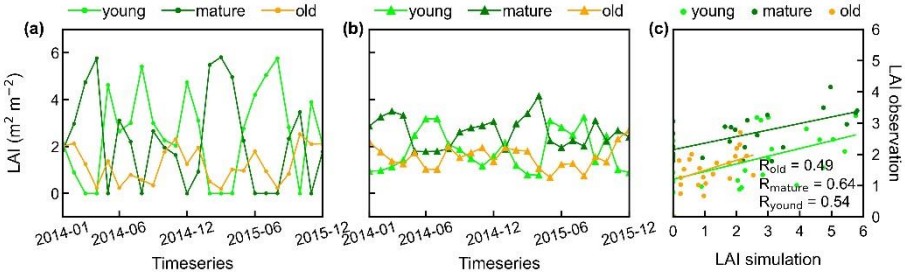


**Figure 6.** Timeseries of simulated LAI$_{young}$, LAI$_{mature}$, and LAI$_{old}$ in comparison with
observed data at Barrocolorado site in Panama. (a) Simulations LAIs; (b) observation
LAIs; and (c) scatterplots between simulated and observed LAIs.

**4.2 Comparison of patterns of grid LAI cohort seasonality with previous climatic**
**and phenological patterns**
The *in situ* measurements of LAI$_{young}$, LAI$_{mature}$, and LAI$_{old}$ suggested diverse
patterns of Lad-LAI seasonality over the TEFs. Nevertheless, the sparse coverage of
these sites raised challenging for a comprehensive and direct evaluation of leaf age-
dependent LAI seasonality product. To continue the grid Lad-LAI seasonality product
at the regional scale, we further conducted spatial clustering analyses of LAI$_{young}$,
LAI$_{mature}$, and LAI$_{old}$ using the *K*-means analysis method.
Surprisingly, the spatial patterns of Lad-LAI product clustered from satellite-based
vegetative signals (**Fig. 7 g-i**) coincide well with those clustered from in-dependent
climatic variables (rainfall and radiation etc.) (**Fig. 7 a-c**). These patterns are also
similar as those of the climate-phenology rhythms mapped by Yang et al. (2021), which
suggest different correlations of litterfall seasonality with canopy phenology between

different climate-phenology rhythms (**Fig. 7 d-f**). In central (sub-region S2) and south (sub-region S3) Amazon (**Fig. 7 g**), the seasonality of $LAI_{young}$, $LAI_{mature}$, and $LAI_{old}$ (**Fig. 8 b, c**) are similar as those of BR-Sa1 and BR-Sa3 sites. And in subtropical Asia (sub-region S6) (**Fig. 7 i**), the seasonality of three LAI cohorts (**Fig. 8 f**) are similar as those of Din, Gutian and Banna sites. Notably, the sub-region S8, located geographically between sub-regions S6 and S7, shows a $LAI_{young}$ peak at July and a bimodal phenology in $LAI_{mature}$ (**Fig. 8 h**). The remaining 4 sub-regions (sub-regions S1, S4, S5, S7) are all located nearby the equator. The magnitudes of seasonal changes in LAI cohorts are smaller than those in sub-regions S2, S3, S6 and S8 away from the Equator. It is worth noting that for these sub-regions around the Equator there is a bimodal seasonality pattern for $LAI_{mature}$, with the first peak around March and the second peak around August (**Fig. 8 a, d, e, g**). This is consistent with the findings of Li et al. (2021) which found that tropical and subtropical TEFs changed from a unimodal phenology at higher-latitudes to a bimodal phenology at lower-latitudes.

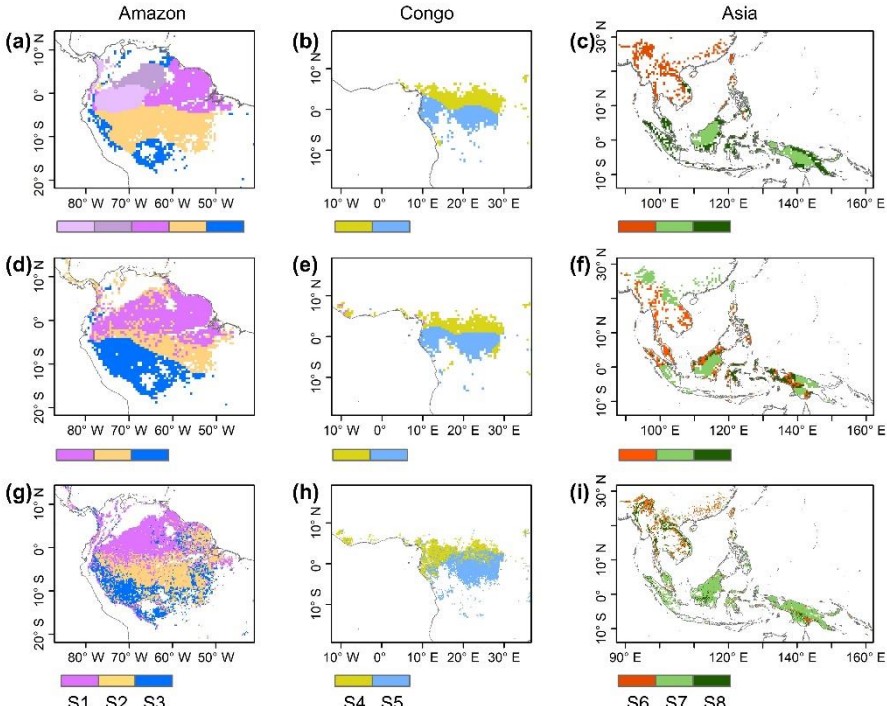

**Figure 7.** Comparison of sub-regions of Lad-LAI products (plots g-i) with those of climatic factors classified by the K-means clustering analysis (plots a-c) (Chen et al., 2021) and those of the three climate-phenology regimes (plots d-f) developed by Yang

486 et al. (2021).

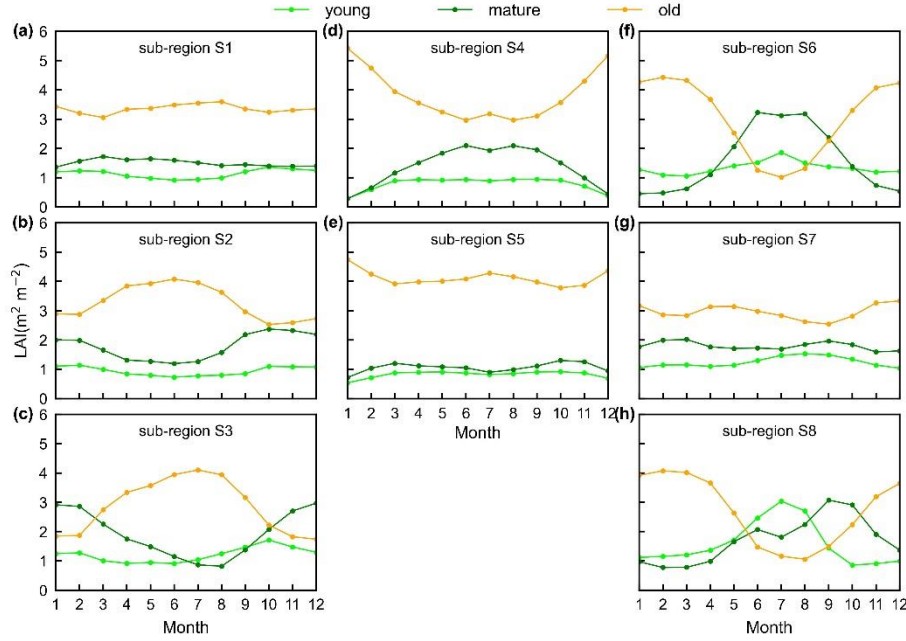

487

**Figure 8.** Seasonality of simulated LAI$_{young}$, LAI$_{mature}$, and LAI$_{old}$ in 8 sub-regions

classified by the K-means clustering analysis.

**4.3 Sub-regional evaluations of grid LAI$_{young+mature}$ seasonality using satellite-based**

**EVI products**

The grid dataset of LAI$_{young+mature}$ seasonality was indirectly evaluated using the

satellite-based EVI products (Wang et al., 2017; de Moura et al., 2017; Xiao et al., 2005;

Wu et al., 2018), as previous studies indicated that EVI can be considered as a proxy

for leaf area change of those leaves with high photosynthesis efficiency (Huete et al.,

2006; Lopes et al., 2016; Wu et al., 2018). It is because that EVI are very sensitive to

changes in near-infrared (NIR) reflectance (Galvão et al., 2011) while young and

mature leaves also reflect more NIR signals than the older leaves they replace (Toomey

et al., 2009). The linear correlation and MSD decompositions (see **methods**) between

simulated and satellite-based EVI were displayed in **Fig. 9**. Overall, the seasonal

LAI$_{young+mature}$ is well correlated with satellite-based EVI (R > 0.40) in 78.26% of the

TEFs and the average correlation coefficient is equaling to 0.61(**Fig. 9 a-c**). The MSD

is smaller than 0.1 in 89.69% of the whole tropical and subtropical TEFs (**Fig. 9 d-f**).

Statistics in the 8 clustered sub-regions show that the seasonal $LAI_{young+mature}$ of Lad-
LAI data mostly correlate better with seasonal EVI in high-latitude areas (sub-region
S2: R=0.65, sub-region S3: R=0.71, sub-region S6: R=0.67) than those in low latitudes
(sub-region S1: R=0.46, sub-region S5: R=0.61, sub-region S7: R=0.44, sub-region S8:
R=0.64) except for sub-region S4 (R=0.72) (**Figs. 10**, **S7**). The MSD components also
confirm the better performance of $LAI_{young+mature}$ seasonality in high-latitude areas (sub-
region S2:$d_{bias}$=0.009, $d_{var}$=0.001, $d_{phase}$=0.030; sub-region S3: $d_{bias}$=0.009, $d_{var}$=0.002,
$d_{phase}$=0.030; sub-region S6: $d_{bias}$=0.016, $d_{var}$=0.005, $d_{phase}$=0.040) than in low-latitude
areas near the Equator (sub-region S1: $d_{bias}$=0.012, $d_{var}$=0.001, $d_{phase}$=0.041; sub-region
S4: $d_{bias}$=0.020, $d_{var}$=0.001, $d_{phase}$=0.031; sub-region S5: $d_{bias}$=0.017, $d_{var}$=0.001,
$d_{phase}$=0.032; sub-region S7: $d_{bias}$=0.018, $d_{var}$=0.002, $d_{phase}$=0.043; sub-region S8:
$d_{bias}$=0.012, $d_{var}$=0.005, $d_{phase}$=0.035) (**Figs. 11**, **S7**). This happens because that the
accuracy of Lad-LAI in representing the seasonality of LAI cohorts depends highly on
that of input SIF data, which is low sensitive to canopy phenology and shows
marginally small seasonal changes nearby the Equator, for example in tropical Asia
(Guan et al., 2015; 2016).

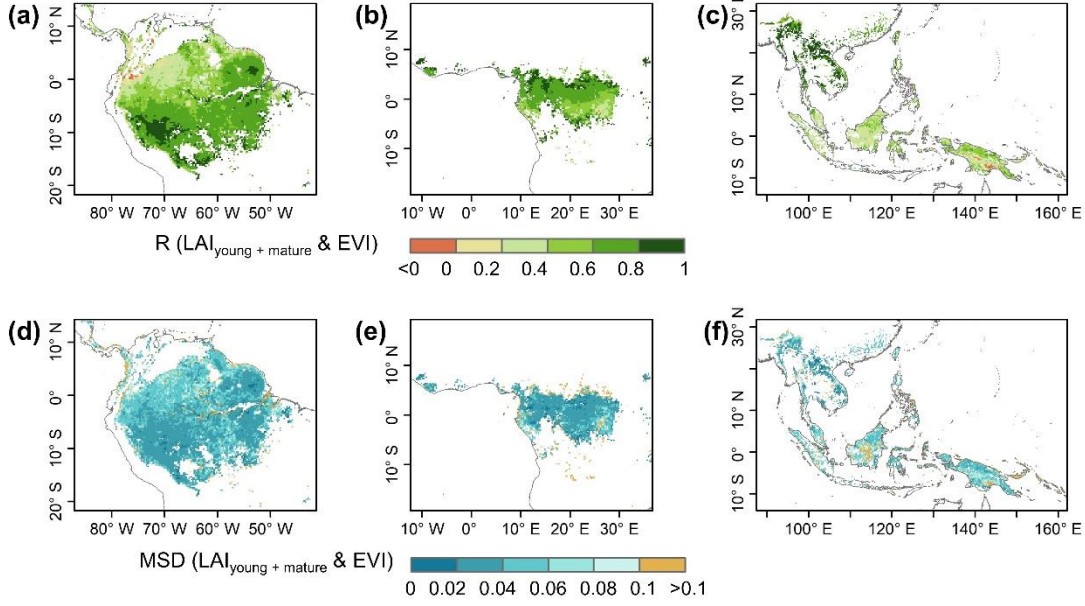


**Figure 9**. Pearson correlation coefficient (R) and mean squared deviation (MSD)
between seasonality of simulated $LAI_{young+mature}$ and MODIS Enhanced Vegetation
Index (EVI).

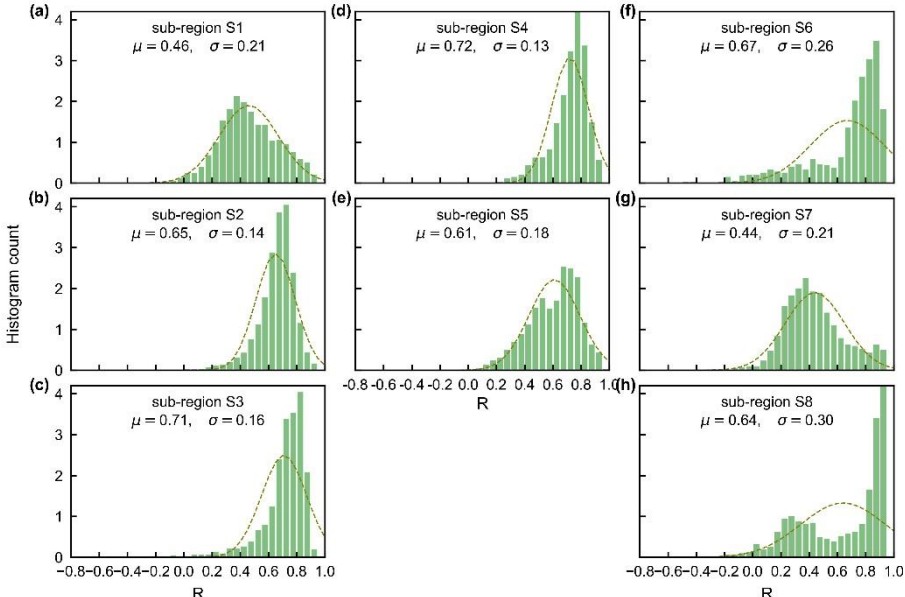


**Figure 10.** Statistics of the Pearson correlation coefficient (R) between seasonality of
simulated $LAI_{young+mature}$ and MODIS Enhanced Vegetation Index (EVI) in the 8
clustered sub-regions.

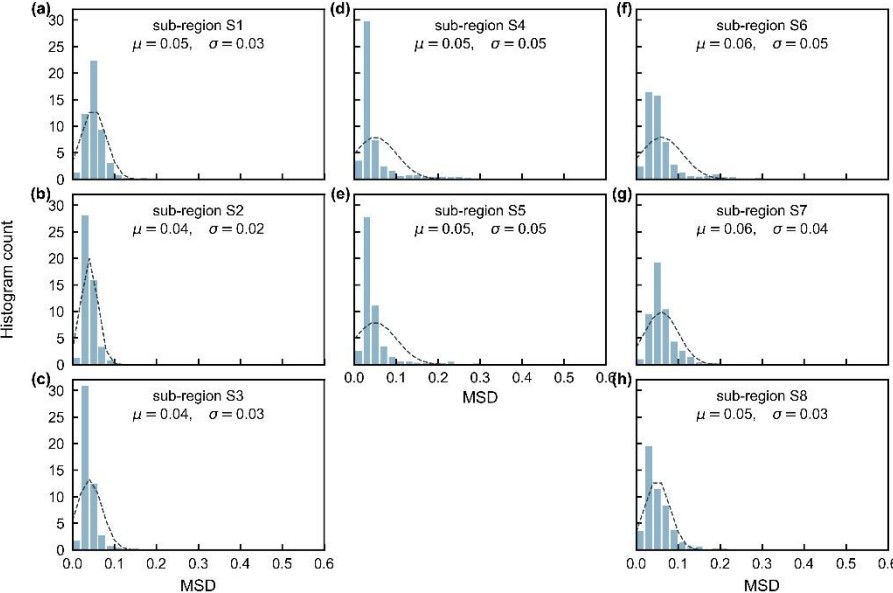


**Figure 11.** Statistics of the mean squared deviation (MSD) between seasonality of
simulated $LAI_{young+mature}$ and MODIS Enhanced Vegetation Index (EVI) in the 8
clustered sub-regions.

Additionally, previous studies indicated large-scale dry-season green-up area over
tropical and subtropical region (i.e., Guan et al., 2015, Tang et al., 2017, Myneni et al.,
2007) where the average annual precipitation exceeds 2,000 mm yr$^{-1}$. Here, we
calculated the differences (Δ) between wet- and dry-season LAI$_{young+mature}$ (i.e.,
LAI$_{young}$+ LAI$_{mature}$), to test whether the Lad-LAI can capture this green-up spatial
pattern. Spatial patterns of ΔLAI$_{young+mature}$ (**Fig. 12**) are similar to those developed by
(Guan et al., 2015), with higher LAI$_{young+mature}$ during the dry season (blue area) in large
areas north of the Equator. This indicates an emergence of new leaf flush and increase
of mature leaves, resulting the canopy "green-up" phenomenon observed by previous
satellite-based signals. It is interesting to note that the total areas (blue regions in **Fig.**
**12**) of this dry-season green up shown by LAI$_{young+mature}$ is smaller than those shown by
SIF signals that almost everywhere north of the Equator. That is because that new and
mature leaves often have quite a higher photosynthetic capacity than old leaves. A slight
or moderate "green-up" in new and mature leaves (i.e., increase in LAI$_{young+mature}$)
would boost strong increase in photosynthesis, inducing significant "green-up" shown
by photosynthesis-related signals, e.g., SIF data. Therefore, photosynthesis proxies
likely overestimate the areas with "green-up" of new leaves during the dry seasons in
the real world.

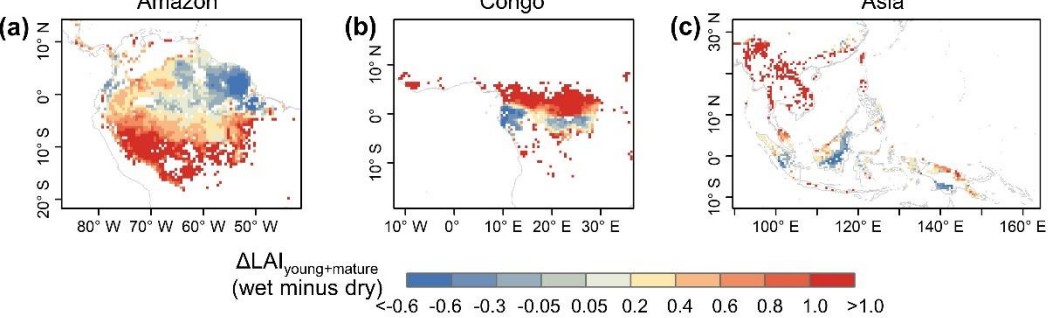


**Figure 12**. Spatial pattern of dry-season green-up using wet-season LAI$_{young+mature}$
minus dry-season LAI$_{young+mature}$.

**4.4 Sub-regional evaluations of grid LAI$_{old}$ seasonality using site-based litterfall**
**observations**
The seasonal patterns of LAI$_{old}$ were evaluated indirectly using ground-based
seasonal litterfall observations from 53 sites over the tropical and subtropical EBFs
(black circles in **Fig. 1**, **Figs. S8-S10**), Here, we selected 9 specific sites (**Fig. 13**) with
different patterns of litterfall seasonality and $LAI_{old}$ seasonality, to illustrate the
analyses results. **Fig. 13 a-i** illustrate the days when there is an abrupt decrease in
monthly $LAI_{old}$, which are closely to monthly litterfall peak. The days when $LAI_{old}$
decreases sharpest ($Day_{LAIold}$) agree well with the days when their monthly litterfall
peaks ($Day_{litterfall}$) (**Fig. 13 j**), mostly distributed near the diagonal lines (R=0.82). This
validation from seasonal litterfall data indirectly demonstrate the robustness of the
$LAI_{old}$ seasonality of the Lad-LAI product.

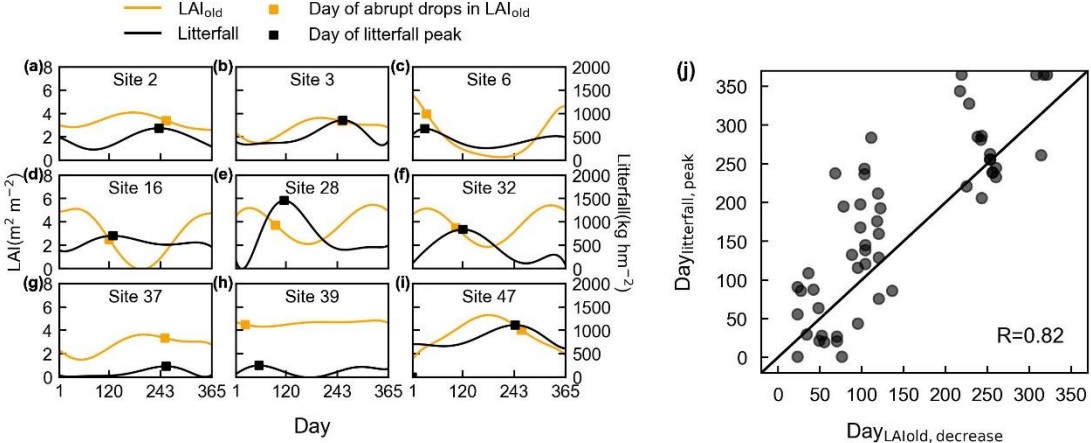


**Figure 13.** Evaluation of simulated $LAI_{old}$ using ground-observed litterfall seasonality.
(a-i) Days of an abrupt decrease in $LAI_{old}$ in comparison with days of corresponding
litterfall peak at 9 specific sites for examples. The orange curves represent simulated
$LAI_{old}$. Dots on the orange curves represent the point with an abrupt decrease in $LAI_{old}$.
The black curves represent observed seasonal litterfall mass. The dots on the black
curves represent the point with litterfall peak. (j) Comparisons of the days when $LAI_{old}$
has an abrupt decrease ($Day_{LAIold}$) against the days when monthly litterfall peaks
($Day_{litterfall}$).

**4.5 Testing potential uncertainties of the Lad-LAI products**
To prove the robustness of the neighbor-based decomposition approach, we
compared the Lad-LAI products generated based on 2*2 neighboring pixels with those
4*4 based on neighboring pixels. Results show that the seasonality of LAI_young,
LAI_mature and LAI_old in the 0.5-degree Lad-LAI products based on 4*4 neighboring
pixels are highly consistent with those of the 0.25-degree one based on 2*2 neighboring
pixels across the whole tropical region (**Fig. 14**), with the correlation coefficients (R)
being equaling to 0.63, 0.68 and 0.95, respectively (**Fig. S11**).

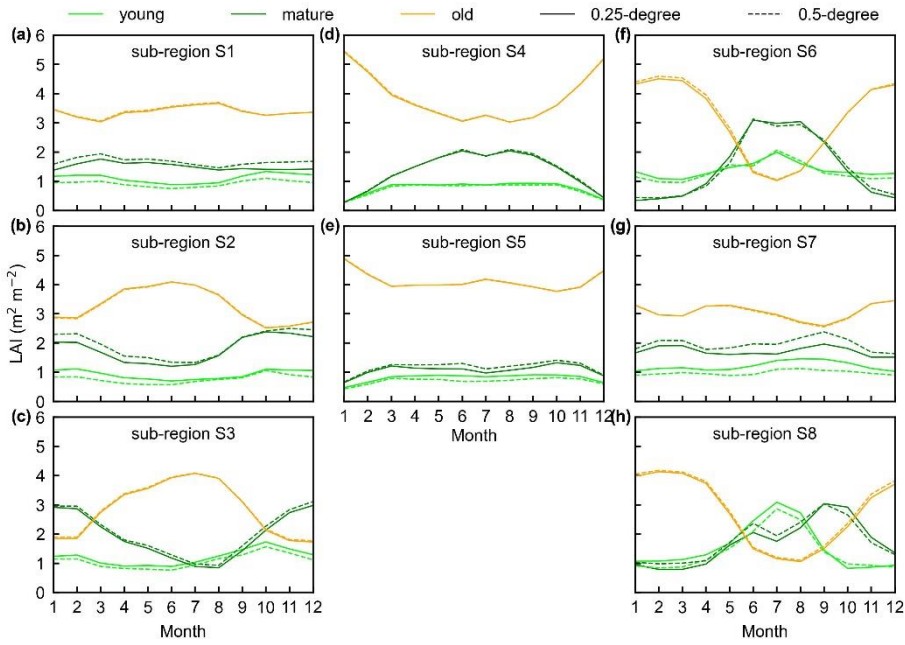


**Figure 14.** The seasonality of LAI_young, LAI_mature, LAI_old between 0.25-degree and 0.5-

degree Lad-LAI datasets in the 8 clustered regions. Limegreen color represents LAI_young;

green color represents LAI_mature; and orange color represents LAI_old. Solid lines

represent 0.25-degree dataset and the dashed lines represent 0.5-degree dataset.


To test the uncertainties caused by the GPP estimation, we added two more GPP
products, i.e., GOSIF-derived GPP (Li and Xiao, 2019) and FLUXCOM GPP (Jung et
al., 2019), to produce another two versions of Lad-LAI products. The GPP seasonality
coincide well between these three data sources across all the 8 sub-regions (**Fig. S12**).
By comparing with the ground-based LAI cohorts at eight observation sites, results
show that the Lad-LAI generated from RTSIF-derived GPP show highest correlation
and minimal deviation with the *in situ* measurements, with R equaling to 0.36, 0.77 and
0.59 and MSD equaling to 0.45, 0.69 and 0.62 for LAI_young, LAI_mature, and LAI_old,
respectively (**Figs. 15-16**, **S13-S15**). Additionally, we also compared the seasonal
variability of LAI$_{young}$, LAI$_{mature}$, and LAI$_{old}$ between three Lad-LAI versions in 8 sub-
regions classified by the K-means clustering analysis (**Fig. 17**). In general, three
versions of Lad-LAI products all performed well in 8 sub-regions with the consistent
seasonal variability (**Fig. 17**). On regional average, sub-regions S4, S5, S6, S7 and S8
show high consistent seasonality of LAI$_{young}$, LAI$_{mature}$, and LAI$_{old}$ between these three
products; whereas the Lad-LAI generated from GOSIF-derived GPP performs a little
poor in capturing the seasonality of LAI cohorts in Amazon (sub-regions S1, S2 and
S3).

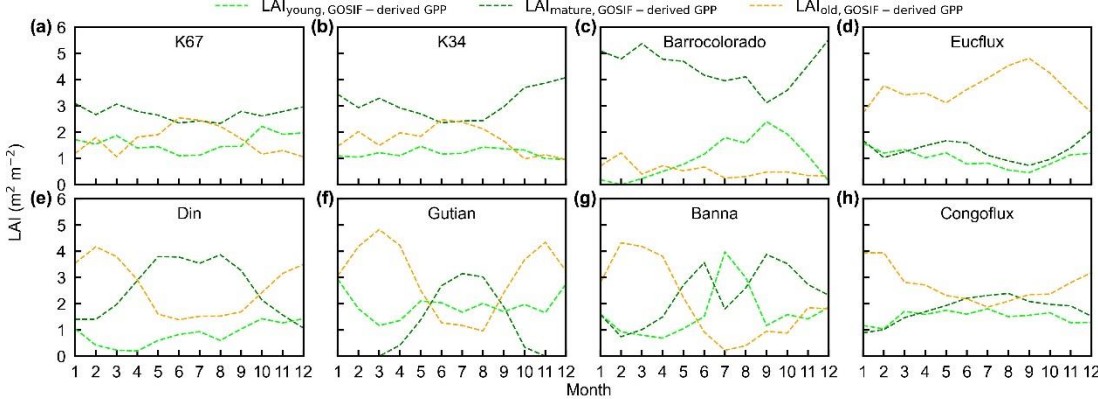


**Figure 15.** Seasonality of simulated LAI$_{young}$, LAI$_{mature}$, and LAI$_{old}$ from GOSIF-
derived GPP in comparison with observed data at 8 sites. (a) K67; (b) K34; (c)
Barrocolorado; (d) Eucflux; (e) Din; (f) Gutian; (g) Banna; (h) Congoflux.

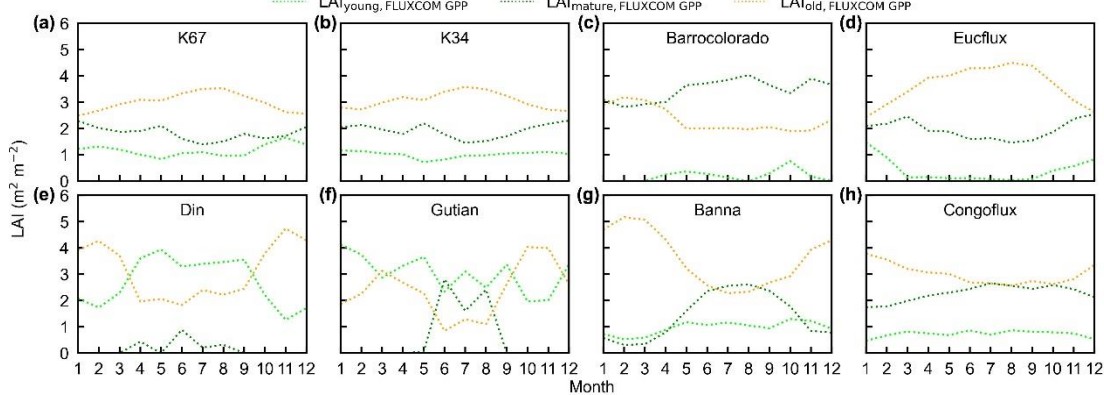


**Figure 16.** Seasonality of simulated LAI$_{young}$, LAI$_{mature}$, and LAI$_{old}$ from FLUXCOM
GPP in comparison with observed data at 8 sites. (a) K67; (b) K34; (c) Barrocolorado;
(d) Eucflux; (e) Din; (f) Gutian; (g) Banna; (h) Congoflux.

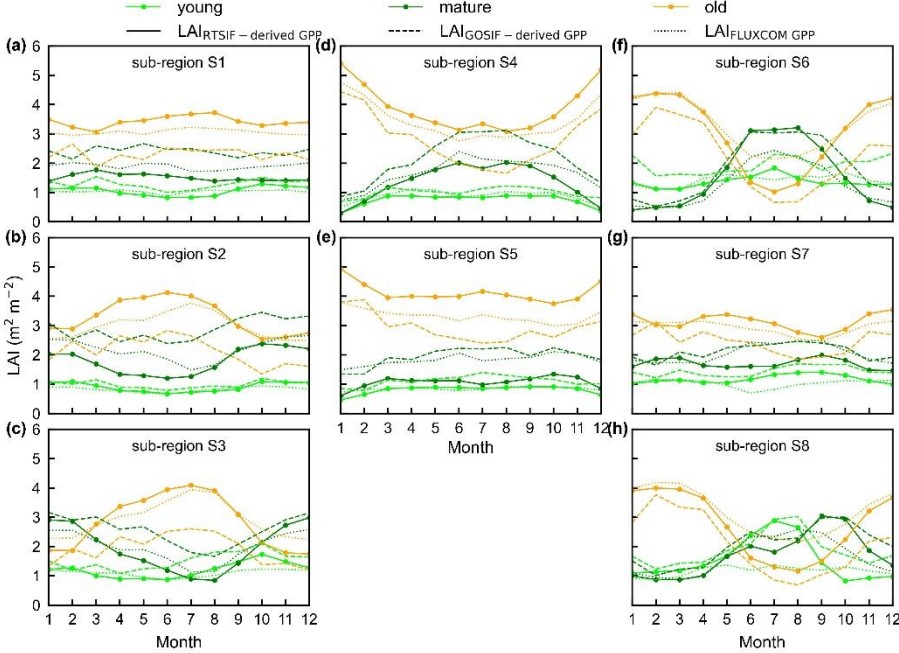


**Figure 17.** Seasonality of simulated $LAI_{young}$, $LAI_{mature}$, and $LAI_{old}$ from three version
products in 8 sub-regions classified by the K-means clustering analysis. Solid lines
represent LAI generated from RTSIF-derived GPP; dashed lines represent LAI
generated from GOSIF-derived GPP; and dotted lines represent LAI generated from
FLUXCOM GPP. Limegreen represents $LAI_{young}$; green represents $LAI_{mature}$; and
orange represents $LAI_{old}$.

## 5 Discussion

Leaf age-dependent LAI performs well in describing the seasonal replacements of
canopy leaves in TEFs (Wu et al., 2016; Chen et al., 2020), showing to be a critical
plant trait for representing the tropical and subtropical phenology (Doughty and
Goulden, 2008; Saleska et al., 2007). However, to our knowledge, there are currently
no continental-scale information of such leaf age-dependent LAI data over the whole
TEFs, as it can neither be mapped from sparse site observations (Wu et al., 2016), nor
be modeled from ESMs which are triggered by unclear climatic drivers (Chen et al.,
2020). These hinder global researches on accurately simulations of large-scale
photosynthesis (GPP) seasonality using remote sensing approaches and ESMs (Chen et
al., 2020).

The Lad-LAI product developed in this study is a new continental-scale grid dataset
of monthly LAI in different leaf age cohorts. Although lacking of enough *in situ*
observations for adequate validations, the seasonality of three LAI cohorts performs
well at the eight sites (four in south America, three in subtropical Asia and one in Congo)
with very fine-scale collections of monthly $LAI_{young}$, $LAI_{mature}$, and $LAI_{old}$. To test the
robustness of the grid Lad-LAI products over the whole TEFs, the seasonality of
$LAI_{mature}$ seasonality are also validated pixel by pixel using satellite-based EVI products
and the phase of $LAI_{old}$ seasonality are compared with the those of seasonal litterfall
data from 53 site measurements, respectively. Moreover, the $LAI_{young+mature}$ from the
new Lad-LAI products can also directly represent the large-scale dry-season green-up
of canopy leaves north of the Equator. Overall, direct and indirect evaluations both
demonstrated the robustness of the developed Lad-LAI products.
It should be noted that, over the regions with large magnitude of annual
precipitation nearby the Equator, there is no obvious dry seasons and thus tree canopy
phenology changes smaller than higher-latitude ones throughout the year (Yang et al.,
2021). The LAI of young, mature and old leaf cohorts all show a bimodal phenology
with marginally small seasonal changes nearby the Equator, which is captured by  the
developed Lad-LAI product. Secondly, we used a constant coefficient to transfer from
SIF data to GPP and also assume a constant value for the total LAI over the whole TEFs,
which might bring unexpected uncertainties. This can be seen from the MSD
evaluations, where the bias-related term dominates the total MSD, especially in regions
nearby the Equator. However, this bring less impacts on the seasonality of Lad-LAI, as
the phase-related term of MSD is much smaller.
Additionally, the maximum carboxylation rate ($V_{c,max}$) of leaves changes
significantly with leaf age (Xu et al., 2017). Currently, most Earth system models
(ESMs) define $V_{c,max}$ as a function of leaf age whereas their relationship is still less well
understood in TEFs due to sparse in-situ measurements (Chen et al., 2020). This
consequentially leads to the poor representation of LAI and GPP seasonality in ESMs
(De Weirdt et al., 2012). To overcome this challenge, here we simplified the tree canopy
into three big leaves (i.e., young, mature and old) in TEFs, similar as the two-big leaves
model developed for temperate and boreal forests (Best et al., 2011; Clark et al., 2011;
Harper et al., 2016), which simplified tree canopy into sun and shade leaves. However,
some uncertain remains on the assumption, as it neglects the spatial and temporal
variations of $V_{c,max}$, which also changes with seasonal climate anomaly and also differs
between nearby pixels in high heterogeneous forest ecosystems. This assumption may
bring uncertainties for simulating seasonal An and therefore influence the seasonality
of Lad-LAI.
In summary, we developed a new method to produce the first global grid dataset of
leaf age-dependent LAI product across the whole EBFs over the continental scale.
Although some uncertainties might remain, the Lad-LAI products could provide
seasonal age-dependent LAI data at the pixel-level to develop a common phenology
model for the whole tropical and subtropical EBFs in ESMs that are currently run at a
coarser resolution. Besides, with the development of remote sensing technology, finer
temporal and spatial resolutions of SIF products will enable finer temporal and spatial
resolutions maps of Lad-LAI products in the future.

## 6. Data availability

The 0.25-degree leaf age-dependent LAI seasonality (Lad-LAI) data from 2001-
2018 are presented in this paper as the main one, and their time-series are as a
supplementary dataset. The two datasets are available at
https://doi.org/10.6084/m9.figshare.21700955.v3 (Yang et al., 2022). Besides, we also
provided another two versions of Lad-LAI generated from GOSIF-derived GPP and
FLUXCOM GPP, respectively. These datasets are compressed in GeoTiff, with a spatial
reference of WGS84. Each file in those dataset is named like "LAI_{leaf age}_{spatial
resolution}_{month/year-month}.tif".

## 7. Conclusion

This study for the first-time mapped continental-scale grid dataset of monthly LAI in three leaf age cohorts from 2001-2018 RTSIF data. The LAI seasonality of young, mature and old leaves was evaluated using *in situ* measurements of seasonal LAI data, satellite based EVI and *in situ* measurements of seasonal litterfall data. The evaluations from these datasets demonstrate the robustness of the seasonality of three leaf age cohorts. The new Lad-LAI products indicate diverse patterns over the whole tropical and subtropical regions. In central and south Amazon, $LAI_{young}$ and $LAI_{mature}$ decrease at early dry season around February and convert to increase at early wet season around June. On the contrary, in subtropical Asia, $LAI_{young}$ and $LAI_{mature}$ increase during the wet season and peak with largest rainfall at June or July. In regions nearby the Equator, the LAI cohorts show a bimodal phenology but with marginally small changes in magnitudes. The proposed method will enable to produce finer temporal and spatial resolutions maps of Lad-LAI products by using precise temporal and spatial resolutions data as the inputs. The Lad-LAI products will be help for diagnosing the adaption of tropical and subtropical forest to climate change; and will also help improve the development of phenology models in ESMs.

**Supplement.** The supplement related to this article will be available online at once accepted.

**Author contributions.** XZ designed the research and wrote the paper. XY performed the analyses. All the authors edited and revised the paper.

**Competing interests.** The authors declare no competing interests.

**Financial support.**

This study was supported by the National Natural Science Foundation of China

(grant numbers U21A6001, 31971458, 41971275), the Guangdong Major Project of
Basic and Applied Basic Research (grant number 2020B0301030004), the Special high-
level plan project of Guangdong Province (grant number 2016TQ03Z354), Innovation
Group Project of Southern Marine Science and Engineering Guangdong Laboratory
(Zhuhai) (grant number 311021009).

**Acknowledgement**
Thanks for Dr. Jin Wu from Hongkong University for providing the observation
data of LAI cohorts at K67 and K34 sites in Amazon. We would also like to thank the
editor and reviewers for their valuable time in reviewing the manuscript.

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
