# Peer review of "A grid dataset of leaf age-dependent LAI seasonality product (Lad- LAI) over tropical and subtropical evergreen broadleaved forests"

_Earth System Science Data, 2022_

## Author Comment (AC2)

**Responses to reviewer 2#'s comments point by point**

MS No.: essd-2022-436

Title: A grid dataset of leaf age-dependent LAI seasonality product (Lad-LAI) over tropical and subtropical evergreen broadleaved forests

Author(s): Xueqin Yang et al.

**General Comments of Reviewer 2#:**

This work produced the first grid dataset of leaf age-dependent LAI product that is classified into young, mature, and old types, over the tropical evergreen broadleaved forests from satellite observations. It is an interesting work, and the overall framework is clear. The topic fits the ESSD, but there are still some major issues in this work that need to be addressed before this manuscript can be published. Some overall and point-to-point are provided below. I hope these comments are useful and constructive to improve this manuscript.

*Response: Thanks for the valuable comments and nice suggestions. We have carefully studied them and made corresponding revisions in the revised manuscript. The point-to-point responses are listed below.*

**Major Comments:**

**Comment 1:** The manuscript need to be thoroughly polished.

*Response: Thanks. We have thoroughly revised the manuscript following the reviewer's comments, e.g. totally rewrote the Introduction (see responses to Comment 4), added Study area and data (see responses to Comment 5), added new sites for validations (see responses to Comment 3), added new analyses of uncertainties (see responses to Reviewer 3#). Finally, we also asked a company to polish our English language, including grammar, syntax, and sentence structure, to improve the readability of the manuscript.*

**Comment 2:** Abstract cannot summarize this work well, particularly for describing results, accuracy, and performance (Lines 37-48). Alternatively, add some quantitative metrics in Abstract, e.g., how much accuracy can be reached for the site- and continental-scale validation and comparison (Lines 38-43), and how LAI cohort perform well with satellite data analysis (Lines 45-48), and also, using concise language to shorten Lines 49-52.

*Response: Thank your suggestions. To better summarize the results, accuracy, and performance, we have added some quantitative metrics to the abstract. Specifically, we found that our approach achieved accuracy of $R^2_{young}=0.41$, $R^2_{mature}=0.62$, $R^2_{old}=0.63$ for $LAI_{young}$, $LAI_{mature}$ and $LAI_{old}$ compared to in situ observation. On the regional average, the mean correlation coefficient between monthly EVI and $LAI_{young+mature}$ was up to 0.61. Furthermore, the Lad-LAI can capture the spatial pattern of dry-season "green-up" in satellite data analysis. Finally, we streamlined the language in original manuscript lines 49-52. The abstract revised as suggested as follows:*

*"Quantification of large-scale leaf age-dependent leaf area index has been lacking in tropical and subtropical evergreen broadleaved forests (TEFs) despite the recognized*

*importance of leaf age in influencing leaf photosynthetic capacity in this region. Here, we simplified the canopy leaves of TEFs into three age cohorts, i.e., young, mature and old one, with different photosynthesis capacity ($V_{c,max}$) and proposed a novel neighbor-based approach to develop a first monthly grid dataset with 0.25-degree spatial resolution of leaf age-dependent LAI product (**referred to as Lad-LAI**) during 2001-2018 over the continental scale from satellite observations of sun-induced chlorophyll fluorescence (SIF) that was reconstructed from MODIS and TROPOMI (the TROPOspheric Monitoring Instrument) as a proxy of leaf photosynthesis. The new Lad-LAI products showed good seasonality of three LAI cohorts, i.e., young ($LAI_{young}$) ($R^2=0.41$), mature ($LAI_{mature}$) ($R^2=0.62$) and old ($LAI_{old}$) ($R^2=0.63$) leaves, at the eight sites (four in south American, three in subtropical Asia and one in Congo) and also performed well in representing their interannual dynamics, with $R^2$ being equal to 0.30, 0.41 and 0.24 for $LAI_{young}$, $LAI_{mature}$ and $LAI_{old}$ at Barrocolorado site, respectively. Additionally, the days when $LAI_{old}$ decreases sharpest are mostly consistent with those of seasonal litterfall peaks at 53 in situ measurements across the whole tropical region (R=0.82). The LAI seasonality of young and mature leaves also agree well with the Enhanced Vegetation Index (EVI) products (R=0.61), which is a good proxy of effective leaves. The spatial patterns clustered from the three LAI cohorts coincide with those clustered from climatic variables and can also capture a dry-season green-up of canopy leaves across the wet Amazonia areas where mean annual precipitation exceeds 2,000 mm yr$^{-1}$, consistent with previous satellite data analysis. We added GOSIF-derived GPP and FLUXCOM GPP to test the potential uncertainties caused by GPP estimation based on SIF-GPP relationship. RTSIF-derived GPP based on simple SIF-GPP relationship showed the highest correlation with $LAI_{young}$, $LAI_{mature}$ and $LAI_{old}$ at 8 observed sites among the three versions. The new Lad-LAI also show stable seasonality in $LAI_{young}$, $LAI_{mature}$ and $LAI_{old}$ across the whole tropical region based on both 2\*2 and 4\*4 neighboring pixels, with R being equal to 0.63, 0.68 and 0.95, respectively. Here, we provide the average seasonality of three LAI cohorts as the main dataset, and their time-series as a supplementary dataset. The Lad-LAI products are available at https://doi.org/10.6084/m9.figshare.21700955.v3 (Yang et al., 2022)."*

**Comment 3:** I just concerned the results were validated by only three sites (one in subtropical Asia and two in Amazon). Can not find more sites to validate? For example, eddy covariance data and may find more details from papers (DOI: 10.1126/science.aad5068; https://doi.org/10.1016/j.agrformet.2013.04.031). More ground validation can show the robustness and accuracy of this dataset.

***Response***: *Thanks. The sites of the first literature provided by the reviewer are K67 and K34 sites in original manuscript Figure 5 that have been used for validations in this study. For the second literature provided by the reviewer, there is no observed LAI seasonality with different leaf age cohorts (young, mature and old) although it also applied a simple leaf-flush model to simulate leaves variability.*

*Following the reviewer's suggestion, in the new version, we added 5 more sites to validate the LAI datasets, e.g. Barrocolorado site in Panama, eucflux site in southern Amazon, congoflux site in Congo, Gutian and Banna sties in subtropical China.*

[Figure]

*Figure 1. Study areas over tropical and sub-tropical for evergreen broadleaves forests. Red triangles: four sites of EC-observed GPP seasonality. Blue pentangles: camera-based observation sites of three LAI cohort seasonality. Black circles: observation sites of litterfall seasonality.*

Till now, there are totally 8 sites for ground validations. Validation results were shown in Figures 3-5. All ground observations are consistent with the proposed Lad-LAI products. Please see details in the revised manuscript.

[revised manuscript text omitted]

**Comment 5:** It would be better to add a Study area and data used session to introduce some relevant information and Figure 1.

***Response:*** *We agree with the reviewer that a "2. Study area and material" section is needed, to introduce some relevant information and Figure 1. The text was added in the "Study area and material" as follows:*

> *"**2. Study area and material***
>
> *2.1 Tropical and subtropical evergreen broadleaved forest biomes***
>
> *In this study, we focused on pan tropical and subtropical evergreen broadleaf forests (TEFs). The pixels that belong to TEFs according to the International Geosphere-Biosphere Program (IGBP) classification were extracted as the study area based on the 0.05° spatial resolution MODIS land cover map (**Figure 1**) (MCD12C1, Sulla-Menashe et al., 2018). The study area contains three regions: South American (30°S–18°N; 40°W–90°W), the world's largest and most biodiverse tropical rain forest, Congo (10°S–10°N; 10°E−30°E), the western part of the Africa TEF region, and Tropical Asia (20°S–30°N; 70°E−150°E), covering the Indo-China Peninsula, the majority of the Malay Archipelago and the northern Australia.*

[Figure]

**Figure 1.** *Study areas over tropical and sub-tropical for evergreen broadleaves forests. Red triangles: four sites of EC-observed GPP seasonality. Blue pentangles: camera-based observation sites of three LAI cohort seasonality. Black circles: observation sites of litterfall seasonality.*

**2.2 Input datasets for calculating GPP and An parameters**

The TROPOMI (the TROPOspheric Monitoring Instrument) Solar-Induced Fluorescence (SIF) data are used to derive the continent-scale GPP (denoted as RTSIF-derived GPP) according to the SIF-GPP relationship established by Chen et al. (2022). The air temperature data from ERA5-land (Zhao, Gao et al., 2020), vapor pressure deficits (VPD) data from ERA-Interim (Yuan et al., 2019) and downward shortwave solar radiation (SW) from Breathing Earth System Simulator (BESS) (Ryu et al., 2018) were used to calculate $K_C$, $K_O$, $\Gamma^*$, $R_{dark}$ and $V_{c,max}$ and thus to calculate An according to equations in **Table S4-part1** and **Table S4-part4**. The calculation processes were illustrated in **Figure 2**. All datasets were aggregated at the same spatial (0.125°) and temporal resolutions (month).

**2.3 Datasets for validating leaf age-dependent LAI seasonality**

Ground-based seasonal LAI cohorts and litterfall data. Top-of-canopy imageries observed by phenology cameras were used to decompose the in situ seasonal $LAI_{young}$, $LAI_{mature}$, and $LAI_{old}$ data. In total, imageries from eight observation sites across the TEFs are used to validate the simulating results (blue pentangles in **Fig. 1**, **Table S1**). Additionally, the seasonal litterfall data from 53 in situ sites (black circles in **Fig. 1**) spanning the TEFs are collected from globally published articles to compare with the phase of $LAI_{old}$ seasonality (see **Methods** for details). The multiyear monthly litterfall data were averaged to the monthly mean to compare with the simulated seasonality of $LAI_{old}$. Four eddy covariance flux tower sites (red triangles in **Fig. 1**) provide in-situ GPP data to evaluate the seasonality of RTSIF-derived GPP.

Satellite-based seasonal MODIS EVI data. To evaluate the LAI seasonality of photosynthesis-effective leaves, i.e. young and mature leaves, this study used satellite-based MODIS Enhanced Vegetation Index (EVI) from independent sensors (Huete et al., 2002; Lopes et al., 2016; Wu et al., 2018) as a remotely sensed proxies alternatives of effective leaf area changes and new leaf flush, i.e., $LAI_{young+mature}$ (Wu et al., 2016; Xu et al., 2015). To prove the robustness of the products over a large spatial coverage, the seasonal LAI cohorts of young and mature leaves are evaluated against the enhanced vegetation index (EVI) product, which is considered as a proxy for leaf area changes of photosynthetic effective leaves (Xu et al., 2015; Wu et al., 2016)."

**Comment 6:** Authors used a constant value (LAI = 7) of total LAI in tropical and subtropical EBFs., but the valid range of LAI is generally 0 to 10. Thus, I expect to see more evidence for selecting 7 or a sensitivity analysis of threshold can also be implemented.

**Response**: *Thanks for your valuable comment regarding the selection of LAI constant in our manuscript. We have thoroughly collected relative studies to determine the appropriate LAI for tropical and subtropical EBFs. Results were shown in Figure R1, R2 and Table R1. Results showed that there are slightly spatial and seasonal variations in totally LAI (around 6.0) across the pantropical forests. Thus, we have revised the LAI constant value to 6 in the revised manuscript and updated Lad-LAI products accordingly.*

**Table R1.** *Information of total LAI mean values from the references.*

| NO. | LAI mean | Sites | Methods | Ref. |
|-----|----------|-------|---------|------|
| 1 | 6.0 | ORCHIDEE TrBE module | Module | de Weirdt et al., 2012 |
| 2 | 5.88 | K34 | observation | Wu et al., 2016 |
| 3 | 5.45 | Tapajo´s National Forest | observation | Asner et al., 2003 |
| 4 | 6.04 | Barro Colorado Island | observation | Wirth et al., 2001 |
| 5 | 6.0 | Costa Rican Forest | observation | Clark et al., 2008; |
| 6 | 5.89 | K67 | observation | Wu et al., 2016 |
| 7 | 5.9 | Tapajo´s National Forest | observation | Brando et al., 2008 |
| 8 | 5.7 | K67 | observation | Smith et al., 2019 |
| 9 | 5.34 | Congo | observation | de Wasseige et al., 2003 |
| 10 | 5.93 | Xishuangbanna | observation | Li et al., 2010 |
| 11 | 5.67 | Dinhushan | observation | Zhao, Chen et al., 2020 |

[Figure]

**Figure R1.** *The measured LAI sites distribution map.*

[Figure]

*Figure R2. The seasonality of observed total LAI values from other studies.*

**Comment 7:** The format of Equation (1) should be: GPP = $LAI_{young}$ × $An_{young}$ + $LAI_{mature}$ × $An_{mature}$ + $LAI_{old}$ × $An_{old}$.

**Response:** *Thanks for the correction. We have revised Equation (1) as GPP = $LAI_{young}$ × $An_{young}$ + $LAI_{mature}$ × $An_{mature}$ + $LAI_{old}$ × $An_{old}$. according to your suggestion.*

**Comment 8:** It is weird why all R values are 0.99 in Fig.3?

**Response:** *It is a typo. We have revised the R values of this figure and moved it to Supplementary Figures as Figure S1 in the revised manuscript.*

[Figure]

*Figure S1. Comparisons between monthly RTSIF-derived GPP (red) and EC-observed GPP (blue). (a-b) Au-Rob, (c-d) BR-Sa1, (e-f) BR-Sa3, and (g-h) GF-Guy.*

**Comment 9:** Fig.3 is not supposed to place at Method part, can move it into results or supplementary materials; and Fig.4 is not a contribution of this work, can move it into supplementary materials.

*Response: Thanks. We have moved Figure 3 and Figure 4 to the Supplementary Figures Figure S1 and Figure S2, respectively, as suggested by the reviewer.*

**Comment 10:** Lines 351-355, can provide some scatterplots between Lad-LAI products and sites observations, rather than providing quantified accuracy metrics only.

*Response: It is a nice suggestion. We have added scatterplots between Lad-LAI products and sites observations in Figures 3-5 right panel. The scatterplots are shown as follows.*

[Figure]

***Figure R3.** The scatterplots of simulated LAIs against observed LAIs at 8 camera-based observation sites across study area.*

---

## Author Comment (AC3)

**Responses to reviewer 3#'s comments point by point**

MS No.: essd-2022-436

Title: A grid dataset of leaf age-dependent LAI seasonality product (Lad-LAI) over tropical and subtropical evergreen broadleaved forests

Author(s): Xueqin Yang et al.

**General Comments of Reviewer 3#:**

This paper introduces a novel dataset of age-dependent LAI for tropical and subtropic evergreen forests. Such a dataset is highly valuable and much in need to understand the dynamics of tropical canopy structure under climate change and improve the robustness of Earth System Models in reconstructing past dynamics and projecting future scenarios. The study estimated three LAI age cohorts based on a neighbor-based decomposition model and SIF-derived GPP data. The seasonality of leaf demography and its spatial variations is evaluated against ground-based measurements, and satellite observations, and analyzed with regard to other independent studies from climate controls. Results suggested a robust representation of the spatial variability in seasonality, which will be useful for improving Earth System Models. Overall, I find the dataset to be valuable and significant. I especially appreciate the authors' efforts in collecting and synthesizing ground-based observations globally to evaluate the products. However, I have some concerns regarding the robustness of the neighbor-based decomposition approach, the absence of evaluation regarding interannual dynamics, and the uncertainties in GPP estimations. I hope the authors will consider these points and provide further clarification in their responses and/or revisions. Please find my major comments and minor for clarification below.

*Response: Thanks so much for the constructive comments and suggestions regarding our manuscript. We have revised the manuscript thoroughly regarding the robustness of the neighbor-based decomposition approach, the absence of evaluation regarding interannual dynamics, and the uncertainties in GPP estimations as commented by reviewer, to*

*(1) To prove the robustness of the neighbor-based decomposition approach, we compared the Lad-LAI products generated based 2\*2 neighbor pixels and 4\*4 neighbor pixels. The seasonality and magnitudes of LAI of young, mature and old leaf cohorts are consistent between the two datasets (Figure R4, R5) (See responses to Comment 1).*

*(2) To evaluate the interannual dynamics of Lad-LAI, we could only find one ground site (Barrocolorado site in Panama) with time-series camera-based phenological imageries. Results showed that Lad-LAI could detect the interannual dynamic but more in situ observations are in need to test the robustness (Figure S3) (See responses to Comment 2).*

*(3) To test the uncertainties caused by the GPP estimation, we added two GPP products, i.e., GOSIF-derived GPP and FLUXCOM GPP for comparisons. Results showed that the Lad-LAI generated from SIF-derived GPP show highest consistent with the in situ observed LAI seasonality of different leaf age cohorts (Figure R6-R10). (See responses to Comment 3).*

**Major Comments:**

**Comment 1:** The approach using spatially adjacent GPP information to solve the leaf age composition is interesting but needs more justification on its robustness. With four observations (from four neighboring pixels) to solve three unknowns (LAI cohorts), the system does not have much space or tolerance for observation uncertainties (that is GPP, please see a related comment below). I suggest providing goodness-of-fit metrics from the least squares to evaluate the model performance. However, this still may not be informative due to a limited number of observations and lack of variations between the neighboring cells. Ideally, one solution would be to include more observations (for example, by increasing the number neighboring pixels from 4 to 8) to improve the robustness and accuracy of the models, but that also means a decrease in the spatial resolution of the product.

*Response: Thanks for nice suggestion in testing the robustness of the neighbor-based decomposition approach. Following your comments, we have increased the number of adjacent pixels from 4 (2\*2) to 16 (4\*4) to produce another version of Lad-LAI products with spatial resolution of 0.5-degree. Then, we compared the monthly $LAI_{young}$, $LAI_{mature}$, $LAI_{old}$ between the two datasets in the 8 clustered regions. Results showed that the seasonality of $LAI_{young}$, $LAI_{mature}$, $LAI_{old}$ are highly consistent in the 8 clustered regions (Figure R4, R5), and the correlation coefficients of $LAI_{young}$, $LAI_{mature}$, $LAI_{old}$ between the two datasets are $R_{young}= 0.63$, $R_{mature}= 0.68$, $R_{old}= 0.95$, respectively, implying the robustness of neighbor-based decomposition approach in decomposing the monthly $LAI_{young}$, $LAI_{mature}$ and $LAI_{old}$ from the monthly GPP using Equation 1.*

[Figure]

***Figure R4.*** *The seasonality of $LAI_{young}$, $LAI_{mature}$, $LAI_{old}$ between 0.25-degree and 0.5-degree LAI cohort datasets in the 8 clustered regions. The limegreen color represents $LAI_{young}$; green color represents $LAI_{mature}$; and orange color represents $LAI_{old}$. The solid lines represent 0.25-degree dataset and the dashed lines represent 0.5-degree datasets.*

[Figure]

**Figure R5.** *The scatterplot of 0.25-degree LAI_young, LAI_mature, LAI_old against 0.5-degree LAI cohort datasets in the 8 clustered regions.*

**Comment 2:** While the age-dependent LAI product is produced at monthly time steps over 2001-2018, it has only been validated and evaluated in terms of its LAI seasonality (i.e. multi-year average climatology). The reliability and usefulness of this product in representing interannual variabilities of leaf demography are highly uncertain. Thus, I strongly encourage the authors to evaluate the interannual temporal dynamics, even if only limited, since ground observations are often insufficient. The reliability of this product in terms of representation seasonality vs. interannual variabilities should be explicitly stated in the abstract, and thoroughly discussed in the main text, to prevent misuse of the dataset. I also suggest providing LAI cohorts seasonality as the main product, and the temporal dynamics as a supplementary dataset with a clear note of usage provided along with the product.

*Response: We appreciate for the reviewer's comment. We totally agree that it is important to evaluate the interannual temporal dynamics of the age-dependent LAI product. As said by the reviewer that the time-series ground observations are very limited, we could only find one ground site (Barrocolorado site in Panama) with time-series camera-based phenological imageries, to evaluate the interannual dynamics of Lad-LAI. Results showed that Lad-LAI could detect the interannual dynamic. The $R^2$ of the timeseries of LAI_young, LAI_mature, LAI_old between the two datasets are 0.30, 0.41, 0.24,*

*respectively (Figure S3). However, more in situ observations are in need to test the robustness. We thoroughly discussed the timeseries variability of LAI cohort dataset. We presented the temporal variations of $LAI_{young}$, $LAI_{mature}$, $LAI_{old}$ across 8 sub-regions classified by the K-means clustering analysis (Figure S4). The results demonstrated a consistent pattern of interannual variation, implying potential good capability of detecting abnormal events (e.g. subregion s7).*

[Figure]

***Figure S3.** Timeseries of simulated $LAI_{young}$, $LAI_{mature}$, and $LAI_{old}$ in comparison with observed data at Barrocolorado site in Panama. (a) simulations LAIs; (b) observation LAIs; and (c) scatterplots between simulated and observed LAIs.*

[Figure]

***Figure S4.** Timeseries of simulated $LAI_{young}$, $LAI_{mature}$, and $LAI_{old}$ in 8 sub-regions classified by the K-means clustering analysis. Limegreen represents $LAI_{young}$; green represents $LAI_{mature}$; and orange represents $LAI_{old}$.*

*We also agree with the suggestion to provide the LAI cohorts seasonality as the main product and the temporal dynamics as a supplementary dataset. In addition, we provided information of data quality control (QC) for the Lad-LAI product to prevent data misuse. In the QC system (**Table S6**), data quality is divided into four levels: level 1 represents the highest quality; level 2 and level 3 represent good and acceptable quality, respectively; and level 4 warns to be used cautiously. This QC product is generated according to the goodness of fit (residual sum of squares, RSS) (Melgosa et al., 2008, 2011) obtained from the constrained least-squares method used to estimate derive monthly Lad-LAI data. Results showed that more than 92.62% of pixels are with QC at best and gool levels and only less than 5.62% are with QC at level 4.*

**Table S6** *Information of data quality control (QC) for the Lad-LAI product*

| QC class | QC value | residual sum of squares (RSS) |
|---|---|---|
| Best | 1 | 0-1 |
| Good | 2 | 1-4 |
| Acceptable | 3 | 4-9 |
| Cautious use | 4 | >9 |

[Figure]

**Figure S8.** *Spatial patterns of seasonal QC datasets.*

**Comment 3:** SIF-GPP relationships used to estimate GPP in this study were based on only four sites with ground observations, that may not fully represent the tropical areas over the globe. Therefore, GPP estimations from SIF are subject to high uncertainties with possibly large biases. Given that the analytical approach used to solve does not consider uncertainties, the impact of GPP estimation uncertainties on age-dependent LAI estimates should be carefully discussed.

*Response: Thank you for the valuable comment. To test the uncertainties caused by the GPP estimation, we added two more GPP products, i.e., GOSIF-derived GPP and FLUXCOM GPP, to produce two versions of Lad-LAI products, for comparisons.*

*Firstly, we need to clarify that the overall regression slope of 15.343 in the 8-day between GPP and RTSIF represent over the regional average (Chen et al., 2022), not from SIF-GPP relationships based on only four sites with ground observations. Chen et al. (2022) established the linear relationship between RTSIF and GPP using 76 sites*

GPP data from the FLUXNET 2015 Tier 1 dataset in both 8-day and annual timescale (Fig. 8 in Chen et al. (2022)), indicating that RTSIF is tightly related to GPP. According to Chen et al. (2022), RTSIF was in good agreement with FLUXNET GPP for almost all biomes at the 8-day timescale, indicating strong SIF-GPP correlations for different biomes.

Second, to test the uncertainties of different SIF-GPP relationship on our analyses, we used the GOSIF-derived GPP products to produce another version of Lad-LAI. The GOSIF-derived GPP are generated based on various SIF-GPP relationships for the period from 2000 to 2020. According to Li and Xiao (2019), at site-level, the universal and biome-specific SIF-GPP relationships are established based on SIF soundings from OCO-2 and GPP data from 64 EC sites. And at grid cell level, a SIF-GPP relationship is established based on 0.05° GOSIF data and tower GPP. All of these SIF-GPP relationships with different forms (universal and biome-specific, with and without intercept) at both site and grid cell levels performed well in estimating GPP globally. We also used an independent GPP product—FLUXCOM GPP products to produce a third version of Lad-LAI. The FLUXCOM GPP are estimated from machine learning to merge carbon flux measurements from FLUXNET eddy covariance towers with remote sensing and meteorological data. We compared the seasonality of three GPP datasets in 8 sub-regions classified by the K-means clustering analysis. Results showed that the GPP seasonality are mostly consistent in 8 sub-regions (Figure R6).

[Figure]

**Figure R6.** *Seasonality of RTSIF-derived GPP (yellow lines), GOSIF-derived GPP (pink lines) and FLUXCOM GPP (blue lines) datasets in 8 sub-regions classified by the K-means clustering analysis. (a-c) South American; (d-e) Congo; (f-h) tropical Asia.*

For the three versions of Lad-LAI products, eight camera-based observation sites are used for compare the accuracy of the corresponding simulated LAI cohorts (Figure R7, R8, R9). We also compared the seasonal variability between three versions products

in 8 sub-regions classified by the K-means clustering analysis (Figure R10). Results showed that the Lad-LAI generated from RTSIF-derived GPP show highest consistent with the in situ observed LAI seasonality of different leaf age cohorts (Figure R7, R8, R9). The highest accuracies of the seasonality of $LAI_{young}$, $LAI_{mature}$, $LAI_{old}$ between the observed sites and the three datasets are all come from the Lad-LAI generated from RTSIF-derived GPP, $R^2_{young\ vs\ RTSIF-derived\ GPP} = 0.41$, $R^2_{mature\ vs\ RTSIF-derived\ GPP} = 0.62$, $R^2_{old\ vs\ RTSIF-derived\ GPP} = 0.63$, respectively.

In general, three versions of Lad-LAI products all performed well in 8 sub-regions with the consistent seasonal variability (Figure R10). In subregion s3, s6 and s8 keep the high consistent seasonal variability among three products, particularly. But the Lad-LAI generated from GOSIF-derived GPP performs a little poor in Amazon (subregion s1, s2 and s3).

[Figure]

**Figure R7.** *Seasonality of simulated $LAI_{young}$, $LAI_{mature}$, and $LAI_{old}$ in comparison with observed data at 4 sites in South American. (panels a, e, i and m) simulated LAIs from RTSIF-derived GPP; (panels b, f, j and n) camera-based observed LAIs; (panels c, g, k and o) simulated LAIs from GOSIF-derived GPP; and (panels d, h, l and p) simulated LAIs from FLUXCOM GPP.*

[Figure]

**Figure R8.** *Seasonality of simulated $LAI_{young}$, $LAI_{mature}$, and $LAI_{old}$ in comparison with observed data at one site in Congo. (a) simulated LAIs from RTSIF-derived GPP; (b) camera-based observed LAIs; (c) simulated LAIs from GOSIF-derived GPP; and (d) simulated LAIs from FLUXCOM GPP.*

[Figure]

***Figure R9.*** *Seasonality of simulated LAI_young, LAI_mature, and LAI_old in comparison with observed data at 3 sites in tropical Asia. (panels a, e and i) simulated LAIs from RTSIF-derived GPP; (panels b, f and j) camera-based observed LAIs; (panels c, g and k) simulated LAIs from GOSIF-derived GPP; and (panels d, h and l) simulated LAIs from FLUXCOM GPP.*

[Figure]

***Figure R10.*** *Seasonality of simulated LAI_young, LAI_mature, and LAI_old from three version products in 8 sub-regions classified by the K-means clustering analysis. Limegreen represents LAI_young; green represents LAI_mature; and orange represents LAI_old.*

**Comment 4:** Please note that evaluation against EVI is not entirely independent, since the RT-SIF dataset was a reconstruction from MODIS NBAR surface reflectance data. The manuscript needs improvements in language and grammar. I suggest carefully revising it to improve clarity.

**Response:** *Thanks for pointing this out. To be cautious, we have removed the statements of using "independent" in the revised manuscript. For the capability of using EVI as a*

*proxy for validating the seasonality young and mature leaves, Huete et al. (2006) found that Amazon rainforests green-up in dry season due to sunlight derive the synchronous canopy leaf turnover the young and mature leaves. And de Moura et al. (2017) compared tower and MODIS data with leaf flush and LAI from young to old leaves, and found an EVI increase toward September that closely tracked the modeled LAI of young/mature leaves (3–5 months). The MODIS EVI products are very sensitive to changes in NIR reflectance (Galvão et al., 2011) and young and mature leaves also could reflect more near-infrared (NIR) light than the older leaves replaced (Toomey et al., 2009). We have added such explanation in the new version.*

*For the improvements in language and grammar of the manuscript, we totally rewrote the Introduction (see responses to Comment 8 and Review 2# Comment 4), largely revised the Study area and data sections (see responses to Review 2# Comment 5). Finally, we also asked a company to polish our English language, including grammar, syntax, and sentence structure, to improve the readability of the manuscript.*

**Specific Comments:**

Abstract:

**Comment 5:** Please specify the temporal span, temporal and spatial resolution of the LAI product.

***Response:*** *Thanks for your comment. We revised it as suggested. The abstract revised as follows: "Here, we simplified the canopy leaves of TEFs into three age cohorts, i.e., young, mature and old one, with different photosynthesis capacity ($V_{c,max}$) and proposed a novel neighbor-based approach to develop a first monthly grid dataset with 0.25-degree spatial resolution of leaf age-dependent LAI product (**referred to as Lad-LAI**) during 2001-2018 over the continental scale from satellite observations of sun-induced chlorophyll fluorescence (SIF) that was reconstructed from MODIS and TROPOMI (the TROPOspheric Monitoring Instrument) as a proxy of leaf photosynthesis."*

**Comment 6:** L36: It should be noted that this is a SIF dataset that was reconstructed from MODIS and TROPOMI to avoid confusion.

***Response:*** *Thanks for your reminder. We have corrected it. (See responses to Comment 5)*

**Comment 7:** L40-41: Since the RTSIF is reconstructed from MODIS surface reflectance data, the evaluation against EVI is not precisely "independent".

***Response:*** *Thanks again. We have removed the statements of using "independent" in the revised manuscript.*

Introduction:

**Comment 8:** L103: The last paragraph of the Introduction should be shortened with a brief summary of the method and findings.

***Response:*** *Done as suggested. We have shortened this paragraph with a brief summary*

*of the method and findings as follows.*

*"To fill the research gap, this study aims to produce a grid dataset of leaf age-dependent LAI seasonality product (Lad-LAI) at the continental scale over the TEF biomes from 2001 to 2018. For this purpose, we simplified that canopy GPP was composed of three parts that are produced from young, mature and old leaves, respectively; and based on this assumption, GPP was expressed as a function of the sum of the product of each LAI cohort (i.e., young, mature and old leaves, denoted as $LAI_{young}$, $LAI_{mature}$, and $LAI_{old}$, respectively) and corresponding net $CO_2$ assimilation rate (An, denoted as $An_{young}$, $An_{mature}$, and $An_{old}$ for young, mature and old leaves, respectively) (**Equation 1**). Then, we proposed a novel neighbor-based approach to derive the values of three LAI cohorts. It is hypothesized that forests in adjacent four cells in the grid map exhibit consistent magnitude and seasonality of GPP, $LAI_{young}$, $LAI_{mature}$, and $LAI_{old}$. By applying **Equation 1** to each of the four selected cells, we combined the four equations to derive the three LAI cohorts using a linear least-squares with constrained method. An is calculated using the Farquhar-von Caemmerer-Berry (FvCB) leaf photochemistry model (Farquhar et al., 1980); and GPP is linearly derived from an arguably better proxy—TROPOMI (the TROPOspheric Monitoring Instrument) Solar-Induced Fluorescence (SIF) calibrated by eddy covariance GPP data (**See Methods for details**). This grid dataset of three LAI cohorts provides new insights into tropical and subtropical phenology with more details of sub-canopy level of leaf seasonality in different leaf age cohorts and will be helpful for developing accurate tropical phenology model in ESMs."*

Method:

**Comment 9:** L132-133: How much are the spatial variations in the constant LAI value?

***Response:*** *We analyzed the measured LAI values mentioned in other studies and found there are slightly spatial and seasonal variations in totally LAI (Figure R1, R2). A constant total LAI value (around 6.0) can be used for most evergreen broadleaf forests.*

[Figure]

***Figure R1.*** *The measured LAI sites distribution map.*

[Figure]

***Figure R2.*** *The seasonality of observed total LAI values from other studies.*

**Comment 10:** L147-168: Using GPP-SIF relationships based on only four sites is suspect to extrapolation issues over the entire areas.

***Response:*** *We apologies for this mistake describe. We have corrected it. The revised as follows: "The GPP is derived from SIF (denoted as RTSIF-derived GPP) using a linear regression model (see sect. 2.2) based on the relationship between RTSIF and EC-observed GPP from 76 sites (Chen et al., 2022)."*

**Comment 11:** L155: VPD data sources are different between Table S3 and Figure 2. ERA5-Land is at 0.1 degree instead of 0.05 deg? Can you double check?

***Response:*** *Thank you for your attention to detail. VPD datasets was calculated from ERA Interim datasets.*

**Comment 12:** L175: Could you please provide the GPP-SIF relationship equation and overall goodness-of-fit?

***Response:*** *Yes. The overall regression slope of 15.343 in the 8-day between GPP and RTSIF actually represent the regional average, which was provided by Chen et al., 2022, not from SIF-GPP relationships based on only four sites with ground observations. Chen et al. (2022) explored the relationship between RTSIF and GPP using 76 sites GPP data from the FLUXNET 2015 Tier 1 dataset, and found that there is a linear relationship between RTSIF and GPP in both 8-day and annual timescale (Fig. 8 in Chen et al. (2022)), indicating that RTSIF is tightly related to GPP. And they also reported RTSIF was in good agreement with FLUXNET GPP for almost all biomes at the 8-day timescale, indicating strong SIF-GPP correlations for different biomes.*

**Comment 13:** L270-271: Note that the RTSIF product is reconstructed from MODIS using the short-term TROPOMI data as a training set. Therefore, the evaluation against EVI is not independent.

***Response:*** *Yes, we agree with your comment and appreciate your reminder. We will use more accurate descriptions in the revised manuscript.*

**Comment 14:** L273: Can you please elaborate on how EVI reflects young and mature leaves, not old ones?

***Response:*** *Previous studies which used independent satellite observations from lidar and optical sensors reported a consistent phenomenon — dry-season greening in*

*Amazon forests (Saleska et al., 2007; Huete et al., 2006; Myneni et al., 2007). And one of the potential biophysical mechanisms of this seasonal greening in Amazon forests is synchronous canopy leaf turnover (Huete et al., 2006; Brando et al., 2010; Doughty et al., 2008) and young leaves flushing. The young leaves could reflect more near-infrared (NIR) light than the older leaves replaced (Toomey et al., 2009). The MODIS EVI products are very sensitive to changes in NIR reflectance (Galvão et al., 2011). As results, when MODIS EVI products were corrected for these effects using the Multi-Angle Implementation of Atmospheric Correction Algorithm (MAIAC), an EVI increase toward September that closely tracked the modeled LAI of young/mature leaves (3–5 months) (de Moura et al., 2017).*

**Comment 15:** L274: Specify MSD

**Response:** *MSD is the abbreviation for Mean Squared Deviation. The analysis of MSD clearly identified the simulation vs. measurement contrasts with larger deviation than others; the correlation–regression approach tended to focus on the contrasts with lower correlation and regression line far from the equality line. It was shown results of the MSD-based analysis were easier to interpret than those of regression analysis. This is because the three MSD components are simply additive and all constituents of the MSD components are explicit. This approach will be useful to quantify the deviation of calculated values obtained with this model from measurements. We have added more details to specify MSD in revision.*

**Comment 16:** Figure S1: the figure is too blur to read.

**Response:** *We divided into 3 classes for all those sites by region, South American, Congo, tropical Asia. Thanks.*

[Figure]

***Figure S5.*** *Seasonality of LAI$_{young}$, LAI$_{mature}$, LAI$_{old}$, litterfall, EVI, RTSIF-derived GPP,*

$T_{air}$, VPD and SW at South American 22 sites.

[Figure]

**Figure S6.** *Seasonality of $LAI_{young}$, $LAI_{mature}$, $LAI_{old}$, litterfall, EVI, RTSIF-derived GPP, $T_{air}$, VPD and SW at Congo 7 sites.*

[Figure]

**Figure S7.** *Seasonality of $LAI_{young}$, $LAI_{mature}$, $LAI_{old}$, litterfall, EVI, RTSIF-derived GPP, $T_{air}$, VPD and SW at tropical Asia 24 sites.*

**Comment 17:** L326: Please specify which variable (x,y) is estimated or observed.
**Response:** *Thanks. In original manuscript L326, in general, simulated value for LAI is denoted as X, and measured value is denoted as Y. Specifically, in original manuscript L351-355, to quantify the sites accuracy, MSD was calculated by X as estimated and Y as observed. In original manuscript Figure 8, to calculate MSD ($LAI_{young+mature}$ & EVI), X is $LAI_{young+mature}$ and Y is EVI.*

Result:
**Comment 18:** Figure 5: It's not clear which is estimated versus observed data.
**Response:** *Thanks for your comments. We have clarified in Figure 5 caption that the left column represents the simulated values, the middle column represents the observed values, and the right column shows the scatterplot.*

[Figure]

***Figure 3***. *Seasonality of simulated LAI_young, LAI_mature, and LAI_old in comparison with observed data at 4 sites in South American. (panels a, d, g and j) simulated LAIs; (panels b, e, h and k) observed LAIs; (panels c, f, i and l) scatterplots between simulated and observed LAIs. Limegreen dots are LAI_young; green dots are LAI_mature; orange dots are LAI_old.*

[Figure]

***Figure 4***. *Seasonality of simulated LAI_young, LAI_mature, and LAI_old in comparison with observed data at one site in Congo. (a) simulated LAIs; (b) observed LAIs; and (c) scatterplots between simulated and observed LAIs. Limegreen dots are LAI_young; green dots are LAI_mature; orange dots are LAI_old.*

[Figure]

**Figure 5.** *Seasonality of simulated LAI_{young}, LAI_{mature}, and LAI_{old} in comparison with observed data at 3 sites in tropical Asia. (panels a, d and g) simulated LAIs; (panels b, e and h) observed LAIs; (panels c, f and i) scatterplots between simulated and observed LAIs. Limegreen dots are LAI_{young}; green dots are LAI_{mature}; orange dots are LAI_{old}.*

**Comment 19:** L355-357: This sentence is a bit unclear. Can you elaborate on the "trade-off"?

*Response: Yes. In tropical and subtropical evergreen broadleaved forests, trees adapt their leaf phenology to avoid unfavorable environments such as limited light and water, and maximize their growth rate (Kikuzawa 1995; Vico et al., 2015). And the "trade-off" between the phenology of mature and old leaves means that these forests exhibit complex leaf shedding and rejuvenation strategies in response to moisture and light availability, and these strategies depend on soil water, atmospheric vapor pressure deficit, and incoming solar radiation. Specifically, leaf shedding in the dry season may be an adaptive response to soil water deficits (Asner et al., 2010; Brando et al., 2010) or atmospheric aridity (Xu et al., 2017). Alternatively, leaf shedding in non-water-limited conditions may constitute an adaptive strategy to replace senescent leaves with efficient young leaves to maximize photosynthesis (Chen et al., 2020).*

**Comment 20:** L359-360: Should one of the "early wet season" be "dry season"?

*Response: Thank you for your comment. The "early wet season" contain a period of dry season rather than refer to the same period as the dry season. The "early wet season" refer to the transitional period between the end of the dry season and the beginning of the wet season.*

**Comment 21:** L397: Chen et al., 2019 is not found in the reference list.

*Response: Thank you for your attention to detail. "Chen et al., 2019" in caption of Figure 6 actually corresponds to this one in the reference list: Chen, X., Ciais, P.,*

*Maignan, F., Zhang, Y., Bastos, A., Liu, L., Bacour, C., Fan, L., Gentine, P., Goll, D., Green, J., Kim, H., Li, L., Liu, Y., Peng, S., Tang, H., Viovy, N., Wigneron, J. P., Wu, J., Yuan, W., and Zhang, H.: Vapor Pressure Deficit and Sunlight Explain Seasonality of Leaf Phenology and Photosynthesis Across Amazonian Evergreen Broadleaved Forest, Global Biogeochemical Cycles, 35, 10.1029/2020gb006893, 2021. We have corrected the mistake cite in the revised version and checked and confirmed all the references.*

**Comment 22:** L395: Is it possible to keep a consistent number of clusters between the three datasets? For example, can you set eight clusters in Lad-LAI, so the southeast Asia area has three clusters consistent with plots d-f. This will make it easier to compare the datasets.

***Response:*** *Thank you for your constructive suggestion. We have updated the southeast Asia area to have three clusters consistent with plots d-f in Lad-LAI dataset, in order to make it easier to compare the datasets. The corresponding statistic figures has been updated accordingly.*

[Figure]

***Figure 6.*** *Comparison of sub-regions of Lad-LAI products (plots g-i) with those of climatic factors classified by the K-means clustering analysis (plots a-c) (Chen et al., 2021) and those of the three climate-phenology regimes (plots d-f) developed by Yang et al. (2021).*

[Figure]

**Figure 7.** *Seasonality of simulated LAI$_{young}$, LAI$_{mature}$, and LAI$_{old}$ in 8 sub-regions classified by the K-means clustering analysis.*

[Figure]

**Figure 9.** *Statistics of the Pearson correlation coefficient between seasonality of simulated LAI$_{young+mature}$ and MODIS Enhanced Vegetation Index (EVI) in the 8 clustered sub-regions. (a-e and g-i): the histogram of correlation coefficients; (f): mean of correlation coefficients in each sub-region*

[Figure]

***Figure 10.*** *Statistics of the Mean squared deviation (MSD) between seasonality of simulated $LAI_{young+mature}$ and MODIS Enhanced Vegetation Index (EVI) in the 8 clustered sub-regions. (a-e and g-i): the histogram of MSD; (f): mean of MSD in each sub-region.*

**Comment 23:** L413: I wonder if you have any hypothesis for the low performance in southeast Asia in comparison with other regions? (Figure 8a-c)

***Response:*** *Yes. Compared to tropical evergreen forests of the Amazon and Africa, tropical Asian regions exhibit the lowest sensitivity of solar-induced chlorophyll fluorescence (SIF) (Guan et al., 2015; 2016).*

**Comment 24:** Figure 12: Please increase font size. It's not clear which line represents site data. Can you also illustrate the meaning of the dots?

***Response:*** *Thank you for your comment. In the revised version, we have increased the font size of Figure 12 and added a legend to clarify which lines represent the site data. The orange dots in plots a-i represent the sharpest decreases day of old leaves and the black dots in plots a-i represent the peak day of litterfall mass. The black dots in plot j (right panel) represent the days when $LAI_{old}$ decreases sharpest ($Day_{LAIold}$) against the days when monthly litterfall peaks ($Day_{litterfall}$).*

[Figure]

***Figure 12.*** *Evaluation of simulated LAI$_{old}$ using site-observed litterfall seasonality. (a-i) Days of a sharping decrease in LAI$_{old}$ in comparison with days of corresponding litterfall peak at 9 specific sites for examples. The orange lines represent old leaves from simulation and dots represent the sharpest decrease day of old leaves. The black lines represent observed litterfall mass and dots represent the peak day of litterfall mass. (j) Comparisons of the days when LAI$_{old}$ decreases sharpest (Day$_{LAIold}$) against the days when monthly litterfall peaks (Day$_{litterfall}$).*

***Reference:***

*Asner, G.P., and Alencar, A.: Drought impacts on the Amazon forest: the remote sensing perspective. New Phytol. 187, 569–578. 2010.*

*Brando, P. M. et al. Seasonal and interannual variability of climate and vegetation indices across the Amazon. Proc. Natl Acad. Sci. USA, 107, 14685–14690. 2010*

*Chen, X., Huang, Y., Nie, C., Zhang, S., Wang, G., Chen, S., and Chen, Z.: A long-term reconstructed TROPOMI solar-induced fluorescence dataset using machine learning algorithms, Sci Data, 9, 427, 10.1038/s41597-022-01520-1, 2022.*

*Chen, X., Maignan, F., Viovy, N., Bastos, A., Goll, D., Wu, J., Liu, L., Yue, C., Peng, S., Yuan, W., Conceição, A. C., O'Sullivan, M., and Ciais, P.: Novel Representation of Leaf Phenology Improves Simulation of Amazonian Evergreen Forest Photosynthesis in a Land Surface Model, Journal of Advances in Modeling Earth Systems, 12, 10.1029/2018ms001565, 2020.*

*Chen, X., Ciais, P., Maignan, F., Zhang, Y., Bastos, A., Liu, L., Bacour, C., Fan, L., Gentine, P., Goll, D., Green, J., Kim, H., Li, L., Liu, Y., Peng, S., Tang, H., Viovy, N., Wigneron, J. P., Wu, J., Yuan, W., and Zhang, H.: Vapor Pressure Deficit and Sunlight Explain Seasonality of Leaf Phenology and Photosynthesis Across Amazonian Evergreen Broadleaved Forest, Global Biogeochemical Cycles, 35, 10.1029/2020gb006893, 2021.*

*de Moura, Y. M., Galvão, L. S., Hilker, T., Wu, J., Saleska, S., do Amaral, C. H., Nelson, B. W., Lopes, A. P., Wiedeman, K. K., Prohaska, N., de Oliveira, R. C., Machado, C. B., Luiz E.O.C. and Aragão.: Spectral analysis of amazon canopy phenology during the dry season using a tower hyperspectral camera and modis observations,*

*ISPRS Journal of Photogrammetry and Remote Sensing, 131, 52-64, https://doi.org/10.1016/j.isprsjprs.2017.07.006. 2017*

*Doughty, C. E. and Goulden, M. L.: Seasonal patterns of tropical forest leaf area index and $CO_2$ exchange. J. Geophys. Res. 113, G00B06. 2008.*

*Farquhar, G. D., von Caemmerer, S., and Berry, J. A.: A biochemical model of photosynthetic CO2 assimilation in leaves of C 3 species, Planta, 149, 78-90, 10.1007/BF00386231, 1980.*

*Galvão, L. S. et al. On intra-annual EVI variability in the dry season of tropical forest: a case study with MODIS and hyperspectral data. Remote Sens. Environ. 115, 2350–2359, 2011.*

*Guan, K., Berry, J. A., Zhang, Y., Joiner, J., Guanter, L., Huete, A. R., ... and Gentine, P.: Improving the monitoring of crop productivity using spaceborne solar-induced fluorescence. Global change biology, 22(2), 716-726. 2016.*

*Guan, K., Pan, M., Li, H., Wolf, A., Wu, J., Medvigy, D., Caylor, K. K., Sheffield, J., Wood, E. F., Malhi, Y., Liang, M., Kimball, J. S., Saleska, Scott R., Berry, J., Joiner, J., and Lyapustin, A. I.: Photosynthetic seasonality of global tropical forests constrained by hydroclimate, Nature Geoscience, 8, 284-289, 10.1038/ngeo2382, 2015.*

*Huete, A. R., Didan, K., Shimabukuro, Y. E., Ratana, P., Saleska, S. R., Hutyra, L. R., Yang, W., Nemani, R. R., and Myneni, R.: Amazon rainforests green-up with sunlight in dry season, Geophysical Research Letters, 33, 10.1029/2005gl025583, 2006.*

*Kikuzawa, K.: Leaf phenology as an optimal strategy for carbon gain in plants. Can. J. Bot. 73, 158–163. 1995.*

*Li, X., and Xiao, J.: Mapping photosynthesis solely from solar-induced chlorophyll fluorescence: A global, fine-resolution dataset of gross primary production derived from OCO-2. Remote Sensing, 11(21), 2563; https://doi.org/10.3390/rs11212563. 2019.*

*Melgosa, M., Huertas, R., and Berns, R. S.: Performance of recent advanced color-difference formulas using the standardized residual sum of squares index. Journal of the Optical Society of America A, 25(7), 1828-1834. https://doi.org/10.1364/JOSAA.25.001828. 2008*

*Melgosa, M., García, P. A., Gómez-Robledo, L., Shamey, R., Hinks, D., Cui, G., and Luo, M. R.: Notes on the application of the standardized residual sum of squares index for the assessment of intra- and inter-observer variability in color-difference experiments. Journal of the Optical Society of America A, 28(5), 949-953. https://doi.org/10.1364/JOSAA.28.000949. 2011*

*Myneni, R. B. et al. Large seasonal swings in leaf area of Amazon rainforests. Proc. Natl Acad. Sci. USA, 104, 4820–4823. 2007.*

*Saleska, S. R., Didan, K., Huete, A. R. and da Rocha, H. R.: Amazon forests green-up during 2005 drought. Science, 308, 612–612. 2007.*

*Toomey, M., Roberts, D. A. and Nelson, B.: The influence of epiphylls on remote sensing of humid forests. Remote Sens. Environ, 113, 1787–1798. 2009.*

*Vico, G., Thompson, S. E., Manzoni, S., Molini, A., Albertson, J. D., Almeida-Cortez, J.*

S., Fay, P. A., Feng, X., Guswa, A. J., Liu, H., Wilson, T. G., and Porporato, A.: Climatic, ecophysiological, and phenological controls on plant ecohydrological strategies in seasonally dry ecosystems. Ecohydrol., 8, 658– 679. doi: 10.1002/eco.1533. 2015.

Xu, X., Medvigy, D., Joseph Wright, S., Kitajima, K., Wu, J., Albert, L. P., Martins, G. A., Saleska, S. R., and Pacala, S. W.: Variations of leaf longevity in tropical moist forests predicted by a trait-driven carbon optimality model, Ecol Lett, 20, 1097-1106, 10.1111/ele.12804, 2017.

---

## Author Response (AR1)

**Responses to reviewers' comments point by point**

MS No.: essd-2022-436
Title: A grid dataset of leaf age-dependent LAI seasonality product (Lad-LAI) over tropical and subtropical evergreen broadleaved forests
Author(s): Xueqin Yang et al.

**Summary:**
**General Comments of Reviewer 1#:**
Yang et al's work mapped the seasonal leaf area index (LAI) of three leaf age cohorts (i.e., young, mature, and old leaves) to interpret the phenological seasonality in tropical and subtropical forests. They accomplished this by calculating gross primary productivity (GPP) from TROPOMI solar-induced chlorophyl fluorescence (SIF) observations. They validated the results with ground-based observations of leaf dynamics, and with a satellite-based vegetation index (EVI). They obtained good agreement between simulated and observed LAI.
***Response:*** *We appreciate the time and efforts of the editor and referees in reviewing this manuscript and the valuable suggestions offered. Please see our response to your comments in the supplement below.*

**Overall evaluation:**
**Comment 1:**
The global relevance of the study is incontestable and is underscored by the novelty of such dataset. When published, it will be an important contribution for the understanding of tropical forests phenology. However, the manuscript needs substantial review of the English style as there are numerous language mistakes, which makes the comprehension of the text difficult.
***Response:*** *Thank you for the positive comments on the novelty of our proposed dataset. We agree with the reviewer that it is essential to ensure that our manuscript is written clearly and effectively in English. We have conducted a thorough review of our manuscript to address any language mistakes and improved the overall readability of the text.*

**Minor Comments:**
Besides extensive review of the English style that I am not listing here, a few minor points need to be observed:
***Response:*** *Thank again for your concern on English style which we have improved thoroughly and asked a company to help polish the English language. For the minor comments raised by the reviewer, the point-to-point responses listed below.*

**Comment 2:** Line 39: "very fine collections of monthly LAI". What does fine collections mean? Fine-scale?
***Response:*** *Yes, "fine collections of monthly LAI" means "fine-scale collections of*

*monthly LAI". This has been revised in the manuscript.*

**Comment 3:** Line 94-95: GPP is not the same thing as photosynthesis!
***Response:*** *Totally agree. To be cautious, we have revised the sentence as "For this purpose, we first simplified that canopy GPP was composed of three parts that were produced from young, mature and old leaves, respectively. GPP was then expressed as a function of the sum of the product of each LAI cohort (i.e., young, mature and old leaves, denoted as $LAI_{young}$, $LAI_{mature}$, and $LAI_{old}$, respectively) and corresponding net $CO_2$ assimilation rate (An, denoted as $An_{young}$, $An_{mature}$, and $An_{old}$ for young, mature and old leaves, respectively) (**Equation 1**).". (In revision lines 123-128)*

**Comment 4:** Figure 5: Improve figure caption by clarifying which panels (left or right) represent the simulated and observed LAIs;
***Response:*** *We apologize for the confusion. We have revised the figure caption of Figure 5 to clarify which panels represent the simulated and observed LAIs, as follows like this one: "**Figure 5.** Seasonality of simulated $LAI_{young}$, $LAI_{mature}$, and $LAI_{old}$ in comparison with observed data at 3 sites in tropical Asia. (Panels a, d and g) simulated LAIs; (panels b, e and h) observed LAIs; (panels c, f and i) scatterplots between simulated and observed LAIs. Limegreen dots are $LAI_{young}$; green dots are $LAI_{mature}$; orange dots are $LAI_{old}$."*

[Figure]

***Figure 5.*** *Seasonality of simulated $LAI_{young}$, $LAI_{mature}$, and $LAI_{old}$ in comparison with observed data at 3 sites in tropical Asia. (Panels a, d and g) simulated LAIs; (panels b, e and h) observed LAIs; (panels c, f and i) scatterplots between simulated and*

*observed LAIs. Limegreen dots are LAI$_{young}$; green dots are LAI$_{mature}$; orange dots are LAI$_{old}$.*

**Comment 5:** The authors provided the reference to Keller et al 2001 as a source for in situ samples of V$_{c,max}$, but I don't think the citation is accurate. The referred paper is about biomass estimates and allometric equations;

*Response: Thank you for your careful review of the manuscript. The correct reference for in situ samples of V$_{c,max}$ is: "Menezes, J., Garcia, S., Grandis, A., Nascimento, H., Domingues, T. F., Guedes, A. V., Aleixo, I., Camargo, P., Campos, J., Damasceno, A., Dias-Silva, R., Fleischer, K., Kruijt, B., Cordeiro, A. L., Martins, N. P., Meir, P., Norby, R. J., Pereira, I., Portela, B., Rammig, A., Ribeiro, A. G., Lapola, D. M., and Quesada, C. A.: Changes in leaf functional traits with leaf age: when do leaves decrease their photosynthetic capacity in Amazonian trees? Tree Physiology, 42(5), 922-938, https://doi.org/10.1093/treephys/tpab042, 2021". We have updated the reference accordingly in the revised version of the manuscript.*

**Comment 6:** A key reference that should be included in the manuscript: https://doi.org/10.1111/nph.15726

*Response: Thank you for the recommendation. We have reviewed the reference and found that the 4-year of measured data in the reference is very useful for us. We have therefore included it in the revised version of our manuscript and cited it appropriately, as follows "Similar mechanism was also observed by the ground-based LiDAR which showed an increasing trend in upper canopy leaf area index (LAI) during the dry season, whereas a decrease in lower canopy LAI (more old leaves) (Smith et al., 2019).". (In revision lines 86-88)*

*Reference:*

*Menezes, J., Garcia, S., Grandis, A., Nascimento, H., Domingues, T. F., Guedes, A. V., Aleixo, I., Camargo, P., Campos, J., Damasceno, A., Dias-Silva, R., Fleischer, K., Kruijt, B., Cordeiro, A. L., Martins, N. P., Meir, P., Norby, R. J., Pereira, I., Portela, B., Rammig, A., Ribeiro, A. G., Lapola, D. M., and Quesada, C. A.: Changes in leaf functional traits with leaf age: when do leaves decrease their photosynthetic capacity in Amazonian trees? Tree Physiology, 42(5), 922-938, https://doi.org/10.1093/treephys/tpab042, 2021.*

*Smith, M. N., Stark, S. C., Taylor, T. C., Ferreira, M. L., de Oliveira, E., Restrepo-Coupe, N., Chen, S., Woodcock, T., dos Santos, D. B., Alves, L. F., Figueira, M., de Camargo, P. B., de Oliveira, R. C., Aragão, L. E. O. C., Falk, D. A., McMahon, S. M., Huxman, T. E. and Saleska, S. R.: Seasonal and drought-related changes in leaf area profiles depend on height and light environment in an Amazon forest. New Phytol, 222: 1284-1297. https://doi.org/10.1111/nph.15726, 2019.*

**General Comments of Reviewer 2#:**
This work produced the first grid dataset of leaf age-dependent LAI product that is classified into young, mature, and old types, over the tropical evergreen broadleaved forests from satellite observations. It is an interesting work, and the overall framework is clear. The topic fits the ESSD, but there are still some major issues in this work that need to be addressed before this manuscript can be published. Some overall and point-to-point are provided below. I hope these comments are useful and constructive to improve this manuscript.

*Response: Thanks for the valuable comments and nice suggestions. We have carefully studied them and made corresponding revisions in the revised manuscript. The point-to-point responses are listed below.*

**Major Comments:**
**Comment 1:** The manuscript need to be thoroughly polished.

*Response: Thanks. We have thoroughly revised the manuscript following the reviewer's comments, e.g., totally rewrote the Introduction (see responses to **Comment 4**), added Study area and data (see responses to **Comment 5**), added new sites for validations (see responses to **Comment 3**), added new analyses of uncertainties (see responses to **Reviewer 3#**). Finally, we also asked a company to polish our English language, including grammar, syntax, and sentence structure, to improve the readability of the manuscript.*

**Comment 2:** Abstract cannot summarize this work well, particularly for describing results, accuracy, and performance (Lines 37-48). Alternatively, add some quantitative metrics in Abstract, e.g., how much accuracy can be reached for the site- and continental-scale validation and comparison (Lines 38-43), and how LAI cohort perform well with satellite data analysis (Lines 45-48), and also, using concise language to shorten Lines 49-52.

*Response: Thank your suggestions. To better summarize the results, accuracy, and performance, we have added some quantitative metrics to the abstract. Specifically, we found that our approach achieved accuracy of $R_{young}=0.36$, $R_{mature}=0.77$, $R_{old}=0.59$ for $LAI_{young}$, $LAI_{mature}$ and $LAI_{old}$ compared to in situ observation. On the regional average, the mean correlation coefficient between monthly EVI and $LAI_{young+mature}$ was up to 0.61. Furthermore, the Lad-LAI can capture the spatial pattern of dry-season "green-up" in satellite data analysis. Finally, we streamlined the language in original manuscript lines 49-52. The abstract revised as suggested as follows:*

*"Quantification of large-scale leaf age-dependent leaf area index has been lacking in tropical and subtropical evergreen broadleaved forests (TEFs) despite the recognized importance of leaf age in influencing leaf photosynthetic capacity in this biome. Here, we simplified the canopy leaves of TEFs into three age cohorts (i.e., young, mature and old one with different photosynthesis capacity ($V_{c,max}$)) and proposed a novel neighbor-based approach to develop a first grid dataset of monthly leaf age-dependent LAI product (**referred to as Lad-LAI**) at 0.25-degree spatial resolution over the continental scale during 2001-2018 from satellite observations of sun-induced chlorophyll*

*fluorescence (SIF) that was reconstructed from MODIS and TROPOMI (the TROPOspheric Monitoring Instrument). The new Lad-LAI products show good performance in capturing the seasonality of three LAI cohorts, i.e., young (LAI$_{young}$) (R=0.36), mature (LAI$_{mature}$) (R=0.77) and old (LAI$_{old}$) (R=0.59) leaves, at the eight sites (four in south America, three in subtropical Asia and one in Congo) and can also represent their interannual dynamics at the Barrocolorado site, with R being equal to 0.54, 0.64 and 0.49 for LAI$_{young}$, LAI$_{mature}$ and LAI$_{old}$, respectively. Additionally, the abrupt drops in LAI$_{old}$ are mostly consistent with the seasonal litterfall peaks at 53 in situ measurements across the whole tropical region (R=0.82). The LAI seasonality of young and mature leaves also agrees well with the seasonal dynamics of Enhanced Vegetation Index (EVI) (R=0.61), which is a good proxy of effective leaves. Spatially, the grid Lad-LAI captures a dry-season green-up of canopy leaves across the wet Amazonia areas where mean annual precipitation exceeds 2,000 mm yr$^{-1}$, consistent with previous satellite-based analyses. The spatial patterns clustered from the three LAI cohorts also coincide with those clustered from climatic variables over the whole TEF region. The seasonality of LAI$_{young}$, LAI$_{mature}$ and LAI$_{old}$ derived from the estimated GPP based on a simple linear SIF-GPP relationship show the highest correlation with the in situ measurements at 8 observed sites compared with those derived from Orbiting Carbon Observatory-2-based SIF (GOSIF) GPP and eddy covariance flux tower measurements (FLUXCOM) GPP. Additionally, the Lad-LAI products developed by the neighbor-based approach using 2\*2 and 4\*4 neighboring pixels show stable seasonality in LAI$_{young}$, LAI$_{mature}$ and LAI$_{old}$ across the whole tropical region, respectively. We provide the average seasonality of three LAI cohorts as the main dataset, and their time-series as a supplementary dataset. These two products are available at https://doi.org/10.6084/m9.figshare.21700955.v3 (Yang et al., 2022)."*

**Comment 3:** I just concerned the results were validated by only three sites (one in subtropical Asia and two in Amazon). Can not find more sites to validate? For example, eddy covariance data and may find more details from papers (DOI: 10.1126/science.aad5068; https://doi.org/10.1016/j.agrformet.2013.04.031). More ground validation can show the robustness and accuracy of this dataset.

***Response****: Thanks. The sites of the first literature provided by the reviewer are K67 and K34 sites in original manuscript Figure 5 that have been used for validations in this study. For the second literature provided by the reviewer, there is no observed LAI seasonality with different leaf age cohorts (young, mature and old) although it also applied a simple leaf-flush model to simulate leaves variability.*

*Following the reviewer's suggestion, in the new version, we added 5 more sites to validate the LAI datasets, e.g., Barrocolorado site in Panama, Eucflux site in southern Amazon, Congoflux site in Congo, Gutian and Banna sties in subtropical China.*

[Figure]

**Figure 1.** *Study areas over tropical and subtropical evergreen broadleaves forests (TEF). Red triangles: observed GPP seasonality at four eddy covariance (EC) tower sites. Blue pentangles: observed LAI cohorts at eight camera-based observation sites. Black circles: observed litterfall seasonality at 53 observation sites.*

Till now, there are totally 8 sites for ground validations. Validation results were shown in Figures 3-5 in revised manuscript. All ground observations are consistent with the proposed Lad-LAI products. We have added the results details in the revised manuscript as follows:

"In south America, at K67, K34 and Eucflux sites, both in situ and simulated $LAI_{young}$ and $LAI_{mature}$ decrease at early dry season around February and convert to increase at early wet season around June (**Fig. 3 a, b, d, e, j, k**). At the Barrocolorado site, $LAI_{young}$ increases from the late dry to early wet season around Mar in response to the increasing incoming shortwave radiation and in contrast, $LAI_{mature}$ starts to increase at wet season around June (**Fig. 3 g, h**). However, in subtropical Asia, $LAI_{young}$ and $LAI_{mature}$ increase during the wet season and peak with largest rainfall at June or July at Din, Gutian and Banna sites (**Fig. 5 a, b, d, e, g, h**). In Congo, we only found one site (Congoflux) with six months observation period (from May to October). The seasonality of $LAI_{young}$ and $LAI_{mature}$ are similar as those in tropical Asia while having smaller variations in magnitude due to the moderate seasonality of sunlight the Equator region (**Fig. 4 a, b**). Overall, there is a reverse pattern for $LAI_{old}$ seasonality compared to $LAI_{mature}$ for all the eight sites." (In revision lines 408-420)

[Figure]

***Figure 3***. *Seasonality of simulated LAI$_{young}$, LAI$_{mature}$, and LAI$_{old}$ in comparison with observed data at 4 sites in south America. (Panels a, d, g and j) simulated LAIs; (panels b, e, h and k) observed LAIs; (panels c, f, i and l) scatterplots between simulated and observed LAIs. Limegreen dots are LAI$_{young}$; green dots are LAI$_{mature}$; orange dots are LAI$_{old}$.*

[Figure]

***Figure 4***. *Seasonality of simulated LAI$_{young}$, LAI$_{mature}$, and LAI$_{old}$ in comparison with observed data at one site in Congo. (a) Simulated LAIs; (b) observed LAIs; and (c) scatterplots between simulated and observed LAIs. Limegreen dots are LAI$_{young}$; green dots are LAI$_{mature}$; orange dots are LAI$_{old}$.*

[Figure]

***Figure 5.*** *Seasonality of simulated LAI_young, LAI_mature, and LAI_old in comparison with observed data at 3 sites in tropical Asia. (Panels a, d and g) simulated LAIs; (panels b, e and h) observed LAIs; (panels c, f and i) scatterplots between simulated and observed LAIs. Limegreen dots are LAI_young; green dots are LAI_mature; orange dots are LAI_old.*

**Comment 4:** Introduction can be considered to re-organize, as the current version seems lack some logics and useful information.

***Response****: Thanks for the comments on the introduction section. We have reorganized it as follows:*

[revised manuscript text omitted]

**Comment 5:** It would be better to add a Study area and data used session to introduce some relevant information and Figure 1.

***Response:*** *We agree with the reviewer that a "2. Study area and material" section is needed, to introduce some relevant information and Figure 1. The text was added in the "Study area and material" as follows:*

**"2. Study area and material**

**2.1 Tropical and subtropical evergreen broadleaved forest biomes**

*In this study, we focused on the whole tropical and subtropical evergreen broadleaf forests (TEFs). The pixels labeled TEFs according to the International Geosphere-Biosphere Program (IGBP) classification were extracted as the study area based on the 0.05° spatial resolution MODIS land cover map (**Fig. 1**) (MCD12C1, Sulla-Menashe et al., 2018). The study area contains three regions: South America (30°S–18°N; 40°W–90°W), the world's largest and most biodiverse tropical rain forest, Congo (10°S–10°N; 10°W–30°E), the western part of the Africa TEF region, and Tropical Asia (20°S–30°N; 70°E–150°E), covering the Indo-China Peninsula, the majority of the Malay Archipelago and the northern Australia.*

[Figure]

***Figure 1.*** *Study areas over tropical and subtropical evergreen broadleaves forests (TEF). Red triangles: observed GPP seasonality at four eddy covariance (EC) tower sites. Blue pentangles: observed LAI cohorts at eight camera-based observation sites. Black circles: observed litterfall seasonality at 53 observation sites.*

**2.2 Input datasets for calculating GPP and An parameters**

The TROPOspheric Monitoring Instrument (TROPOMI) Solar-Induced Fluorescence (SIF) data were used to derive the continent-scale GPP (denoted as RTSIF-derived GPP) according to the SIF-GPP relationship established by Chen et al. (2022) which used 15.343 as a transformation coefficient to covert SIF to GPP. The air temperature data from ERA5-Land (Zhao, Gao et al., 2020), vapor pressure deficits (VPD) data from ERA-Interim (Yuan et al., 2019) and downward shortwave solar radiation (SW) from Breathing Earth System Simulator (BESS) (Ryu et al., 2018) were used to calculate $K_C$, $K_O$, $\Gamma^*$, $R_{dark}$ and $V_{c,max}$ and thus to calculate An according to equations in **Table S4** . The calculation processes were illustrated in **Fig. 2**. All datasets were aggregated at the same spatial (0.125°) and temporal resolutions (month) (**Table S3**).

**2.3 Datasets for validating leaf age-dependent LAI seasonality**

**Ground-based seasonal LAI cohorts and litterfall data**. Top-of-canopy imageries observed by ground-based phenology cameras were used to decompose canopy LAI into $LAI_{young}$, $LAI_{mature}$ and $LAI_{old}$. In total, imageries from eight observation sites across the whole TEF region were used to validate the simulating results (blue pentagles in **Fig. 1**, **Table S1**). Additionally, the seasonal litterfall data from 53 in situ sites (black circles in **Fig. 1**, **Table S6**) spanning the TEFs were collected from globally published articles to compare with the phase of simulated $LAI_{old}$ seasonality (see **Methods** for details). The multiyear monthly litterfall data were averaged to the monthly mean to compare with the seasonality of simulated $LAI_{old}$. Four eddy covariance flux tower sites (red triangles in **Fig. 1**, **Table S2**) provided in situ seasonal GPP data to evaluate the seasonality of RTSIF-derived GPP.

**Satellite-based seasonal EVI data**. To evaluate the LAI seasonality of photosynthesis-effective leaves, i.e., young and mature leaves, this study used satellite-based MODIS Enhanced Vegetation Index (EVI) (Huete et al., 2002; Lopes et al., 2016; Wu et al., 2018) as a remotely sensed proxies alternatives of effective leaf area changes and new leaf flush, i.e., $LAI_{young+mature}$ (Wu et al., 2016; Xu et al., 2015). To prove the robustness of the products over a large spatial coverage, the seasonal LAI cohorts of young and mature leaves were evaluated against the enhanced vegetation index (EVI) product, which was considered as a proxy for leaf area changes of photosynthetic effective leaves (Xu et al., 2015; Wu et al., 2016; de Moura et al., 2017)."

**Comment 6:** Authors used a constant value (LAI = 7) of total LAI in tropical and subtropical EBFs., but the valid range of LAI is generally 0 to 10. Thus, I expect to see more evidence for selecting 7 or a sensitivity analysis of threshold can also be implemented.

*Response: Thanks for your valuable comment regarding the selection of LAI constant in our manuscript. We have thoroughly collected relative studies to determine the appropriate LAI for tropical and subtropical EBFs. Results were shown in **Figure S3**, **S4** and **Table S5**. Results showed that there are slightly spatial and seasonal variations*

*in totally LAI (around 6.0) across the pantropical forests. Thus, we have revised the LAI constant value to 6.0 in the revised manuscript and updated Lad-LAI products accordingly.*

**Table S5.** *Information of total LAI mean values from previously published literatures.*

| NO. | LAI mean | Sites | Methods | Ref. |
|---|---|---|---|---|
| 1 | 6.0 | ORCHIDEE TrBE module | Module | De Weirdt et al., 2012 |
| 2 | 5.88 | K34 | observation | Wu et al., 2016 |
| 3 | 5.45 | Tapajo´s National Forest | observation | Asner et al., 2003 |
| 4 | 6.04 | Barro Colorado Island | observation | Wirth et al., 2001 |
| 5 | 6.0 | Costa Rican Forest | observation | Clark et al., 2008; |
| 6 | 5.89 | K67 | observation | Wu et al., 2016 |
| 7 | 5.9 | Tapajo´s National Forest | observation | Brando et al., 2008 |
| 8 | 5.7 | K67 | observation | Smith et al., 2019 |
| 9 | 5.34 | Congo | observation | de Wasseige et al., 2003 |
| 10 | 5.93 | Xishuangbanna | observation | Li et al., 2010 |
| 11 | 5.67 | Dinghushan | observation | Zhao, Chen et al., 2020 |

[Figure]

**Figure S3.** *The distribution map of measured LAI sites from previously published literatures.*

[Figure]

**Figure S4.** *The seasonality of observed total LAI values from previously published literatures.*

**Comment 7:** The format of Equation (1) should be: GPP = *LAIyoung* $\times$ *Anyoung* + *LAImature* $\times$ *Anmature* + *LAIold* $\times$ *Anold*.

**Response:** *Thanks for the correction. We have revised Equation (1) as GPP = LAI$_{young}$ $\times$ An$_{young}$ + LAI$_{mature}$ $\times$ An$_{mature}$ + LAI$_{old}$ $\times$ An$_{old}$ according to your suggestion.*

**Comment 8:** It is weird why all R values are 0.99 in Fig.3?

**Response:** *It is a typo. We have revised the R values of this figure and moved it to Supplementary Figures as **Figure S1** in the revised manuscript.*

[Figure]

***Figure S1.*** *Comparisons between monthly RTSIF-derived GPP (red) and observed GPP at eddy covariance (EC) tower sites (blue). (a-b) Au-Rob, (c-d) BR-Sa1, (e-f) BR-Sa3, and (g-h) GF-Guy.*

**Comment 9:** Fig.3 is not supposed to place at Method part, can move it into results or supplementary materials; and Fig.4 is not a contribution of this work, can move it into supplementary materials.

**Response:** *Thanks. We have moved Figure 3 and Figure 4 to the Supplementary Figures **Figure S1** and **Figure S2**, respectively, as suggested by the reviewer.*

**Comment 10:** Lines 351-355, can provide some scatterplots between Lad-LAI products and sites observations, rather than providing quantified accuracy metrics only.

**Response:** *It is a nice suggestion. We have added scatterplots between Lad-LAI*

*products and sites observations in **Figures 3-5** right panel. The scatterplots are shown as follows.*

[Figure]

**Figure R1.** *The scatterplots of simulated LAIs generated from RTSIF-derived GPP against observed LAIs at 8 camera-based observation sites across study area.*

**General Comments of Reviewer 3#:**

This paper introduces a novel dataset of age-dependent LAI for tropical and subtropic evergreen forests. Such a dataset is highly valuable and much in need to understand the dynamics of tropical canopy structure under climate change and improve the robustness of Earth System Models in reconstructing past dynamics and projecting future scenarios. The study estimated three LAI age cohorts based on a neighbor-based decomposition model and SIF-derived GPP data. The seasonality of leaf demography and its spatial variations is evaluated against ground-based measurements, and satellite observations, and analyzed with regard to other independent studies from climate controls. Results suggested a robust representation of the spatial variability in seasonality, which will be useful for improving Earth System Models. Overall, I find the dataset to be valuable and significant. I especially appreciate the authors' efforts in collecting and synthesizing ground-based observations globally to evaluate the products. However, I have some concerns regarding the robustness of the neighbor-based decomposition approach, the absence of evaluation regarding interannual dynamics, and the uncertainties in GPP estimations. I hope the authors will consider these points and provide further clarification in their responses and/or revisions. Please find my major comments and minor for clarification below.

*Response: Thanks so much for the constructive comments and suggestions regarding our manuscript. We have revised the manuscript thoroughly regarding the robustness of the neighbor-based decomposition approach, the absence of evaluation regarding interannual dynamics, and the uncertainties in GPP estimations as commented by reviewer, to*

    *(1) To prove the robustness of the neighbor-based decomposition approach, we compared the Lad-LAI products generated based 2\*2 neighbor pixels and 4\*4 neighbor pixels. The seasonality and magnitudes of LAI of young, mature and old leaf cohorts are consistent between the two datasets (**Figure 14**, **S11**). (See responses to **Comment 1**).*

    *(2) To evaluate the interannual dynamics of Lad-LAI, we could only find one ground site (Barrocolorado site in Panama) with time-series camera-based phenological imageries. Results showed that Lad-LAI could detect the interannual dynamic but more in situ observations are in need to test the robustness (**Figure 6**). (See responses to **Comment 2**).*

    *(3) To test the uncertainties caused by the GPP estimation, we added two GPP products, i.e., GOSIF-derived GPP and FLUXCOM GPP for comparisons. Results showed that the Lad-LAI generated from SIF-derived GPP show highest consistent with the in situ observed LAI seasonality of different leaf age cohorts (**Figure 17**, **R2-R4**). (See responses to **Comment 3**).*

**Major Comments:**

**Comment 1:** The approach using spatially adjacent GPP information to solve the leaf age composition is interesting but needs more justification on its robustness. With four observations (from four neighboring pixels) to solve three unknowns (LAI cohorts), the system does not have much space or tolerance for observation uncertainties (that is GPP,

please see a related comment below). I suggest providing goodness-of-fit metrics from the least squares to evaluate the model performance. However, this still may not be informative due to a limited number of observations and lack of variations between the neighboring cells. Ideally, one solution would be to include more observations (for example, by increasing the number neighboring pixels from 4 to 8) to improve the robustness and accuracy of the models, but that also means a decrease in the spatial resolution of the product.

**Response:** *Thanks for nice suggestion in testing the robustness of the neighbor-based decomposition approach. Following your comments, we have increased the number of adjacent pixels from 4 (2\*2) to 16 (4\*4) to produce another version of Lad-LAI products with spatial resolution of 0.5-degree. Then, we compared the monthly $LAI_{young}$, $LAI_{mature}$, $LAI_{old}$ between the two datasets in the 8 clustered regions. Results showed that the seasonality of $LAI_{young}$, $LAI_{mature}$, $LAI_{old}$ are highly consistent in the 8 clustered regions (**Figure 14**, **S11**), and the correlation coefficients of $LAI_{young}$, $LAI_{mature}$, $LAI_{old}$ between the two datasets are $R_{young}= 0.63$, $R_{mature}= 0.68$, $R_{old}= 0.95$, respectively, implying the robustness of neighbor-based decomposition approach in decomposing the monthly $LAI_{young}$, $LAI_{mature}$ and $LAI_{old}$ from the monthly GPP using Equation 1.*

[Figure]

**Figure 14.** *The seasonality of $LAI_{young}$, $LAI_{mature}$, $LAI_{old}$ between 0.25-degree and 0.5-degree Lad-LAI datasets in the 8 clustered regions. Limegreen color represents $LAI_{young}$; green color represents $LAI_{mature}$; and orange color represents $LAI_{old}$. Solid lines represent 0.25-degree dataset and the dashed lines represent 0.5-degree dataset.*

[Figure]

***Figure S11.*** *The scatterplot of 0.25-degree $LAI_{young}$, $LAI_{mature}$, $LAI_{old}$ against 0.5-degree LAI cohort datasets in the 8 clustered regions.*

**Comment 2:** While the age-dependent LAI product is produced at monthly time steps over 2001-2018, it has only been validated and evaluated in terms of its LAI seasonality (i.e. multi-year average climatology). The reliability and usefulness of this product in representing interannual variabilities of leaf demography are highly uncertain. Thus, I strongly encourage the authors to evaluate the interannual temporal dynamics, even if only limited, since ground observations are often insufficient. The reliability of this product in terms of representation seasonality vs. interannual variabilities should be explicitly stated in the abstract, and thoroughly discussed in the main text, to prevent misuse of the dataset. I also suggest providing LAI cohorts seasonality as the main product, and the temporal dynamics as a supplementary dataset with a clear note of usage provided along with the product.

***Response:*** *We appreciate for the reviewer's comment. We totally agree that it is important to evaluate the interannual temporal dynamics of the age-dependent LAI product. As said by the reviewer that the time-series ground observations are very limited, we could only find one ground site (Barrocolorado site in Panama) with time-series camera-based phenological imageries, to evaluate the interannual dynamics of Lad-LAI. Results showed that Lad-LAI could detect the interannual dynamic. The correlation coefficients (R) of the timeseries of $LAI_{young}$, $LAI_{mature}$, $LAI_{old}$ between the two datasets are 0.54, 0.64, 0.49, respectively (**Figure 6**). However, more in situ*

*observations are in need to test the robustness. We thoroughly discussed the timeseries variability of LAI cohort dataset. We presented the temporal variations of $LAI_{young}$, $LAI_{mature}$, $LAI_{old}$ across 8 sub-regions classified by the K-means clustering analysis (**Figure S6**). Results showed that, for example, the $LAI_{mature}$ increased significantly due to 2015 drought in Amazon basin (e.g., sub-region S2, **Figure S6**) and southeast Asia (e.g., sub-region S7, **Figure S6**), indicating good capability of detecting the dynamics of $LAI_{young}$, $LAI_{mature}$, $LAI_{old}$ in response to climate disturbances.*

[Figure]

**Figure 6.** *Timeseries of simulated $LAI_{young}$, $LAI_{mature}$, and $LAI_{old}$ in comparison with observed data at Barrocolorado site in Panama. (a) Simulations LAIs; (b) observation LAIs; and (c) scatterplots between simulated and observed LAIs.*

[Figure]

**Figure S6.** *Timeseries of simulated $LAI_{young}$, $LAI_{mature}$, and $LAI_{old}$ in 8 sub-regions classified by the K-means clustering analysis. Limegreen represents $LAI_{young}$; green represents $LAI_{mature}$; and orange represents $LAI_{old}$.*

*We also agree with the suggestion to provide the LAI cohorts seasonality as the main product and the temporal dynamics as a supplementary dataset. In addition, we provided information of data quality control (QC) for the Lad-LAI product to prevent data misuse. In the QC system (**Table S7**), data quality is divided into four levels: level 1 represents the highest quality; level 2 and level 3 represent good and acceptable quality, respectively; and level 4 warns to be used cautiously. This QC product is generated according to the goodness of fit (residual sum of squares, RSS) (Melgosa et al., 2008) obtained from the constrained least-squares method used to estimate derive monthly Lad-LAI data. Results showed that more than 92.62% of pixels are with QC at best and gool levels and only less than 5.62% are with QC at level 4 (**Figure S5**).*

**Table S7** *Information of data quality control (QC) for the Lad-LAI product*

| QC class | QC value | residual sum of squares (RSS) |
|---|---|---|
| Best | 1 | 0-1 |
| Good | 2 | 1-4 |
| Acceptable | 3 | 4-9 |
| Cautious use | 4 | >9 |

[Figure]

**Figure S5.** *Spatial patterns of seasonal quality control (QC) datasets.*

**Comment 3:** SIF-GPP relationships used to estimate GPP in this study were based on only four sites with ground observations, that may not fully represent the tropical areas over the globe. Therefore, GPP estimations from SIF are subject to high uncertainties with possibly large biases. Given that the analytical approach used to solve does not consider uncertainties, the impact of GPP estimation uncertainties on age-dependent LAI estimates should be carefully discussed.

*Response: Thank you for the valuable comment. To test the uncertainties caused by the GPP estimation, we added two more GPP products, i.e., GOSIF-derived GPP and FLUXCOM GPP, to produce two versions of Lad-LAI products, for comparisons.*

*Firstly, we need to clarify that the overall regression slope of 15.343 in the 8-day between GPP and RTSIF represent over the regional average (Chen et al., 2022), not from SIF-GPP relationships based on only four sites with ground observations. Chen*

*et al. (2022) established the linear relationship between RTSIF and GPP using 76 sites GPP data from the FLUXNET 2015 Tier 1 dataset in both 8-day and annual timescale (Fig. 8 in Chen et al. (2022)), indicating that RTSIF is tightly related to GPP. According to Chen et al. (2022), RTSIF was in good agreement with FLUXNET GPP for almost all biomes at the 8-day timescale, indicating strong SIF-GPP correlations for different biomes.*

*Second, to test the uncertainties of different SIF-GPP relationship on our analyses, we used the GOSIF-derived GPP products to produce another version of Lad-LAI. The GOSIF-derived GPP are generated based on various SIF-GPP relationships for the period from 2000 to 2020. According to Li and Xiao (2019), at site-level, the universal and biome-specific SIF-GPP relationships are established based on SIF soundings from OCO-2 and GPP data from 64 EC sites. And at grid cell level, a SIF-GPP relationship is established based on 0.05° GOSIF data and tower GPP. All these SIF-GPP relationships with different forms (universal and biome-specific, with and without intercept) at both site and grid cell levels performed well in estimating GPP globally. We also used an independent GPP product—FLUXCOM GPP products to produce a third version of Lad-LAI. The FLUXCOM GPP are estimated from machine learning to merge carbon flux measurements from FLUXNET eddy covariance towers with remote sensing and meteorological data. We compared the seasonality of three GPP datasets in 8 sub-regions classified by the K-means clustering analysis. Results showed that the GPP seasonality are mostly consistent in 8 sub-regions (**Figure S12**).*

[Figure]

***Figure S12.*** *Seasonality of RTSIF-derived GPP (yellow lines), GOSIF-derived GPP (pink lines) and FLUXCOM GPP (blue lines) datasets in 8 sub-regions classified by the K-means clustering analysis. (a-c) South America; (d-e) Congo; (f-h) tropical Asia.*

*For the three versions of Lad-LAI products, eight camera-based observation sites are used for compare the accuracy of the corresponding simulated LAI cohorts*

*(**Figures R2-R4**). We also compared the seasonal variability between three versions products in 8 sub-regions classified by the K-means clustering analysis (**Figure 17**). Results showed that the Lad-LAI generated from RTSIF-derived GPP show highest consistent with the in situ observed LAI seasonality of different leaf age cohorts (**Figures R2-R4**). The highest accuracies of the seasonality of $LAI_{young}$, $LAI_{mature}$, $LAI_{old}$ between the observed sites and the three datasets are all come from the Lad-LAI generated from RTSIF-derived GPP, $R_{young\ vs\ RTSIF\text{-}derived\ GPP}$= 0.36, $R_{mature\ vs\ RTSIF\text{-}derived\ GPP}$ = 0.77, $R_{old\ vs\ RTSIF\text{-}derived\ GPP}$ = 0.59, respectively.*

*In general, three versions of Lad-LAI products all performed well in 8 sub-regions with the consistent seasonal variability (**Figure 17**). On regional average, sub-regions S4, S5, S6, S7 and S8 show high consistent seasonality of $LAI_{young}$, $LAI_{mature}$, and $LAI_{old}$ between these three products; whereas the Lad-LAI generated from GOSIF-derived GPP performs a little poor in capturing the seasonality of LAI cohorts in Amazon (sub-regions S1, S2 and S3).*

[Figure]

**Figure R2.** *Seasonality of simulated $LAI_{young}$, $LAI_{mature}$, and $LAI_{old}$ in comparison with observed data at 4 sites in south America. (Panels a, e, i and m) simulated LAIs from RTSIF-derived GPP; (panels b, f, j and n) camera-based observed LAIs; (panels c, g, k and o) simulated LAIs from GOSIF-derived GPP; and (panels d, h, l and p) simulated LAIs from FLUXCOM GPP.*

[Figure]

**Figure R3.** *Seasonality of simulated $LAI_{young}$, $LAI_{mature}$, and $LAI_{old}$ in comparison with observed data at one site in Congo. (a) Simulated LAIs from RTSIF-derived GPP; (b) camera-based observed LAIs; (c) simulated LAIs from GOSIF-derived GPP; and (d)*

*simulated LAIs from FLUXCOM GPP.*

[Figure]

***Figure R4.*** *Seasonality of simulated LAI_{young}, LAI_{mature}, and LAI_{old} in comparison with observed data at 3 sites in tropical Asia. (Panels a, e and i) simulated LAIs from RTSIF-derived GPP; (panels b, f and j) camera-based observed LAIs; (panels c, g and k) simulated LAIs from GOSIF-derived GPP; and (panels d, h and l) simulated LAIs from FLUXCOM GPP.*

[Figure]

***Figure 17.*** *Seasonality of simulated LAI_{young}, LAI_{mature}, and LAI_{old} from three version products in 8 sub-regions classified by the K-means clustering analysis. Solid lines represent LAI generated from RTSIF-derived GPP; dashed lines represent LAI generated from GOSIF-derived GPP; and dotted lines represent LAI generated from FLUXCOM GPP. Limegreen represents LAI_{young}; green represents LAI_{mature}; and orange represents LAI_{old}.*

**Comment 4:** Please note that evaluation against EVI is not entirely independent, since the RT-SIF dataset was a reconstruction from MODIS NBAR surface reflectance data.

*Response: Thanks for pointing this out. To be cautious, we have removed the statements of using "independent" in the revised manuscript. For the capability of using EVI as a proxy for validating the seasonality young and mature leaves, Huete et al. (2006) found that Amazon rainforests green-up in dry season due to sunlight derive the synchronous canopy leaf turnover the young and mature leaves. And de Moura et al. (2017) compared tower and MODIS data with leaf flush and LAI from young to old leaves, and found an EVI increase toward September that closely tracked the modeled LAI of young/mature leaves (3–5 months). The MODIS EVI products are very sensitive to changes in NIR reflectance (Galvão et al., 2011) and young and mature leaves also could reflect more near-infrared (NIR) light than the older leaves replaced (Toomey et al., 2009). We have added such explanation in the new version.*

*"It is because that EVI are very sensitive to changes in near-infrared (NIR) reflectance (Galvão et al., 2011) while young and mature leaves also reflect more NIR signals than the older leaves they replace (Toomey et al., 2009)." (In revision lines 497-500).*

**Comment 5:** The manuscript needs improvements in language and grammar. I suggest carefully revising it to improve clarity.

*Response: For the improvements in language and grammar of the manuscript, we totally rewrote the Introduction (see responses to **Comment 9** and **Review 2# Comment 4**), largely revised the Study area and data sections (see responses to **Review 2# Comment 5**). Finally, we also asked a company to polish our English language, including grammar, syntax, and sentence structure, to improve the readability of the manuscript.*

**Specific Comments:**

Abstract:

**Comment 6:** Please specify the temporal span, temporal and spatial resolution of the LAI product.

*Response: Thanks for your comment. We revised it as suggested. The abstract revised as follows:*

*"Here, we simplified the canopy leaves of TEFs into three age cohorts (i.e., young, mature and old one with different photosynthesis capacity ($V_{c,max}$)) and proposed a novel neighbor-based approach to develop a first grid dataset of monthly leaf age-dependent LAI product (**referred to as Lad-LAI**) at 0.25-degree spatial resolution over the continental scale during 2001-2018 from satellite observations of sun-induced chlorophyll fluorescence (SIF) that was reconstructed from MODIS and TROPOMI (the TROPOspheric Monitoring Instrument)." (In revision lines 32-39)*

**Comment 7:** L36: It should be noted that this is a SIF dataset that was reconstructed from MODIS and TROPOMI to avoid confusion.

*Response: Thanks for your reminder. We have corrected it. (See responses to **Comment 6**)*

**Comment 8:** L40-41: Since the RTSIF is reconstructed from MODIS surface reflectance data, the evaluation against EVI is not precisely "independent".
***Response:*** *Thanks again. We have removed the statements of using "independent" in the revised manuscript.*

Introduction:
**Comment 9:** L103: The last paragraph of the Introduction should be shortened with a brief summary of the method and findings.
***Response:*** *Done as suggested. We have shortened this paragraph with a brief summary of the method and findings as follows.*

*"To fill the research gap, this study aims to produce a global grid dataset of leaf age-dependent LAI seasonality product (Lad-LAI) over the whole TEF biomes from 2001 to 2018. For this purpose, we first simplified that canopy GPP was composed of three parts that were produced from young, mature and old leaves, respectively. GPP was then expressed as a function of the sum of the product of each LAI cohort (i.e., young, mature and old leaves, denoted as $LAI_{young}$, $LAI_{mature}$, and $LAI_{old}$, respectively) and corresponding net $CO_2$ assimilation rate (An, denoted as $An_{young}$, $An_{mature}$, and $An_{old}$ for young, mature and old leaves, respectively) (**Equation 1**). Then, we proposed a novel neighbor-based approach to derive the values of three LAI cohorts. It was hypothesized that forests in adjacent four cells in the grid map exhibited consistent seasonality in both GPP, and LAI cohorts ($LAI_{young}$, $LAI_{mature}$, and $LAI_{old}$). Based on this assumption, we applied **Equation 1** to each pixel and combined the four equations of 2\*2 neighboring pixels to derive the three LAI cohorts using a linear least-squares with constrained method. An was calculated using the Farquhar-von Caemmerer-Berry (FvCB) leaf photochemistry model (Farquhar et al., 1980); and GPP was linearly derived from an arguably better proxy—TROPOMI (the TROPOspheric Monitoring Instrument) Solar-Induced Fluorescence (SIF) based on a simple SIF-GPP relationship established by Chen et al. (2022) (see **Methods** for details). This grid dataset of three LAI cohorts provides new insights into tropical and subtropical phenology with more details of sub-canopy level of leaf seasonality in different leaf age cohorts and will be helpful for developing accurate tropical phenology model in ESMs."*

Method:
**Comment 10:** L132-133: How much are the spatial variations in the constant LAI value?
***Response:*** *We analyzed the measured LAI values from previously published literatures and found there are slightly spatial and seasonal variations in totally LAI (Figure S3, S4). A constant total LAI value (around 6.0) can be used for most evergreen broadleaf forests.*

[Figure]

*Figure S3. The distribution map of measured LAI sites from previously published literatures.*

[Figure]

*Figure S4. The seasonality of observed total LAI values from previously published literatures.*

**Comment 11:** L147-168: Using GPP-SIF relationships based on only four sites is suspect to extrapolation issues over the entire areas.

***Response:*** *We apologies for this mistake describe. We have corrected it. The revised as follows:*

*"The grid GPP data over the whole EBFs were derived from SIF (denoted as RTSIF-derived GPP) using a linear SIF-GPP regression model (see sect. 3.2) which were established based on in situ GPP from 76 eddy covariance (EC) sites (Chen et al., 2022)." (In revision lines 219-221)*

**Comment 12:** L155: VPD data sources are different between Table S3 and Figure 2. ERA5-Land is at 0.1 degree instead of 0.05 deg? Can you double check?

***Response:*** *Thank you for your attention to detail. VPD datasets was calculated from ERA Interim datasets. And the spatial resolution of air temperature datasets from ERA5-Land is at 0.1 degree. We corrected it in revised Supplementary **Table S3**.*

**Comment 13:** L175: Could you please provide the GPP-SIF relationship equation and overall goodness-of-fit?

***Response:*** *Yes. The overall regression slope of 15.343 in the 8-day between GPP and RTSIF represent the regional average, which was provided by Chen et al., 2022, not from SIF-GPP relationships based on only four sites with ground observations. Chen et al. (2022) explored the relationship between RTSIF and GPP using 76 sites GPP data from the FLUXNET 2015 Tier 1 dataset, and found that there is a linear relationship between RTSIF and GPP in both 8-day and annual timescale (Fig. 8 in Chen et al. (2022)), indicating that RTSIF is tightly related to GPP. And they also*

*reported RTSIF was in good agreement with FLUXNET GPP for almost all biomes at the 8-day timescale, indicating strong SIF-GPP correlations for different biomes. We also provided the GPP-SIF relationship equation and goodness-of-fit (R) at each eddy covariance (EC) tower site in* **Figure S1 a, c, e,** *and* **g** *plots.*

[Figure]

**Figure S1.** *Comparisons between monthly RTSIF-derived GPP (red) and observed GPP at eddy covariance (EC) tower sites (blue). (a-b) Au-Rob, (c-d) BR-Sa1, (e-f) BR-Sa3, and (g-h) GF-Guy.*

**Comment 14:** L270-271: Note that the RTSIF product is reconstructed from MODIS using the short-term TROPOMI data as a training set. Therefore, the evaluation against EVI is not independent.
***Response:*** *Yes, we agree with your comment and appreciate your reminder. To be cautious, we have removed the statements of using "independent" in the revised manuscript.*

**Comment 15:** L273: Can you please elaborate on how EVI reflects young and mature leaves, not old ones?
***Response:*** *Previous studies which used independent satellite observations from lidar*

*and optical sensors reported a consistent phenomenon ― dry-season greening in Amazon forests (Saleska et al., 2007; Huete et al., 2006; Myneni et al., 2007). And one of the potential biophysical mechanisms of this seasonal greening in Amazon forests is synchronous canopy leaf turnover (Huete et al., 2006; Brando et al., 2010; Doughty et al., 2008) and young leaves flushing. The young leaves could reflect more near-infrared (NIR) light than the older leaves replaced (Toomey et al., 2009). The MODIS EVI products are very sensitive to changes in NIR reflectance (Galvão et al., 2011). As results, when MODIS EVI products were corrected for these effects using the Multi-Angle Implementation of Atmospheric Correction Algorithm (MAIAC), an EVI increase toward September that closely tracked the modeled LAI of young/mature leaves (3–5 months) (de Moura et al., 2017). We have added such explanation in the new version.*

*"It is because that EVI are very sensitive to changes in near-infrared (NIR) reflectance (Galvão et al., 2011) while young and mature leaves also reflect more NIR signals than the older leaves they replace (Toomey et al., 2009)." (In revision lines 497-500).*

**Comment 16:** L274: Specify MSD

*Response: MSD is the abbreviation for Mean Squared Deviation. The analysis of MSD clearly identified the simulation vs. measurement contrasts with larger deviation than others; the correlation–regression approach tended to focus on the contrasts with lower correlation and regression line far from the equality line. It was shown results of the MSD-based analysis were easier to interpret than those of regression analysis. This is because the three MSD components are simply additive and all constituents of the MSD components are explicit. This approach will be useful to quantify the deviation of calculated values obtained with this model from measurements. In revision lines 330-331, we have specified MSD.*

**Comment 17:** Figure S1: the figure is too blur to read.

*Response: We divided into 3 classes for all those sites by region, south America, Congo, tropical Asia. Thanks.*

[Figure]

***Figure S8.*** *Seasonality of LAI$_{young}$, LAI$_{mature}$, LAI$_{old}$, litterfall, RTSIF-derived GPP, EVI, T$_{air}$, VPD and SW at 22 sites in south America.*

[Figure]

***Figure S9.*** *Seasonality of LAI$_{young}$, LAI$_{mature}$, LAI$_{old}$, litterfall, RTSIF-derived GPP, EVI, T$_{air}$, VPD and SW at 7 sites in Congo.*

[Figure]

***Figure S10.*** *Seasonality of LAI_young, LAI_mature, LAI_old, litterfall, RTSIF-derived GPP, EVI, T_air, VPD and SW at 24 sites in tropical Asia.*

**Comment 18:** L326: Please specify which variable (x,y) is estimated or observed.

***Response:*** *Thanks. In original manuscript L326, in general, simulated value for LAI is denoted as X, and measured value is denoted as Y. Specifically, in original manuscript L351-355, to quantify the sites accuracy, MSD was calculated by X as estimated and Y as observed. In original manuscript Figure 8, to calculate MSD (LAI_young+mature & EVI), X is LAI_young+mature and Y is EVI. We have added such explanation in the new version.*

*"and $x_i$ is the simulated data at time t, and $y_i$ is the observed one at time t (month)." (In revision lines 377-378)*

Result:

**Comment 19:** Figure 5: It's not clear which is estimated versus observed data.

***Response:*** *Thanks for your comments. We have clarified in Figure 5 caption that the left column represents the simulated values, the middle column represents the observed values, and the right column shows the scatterplot.*

[Figure]

***Figure 3***. *Seasonality of simulated LAI$_{young}$, LAI$_{mature}$, and LAI$_{old}$ in comparison with observed data at 4 sites in south America. (Panels a, d, g and j) simulated LAIs; (panels b, e, h and k) observed LAIs; (panels c, f, i and l) scatterplots between simulated and observed LAIs. Limegreen dots are LAI$_{young}$; green dots are LAI$_{mature}$; orange dots are LAI$_{old}$.*

[Figure]

***Figure 4***. *Seasonality of simulated LAI$_{young}$, LAI$_{mature}$, and LAI$_{old}$ in comparison with observed data at one site in Congo. (a) Simulated LAIs; (b) observed LAIs; and (c) scatterplots between simulated and observed LAIs. Limegreen dots are LAI$_{young}$; green dots are LAI$_{mature}$; orange dots are LAI$_{old}$.*

[Figure]

***Figure 5.*** *Seasonality of simulated LAI$_{young}$, LAI$_{mature}$, and LAI$_{old}$ in comparison with observed data at 3 sites in tropical Asia. (Panels a, d and g) simulated LAIs; (panels b, e and h) observed LAIs; (panels c, f and i) scatterplots between simulated and observed LAIs. Limegreen dots are LAI$_{young}$; green dots are LAI$_{mature}$; orange dots are LAI$_{old}$.*

**Comment 20:** L355-357: This sentence is a bit unclear. Can you elaborate on the "trade-off"?

***Response:*** *Yes. In tropical and subtropical evergreen broadleaved forests, trees adapt their leaf phenology to avoid unfavorable environments such as limited light and water, and maximize their growth rate (Kikuzawa 1995; Vico et al., 2015). And the "trade-off" between the phenology of mature and old leaves means that these forests exhibit complex leaf shedding and rejuvenation strategies in response to moisture and light availability, and these strategies depend on soil water, atmospheric vapor pressure deficit, and incoming solar radiation. Specifically, leaf shedding in the dry season may be an adaptive response to soil water deficits (Asner et al., 2010; Brando et al., 2010) or atmospheric aridity (Xu et al., 2017). Alternatively, leaf shedding in non-water-limited conditions may constitute an adaptive strategy to replace senescent leaves with efficient young leaves to maximize photosynthesis (Chen et al., 2020). To avoid unclear description, we no longer use "trade-off" in revised manuscript.*

**Comment 21:** L359-360: Should one of the "early wet season" be "dry season"?

***Response:*** *Yes, we apologies for this mistake. The first one should be "early dry season". We corrected it in revised as follow: "In south America, at K67, K34 and Eucflux sites, both in situ and simulated LAI$_{young}$ and LAI$_{mature}$ decrease at early dry season around February and convert to increase at early wet season around June (**Fig. 3 a, b, d, e, j, k**)." (Lines 408-411 in revised manuscript)*

**Comment 22:** L397: Chen et al., 2019 is not found in the reference list.

*Response: Thank you for your attention to detail. "Chen et al., 2019" in caption of Figure 6 actually corresponds to this one in the reference list: Chen, X., Ciais, P., Maignan, F., Zhang, Y., Bastos, A., Liu, L., Bacour, C., Fan, L., Gentine, P., Goll, D., Green, J., Kim, H., Li, L., Liu, Y., Peng, S., Tang, H., Viovy, N., Wigneron, J. P., Wu, J., Yuan, W., and Zhang, H.: Vapor Pressure Deficit and Sunlight Explain Seasonality of Leaf Phenology and Photosynthesis Across Amazonian Evergreen Broadleaved Forest, Global Biogeochemical Cycles, 35, 10.1029/2020gb006893, 2021. We have corrected the mistake cite in the revised version and checked and confirmed all the references.*

**Comment 23:** L395: Is it possible to keep a consistent number of clusters between the three datasets? For example, can you set eight clusters in Lad-LAI, so the southeast Asia area has three clusters consistent with plots d-f. This will make it easier to compare the datasets.

*Response: Thank you for your constructive suggestion. We have updated the southeast Asia area to have three clusters consistent with plots d-f in Lad-LAI dataset, in order to make it easier to compare the datasets. The corresponding statistic figures has been updated accordingly.*

[Figure]

***Figure 7.*** *Comparison of sub-regions of Lad-LAI products (plots g-i) with those of climatic factors classified by the K-means clustering analysis (plots a-c) (Chen et al., 2021) and those of the three climate-phenology regimes (plots d-f) developed by Yang et al. (2021).*

[Figure]

**Figure 8.** *Seasonality of simulated LAI_young, LAI_mature, and LAI_old in 8 sub-regions classified by the K-means clustering analysis.*

[Figure]

**Figure 10.** *Statistics of the Pearson correlation coefficient (R) between seasonality of simulated LAI_{young+mature} and MODIS Enhanced Vegetation Index (EVI) in the 8 clustered sub-regions.*

[Figure]

***Figure 11.*** *Statistics of the mean squared deviation (MSD) between seasonality of simulated LAI$_{young+mature}$ and MODIS Enhanced Vegetation Index (EVI) in the 8 clustered sub-regions.*

**Comment 24:** L413: I wonder if you have any hypothesis for the low performance in southeast Asia in comparison with other regions? (Figure 8a-c)

***Response:*** *Yes. Compared to tropical evergreen forests of the Amazon and Africa, tropical Asian forests exhibit the lowest sensitivity of solar-induced chlorophyll fluorescence (SIF) due to the combined effects of very few dry-season observations and more pronounced cloudiness effects (Guan et al., 2015; 2016). We have added such explanation in the new version.*

*"This happens because that the accuracy of Lad-LAI in representing the seasonality of LAI cohorts depends highly on that of input SIF data, which is low sensitive to canopy phenology and shows marginally small seasonal changes nearby the Equator, for example in tropical Asia (Guan et al., 2015; 2016)." (In revision lines 516-520)*

**Comment 25:** Figure 12: Please increase font size. It's not clear which line represents site data. Can you also illustrate the meaning of the dots?

***Response:*** *Thank you for your comment. In the revised version, we have increased the font size of Figure 13 in new version and added a legend to clarify which lines represent the site data. The orange dots in plots a-i represent the point with an abrupt decrease in LAI$_{old}$ and the black dots in plots a-i represent the point with litterfall peak. The black dots in plot j (right panel) represent the days when LAI$_{old}$ has an abrupt decrease (Day$_{LAIold}$) against the days when monthly litterfall peaks (Day$_{litterfall}$).*

[Figure]

**Figure 13.** *Evaluation of simulated LAI_old using ground-observed litterfall seasonality. (a-i) Days of an abrupt decrease in LAI_old in comparison with days of corresponding litterfall peak at 9 specific sites for examples. The orange curves represent simulated LAI_old. Dots on the orange curves represent the point with an abrupt decrease in LAI_old. The black curves represent observed seasonal litterfall mass. The dots on the black curves represent the point with litterfall peak. (j) Comparisons of the days when LAI_old has an abrupt decrease (Day_LAIold) against the days when monthly litterfall peaks (Day_litterfall).*

---

## Author Response (AR2)

**Responses to editor's and reviewers' comments point by point**

MS No.: essd-2022-436

Title: A gridded dataset of leaf age-dependent LAI seasonality product (Lad-LAI) over tropical and subtropical evergreen broadleaved forests

Author(s): Xueqin Yang et al.

**Comment of Topical Editor:**

Topical editor decision: Publish subject to minor revisions (review by editor)

Public justification (visible to the public if the article is accepted and published):

The reviewers are satisfied with the authors' revision and provide few comments/suggestions for the authors' consideration. Please carefully resolve these comments.

*Response: Thanks for your valuable time in handling our manuscript; and thanks also for the constructive comments and suggestions from two excellent reviewers. We discussed the accuracy of leaf-age dependent LAI product generated from CSIF-derived GPP and the performance of NDVI seasonality in representing tropical young leaf phenology. We also added the root-mean-square error (RMSE) to establish the quality control (QC) band. Finally, we conducted a thorough proofreading and corrected some grammar issues to improve the readability of the manuscript. Below, we address each point in detail.*

**General Comments of Reviewer 1#:**

Thanks authors' work and effort for addressing my questions and comments. The manuscript and figures look better, but I still have few minor issues after I read the new version:

*Response: We sincerely appreciate your thorough review of the manuscript. Your comments and suggestions are invaluable for the revised manuscript. Please see our point-to-point responses to your comments below.*

**Minor issues:**

**Comment 1:**

(1) Abstract is too long. It should be highly shortened and simplified.

*Response: Thanks for the comments on the abstract. We have shortened and simplified it as follows:*

*Quantification of large-scale leaf age-dependent leaf area index has been lacking in tropical and subtropical evergreen broadleaved forests (TEFs) despite the recognized importance of leaf age in influencing leaf photosynthetic capacity in this biome. Here, we simplified the canopy leaves of TEFs into three age cohorts (i.e., young, mature and old one with different photosynthesis capacities ($V_{c,max}$)) and proposed a novel neighbor-based approach to develop the first gridded dataset of monthly leaf age-dependent LAI product (**referred to as Lad-LAI**) at 0.25-degree spatial resolution over the continental scale during 2001-2018 from satellite observations of sun-induced chlorophyll fluorescence (SIF) that was reconstructed from MODIS and TROPOMI (the TROPOspheric Monitoring Instrument). The new Lad-LAI products show good*

*performance in capturing the seasonality of three LAI cohorts, i.e., young (LAIyoung)* *(the Pearson correlation coefficient, R=0.36), mature (LAImature) (R=0.77) and old (LAIold) (R=0.59) leaves at eight camera-based observation sites (four in south America, three in subtropical Asia and one in Congo) and can also represent their interannual dynamics, validated only at the Barrocolorado site with R being equal to 0.54, 0.64 and 0.49 for LAIyoung, LAImature and LAIold, respectively. Additionally, the abrupt drops in LAIold are mostly consistent with the seasonal litterfall peaks at 53 in situ measurements across the whole tropical region (R=0.82). The LAI seasonality of young and mature leaves also agrees well with the seasonal dynamics of Enhanced Vegetation Index (EVI) (R=0.61), a proxy of photosynthetically effective leaves. Spatially, the gridded Lad-LAI data capture a dry-season green-up of canopy leaves across the wet Amazonia areas where mean annual precipitation exceeds 2,000 mm yr$^{-1}$, consistent with previous satellite-based analyses. The spatial patterns clustered from the three LAI cohorts also coincide with those clustered from climatic variables over the whole TEF region. Herein we provide the average seasonality of three LAI cohorts as the main dataset, and their time-series as a supplementary dataset. These Lad-LAI products are available at https://doi.org/10.6084/m9.figshare.21700955.v4 (Yang et al., 2022).*

**Comment 2:**

(2) Authors used TROPOMI SIF to produce GPP for decomposing LAI cohorts, why not to use CSIF (doi.org/10.5194/bg-15-5779-2018)? Or whether the results may be different when using different SIF products.

***Response:*** *Thanks for your suggestions and recommendation. We compared the seasonality of RTSIF and CSIF, and found that CSIF data usually had smaller values than RTSIF ones but had greater fluctuation amplitude at Congoflux site, which was likely strongly dependent on incoming solar radiation (Figure R1). Then, as suggested by the reviewer, we used CSIF-derived GPP to generate another version of LAI cohorts product. And we compared the seasonality of LAIyoung, LAImature and LAIold products generated from CSIF-derived GPP with the camera-based observation at the 8 sites over the whole tropical forests, respectively. Results showed that the Lad-LAI generated from CSIF-derived GPP performed less well as those from other GPP products. (Figures R2, R3).*

[Figure]

***Figure R1.*** *Comparison of the seasonality of RTSIF and CSIF at 8 sites. (a) K67; (b) K34; (c) Barrocolorado; (d) Eucflux; (e) Din; (f) Gutian; (g) Banna; (h) Congoflux.*

[Figure]

***Figure R2.*** *Seasonality of simulated LAI$_{young}$, LAI$_{mature}$, and LAI$_{old}$ in comparison with observed data at 4 sites in south America. (Panels a, d, g and j) simulated LAIs from RTSIF-derived GPP; (panels b, e, h and k) camera-based observed LAIs; and (panels c, f, i and l) simulated LAIs from CSIF-derived GPP*

[Figure]

***Figure R3.*** *Seasonality of simulated LAI$_{young}$, LAI$_{mature}$, and LAI$_{old}$ in comparison with observed data at one site in Congo. (a) Simulated LAIs from RTSIF-derived GPP; (b) camera-based observed LAIs; and (c) simulated LAIs from CSIF-derived GPP*

**Comment 3:**

(3) EVI cannot capture greenness seasonality well, particularly for the tropical areas (doi:10.1038/nature13006). Thus, I suggested to add NDVI to analyze the relationship between the LAI seasonality and seasonal dynamics of NDVI.

***Response:*** *Thanks for your comments and recommendation. We compared the seasonality of three versions of NDVI (GIMMS3g NDVI, MOD13C2 NDVI and MYD13C2 NDVI) and EVI (MYD13C2 EVI) at eight camera-based observation sites. Results showed that three versions of NDVI were highly consistent. However, the seasonal dynamics of NDVI at the site level were not as effective as those of EVI for*

*representing seasonality of LAI$_{young+mature}$ (Figure R4). And, several studies indicated EVI performs be a better proxy for tropical leaf phenology (Guan et al., 2015). Therefore, we believe that it might not be suitable to add NDVI for validating the results of LAI cohorts.*

[Figure]

***Figure R4.** Comparison of the seasonality of LAI$_{young+mature}$ observations and vegetation indexs (VIs) at eight camera-based observation sites. Green lines with circle markers present LAI observations; blue lines with short vertical bar markers present GIMMS3g NDVI; blue lines with square box markers present MOD13C2 NDVI; blue lines with circle markers present MOD13C2 NDVI; and olive lines with triangle markers present EVI. (a) K67; (b) K34; (c) Barrocolorado; (d) Eucflux; (e) Din; (f) Gutian; (g) Banna; (h) Congoflux*

***Reference:***

Guan, K., Pan, M., Li, H., Wolf, A., Wu, J., Medvigy, D., Caylor, K. K., Sheffield, J., Wood, E. F., Malhi, Y., Liang, M., Kimball, J. S., Saleska, Scott R., Berry, J., Joiner, J., and Lyapustin, A. I.: Photosynthetic seasonality of global tropical forests constrained by hydroclimate. Nature Geoscience, 8(4), 284-289. https://doi.org/10.1038/ngeo2382, 2015.

Morton, D. C., Nagol, J., Carabajal, C. C., Rosette, J., Palace, M., Cook, B. D., Vermote, E. F., Harding, D. J., and North, P. R.: Amazon forests maintain consistent canopy structure and greenness during the dry season. Nature, 506(7487), 221-224. https://doi.org/10.1038/nature13006. 2014.

Zhang, Y., Joiner, J., Alemohammad, S. H., Zhou, S., and Gentine, P.: A global spatially contiguous solar-induced fluorescence (CSIF) dataset using neural networks. Biogeosciences, 15(19), 5779-5800. https://bg.copernicus.org/articles/15/5779/2018/. 2018.

**General Comments of Reviewer 2#:**

I would like to thank the authors for their careful attention to my comments and meticulous efforts in improving the manuscript. I especially appreciate the inclusion of a quality control band associated with the product, which is highly useful. All my concerns and suggestions have been thoroughly addressed, and the manuscript has been substantially improved. I believe this paper and the Lad-LAI dataset have made an

impactful contribution to understanding tropical ecosystem dynamics.

***Response:*** *We would like to express our gratitude for your time and expertise in reviewing our work. Your comments and suggestions are very valuable for the revised manuscript. For the minor comments raised by the reviewer, the point-to-point responses are listed below.*

**Minor comments:**

**Comment 4:**

1. Figure S11: Note that the x-axis label should probably be LAI. It would also be helpful to indicate the resolution of LAI in the x and y axes.

***Response:*** *Thanks for your carefulness and nice suggestions. We have corrected the x-axis label as "$LAI_{0.25\text{-}degree}$ ($m^2\ m^{-2}$)" and revised y-axis label as "$LAI_{0.5\text{-}degree}$ ($m^2\ m^{-2}$)".*

[Figure]

***Figure S13.*** *The scatterplot of 0.25-degree $LAI_{young}$, $LAI_{mature}$, $LAI_{old}$ against 0.5-degree LAI cohort datasets in the 8 clustered regions.*

**Comment 5:**

2. One suggestion for the QC band is to provide an error metric in the unit of LAI, such as RMSE, in addition to RSS. This would make the QC values more easily interpretable and informative.

***Response:*** *Thanks for the valuable comments and nice suggestions. We added the root-mean-square error (RMSE) (Chen et al., 2020) as an error metric in the unit of LAI to establish QC classes with RSS (Melgosa et al., 2008) (Table S7). The new QC band is*

generated according to RSS and RMSE, obtained from the constrained least-squares method that was used to estimate and derive monthly Lad-LAI data. Results showed that more than 92.62% of pixels are with QC at best and gool levels and only less than 5.62% are with QC at level 4 (Figure S5).

**Table S7.** *Information of data quality control (QC) for the Lad-LAI product*

| QC class | QC value | RSS | RMSE ($m^2 \ m^{-2}$) |
|---|---|---|---|
| Best | 1 | 0-1 | 0-1 |
| Good | 2 | 1-4 | 1-2 |
| Acceptable | 3 | 4-9 | 2-3 |
| Cautious use | 4 | >9 | >3 |

[Figure]

**Figure S5.** *Spatial patterns of seasonal quality control (QC) datasets.*

**Comment 6:**
3. The abstract has been greatly improved, but it appears lengthy. I think the authors could consider omitting the discussion about GOSIF/FLUXCOM and the analysis of neighboring window size. These analyses primarily serve as verifications of the methodology rather than the main findings.

*Response: Thanks for the valuable comments and nice suggestions. We agreed with the reviewer's suggestion to omit the discussion about GOSIF/FLUXCOM and the analysis of neighboring window size from the abstract. We have removed those discussions as follows:*

*Quantification of large-scale leaf age-dependent leaf area index has been lacking in tropical and subtropical evergreen broadleaved forests (TEFs) despite the recognized importance of leaf age in influencing leaf photosynthetic capacity in this biome. Here, we simplified the canopy leaves of TEFs into three age cohorts (i.e., young, mature and old one with different photosynthesis capacities ($V_{c,max}$)) and proposed a novel neighbor-based approach to develop the first gridded dataset of monthly leaf age-dependent LAI product (**referred to as Lad-LAI**) at 0.25-degree spatial resolution over the continental scale during 2001-2018 from satellite observations of sun-induced chlorophyll fluorescence (SIF) that was reconstructed from MODIS and TROPOMI (the TROPOspheric Monitoring Instrument). The new Lad-LAI products show good performance in capturing the seasonality of three LAI cohorts, i.e., young ($LAI_{young}$) (the Pearson correlation coefficient, R=0.36), mature ($LAI_{mature}$) (R=0.77) and old ($LAI_{old}$) (R=0.59) leaves at eight camera-based observation sites (four in south America, three in subtropical Asia and one in Congo) and can also represent their interannual dynamics, validated only at the Barrocolorado site with R being equal to 0.54, 0.64 and 0.49 for $LAI_{young}$, $LAI_{mature}$ and $LAI_{old}$, respectively. Additionally, the abrupt drops in $LAI_{old}$ are mostly consistent with the seasonal litterfall peaks at 53 in situ measurements across the whole tropical region (R=0.82). The LAI seasonality of young and mature leaves also agrees well with the seasonal dynamics of Enhanced Vegetation Index (EVI) (R=0.61), a proxy of photosynthetically effective leaves. Spatially, the gridded Lad-LAI data capture a dry-season green-up of canopy leaves across the wet Amazonia areas where mean annual precipitation exceeds 2,000 mm $yr^{-1}$, consistent with previous satellite-based analyses. The spatial patterns clustered from the three LAI cohorts also coincide with those clustered from climatic variables over the whole TEF region. Herein we provide the average seasonality of three LAI cohorts as the main dataset, and their time-series as a supplementary dataset. These Lad-LAI products are available at https://doi.org/10.6084/m9.figshare.21700955.v4 (Yang et al., 2022).*

**Comment 7:**
4. The introduction has been substantially enhanced and now reads very well! However, I noticed that there are still some minor grammar errors in the abstract, result, discussion and conclusion sections. Therefore, an additional grammar check may be necessary. Here are a few minor issues that I noted. Please note that this might not be an exhaustive list.
***Response:*** *Thanks, we apologize for those minor grammar errors. We have conducted an additional grammar check to rectify these issues.*

**Comment 8:**
Line 35: "a first grid" -> "the first gridded" dataset
***Response:*** *Thanks, we revised it as suggested.*

**Comment 9:**
Line 41: specify "R"
***Response:*** *In revision line 47, we have specified R.*

**Comment 10:**
Line 48: "which is a good proxy" -> a proxy of photosynthetically effect leaves?
*Response: Thanks. We revised it as suggested. (In revision line 55)*

**Comment 11:**
L192: photosynthetically
*Response: Corrected.*

**Comment 12:**
L392: Remove "to warn potential uncertainties" or revise it to improve clarity
*Response: Thanks. To be more cautious, we removed this part in the revised manuscript.*

**Comment 13:**
L547: remove "quite"
*Response: Thanks, removed.*

**Comment 14:**
L651: potential grammar issue
*Response: We have revised this sentence to avoid potential grammar issues. The revised sentence was as follows:*

*It should be noted that over the regions with a large magnitude of annual precipitation nearby the Equator, there are no obvious dry seasons, and thus tree canopy phenology changes are smaller than higher-latitude ones throughout the year (Yang et al., 2021). (In revision lines 652-655)*

**Comment 15:**
L656: "unexpected" -> "additional"
*Response: Done.*

**Comment 16:**
L701: "convert" -> "start"
*Response: Done.*

**Comment 17:**
L707: "help" -> "helpful"
*Response: Done.*

***Reference:***
*Chen, X., Maignan, F., Viovy, N., Bastos, A., Goll, D., Wu, J., Liu, L., Yue, C., Peng, S., Yuan, W., Conceição, A. C., O'Sullivan, M., and Ciais, P.: Novel Representation of Leaf Phenology Improves Simulation of Amazonian Evergreen Forest Photosynthesis in a Land Surface Model, Journal of Advances in Modeling Earth Systems, 12, 10.1029/2018ms001565, 2020.*

Melgosa, M., Huertas, R., and Berns, R. S.: Performance of recent advanced color-difference formulas using the standardized residual sum of squares index. *Journal of the Optical Society of America A*, 25(7), 1828-1834. https://doi.org/10.1364/JOSAA.25.001828. 2008

Yang, X., Wu, J., Chen, X., Ciais, P., Maignan, F., Yuan, W., Piao, S., Yang, S., Gong, F., Su, Y., Dai, Y., Liu, L., Zhang, H., Bonal, D., Liu, H., Chen, G., Lu, H., Wu, S., Fan, L., Gentine, P., and Wright, S. J.: A comprehensive framework for seasonal controls of leaf abscission and productivity in evergreen broadleaved tropical and subtropical forests, *Innovation (Camb)*, 2, 100154, 10.1016/j.xinn.2021.100154, 2021.

---

## Author Response (AR3)

**Responses to editor's comments point by point**

MS No.: essd-2022-436

Title: A gridded dataset of leaf age-dependent LAI seasonality product (Lad-LAI) over tropical and subtropical evergreen broadleaved forests

Author(s): Xueqin Yang et al.

**Comment of Topical Editor:**

Topical editor decision: Publish as is

Public justification (visible to the public if the article is accepted and published):

Thank the authors for their efforts for improving their manuscript! Looking forward to more exciting work from the authors on this topic (i.e., extend to other PFTs, and even global forest)!

*Response: Thank the editor and two excellent reviewers for your valuable time in handling our manuscript. We have successfully uploaded all the required files. We fully agree with your suggestion and are willing to put in continuous efforts to implement it.*